# Modeling Mixing and Melting in Laminar Seawater Intrusions Under Grounded Ice

Madeline S. Mamer [1], Alexander A. Robel [1], Chris C. K. Lai [2], Earle Wilson [3], and Peter Washam [4]

[1]Department of Earth and Atmospheric Sciences, Georgia Institute of Technology, Atlanta, USA
[2]Department of Civil and Environmental Engineering, Georgia Institute of Technology, Atlanta, USA
[3]Department of Earth System Science, Stanford University, Stanford, USA
[4]Department of Astronomy, Cornell University, Ithaca, USA

**Correspondence:** Madeline Mamer (mmamer3@gatech.edu)

**Abstract.** Small-scale ice-ocean interactions near and within grounding zones play an important role in determining the current and future contribution of marine ice sheets to sea level rise. However, the processes mediating these interactions are simplified in large-scale models due to limited observations and computational resources, contributing to uncertainty in future projections. Previous modeling studies have demonstrated that seawater can interact with subglacial discharge upstream of the grounding zone and recent observations appear to support this possibility. In this study, we investigate turbulent mixing of quasi laminar intruded seawater and glacial meltwater under grounded ice using a computational fluid dynamics solver. In agreement with previous work, we demonstrate the strongest control on intrusion distance is the speed of subglacial discharge and the geometry of the subglacial environment. We show that, in the fluid regimes simulated here, and expected at ice shelf grounding zones, turbulent mixing plays a negligible role in setting intrusion distance. Basal melting from seawater intrusion produces buoyant meltwater which may create a negative feedback by chilling and freshening near-ice water, therefore reducing further melting, however, this remains unquantified. The magnitude of modeled basal melt rates from seawater intrusion can be replicated by existing sub-ice shelf melt parameterizations by modifying the traditionally used transfer coefficients. We conclude that, in times or places when subglacial discharge is slow, seawater intrusion can be an important mechanism of ocean-forced basal melting of marine ice sheets when considering added geometric complexities and ocean conditions.

## 1 Introduction

Marine-terminating glaciers in Greenland and Antarctica have experienced accelerating ice loss over the past several decades (Otosaka et al., 2023). Ocean melting of marine-terminating glaciers has driven a considerable amount of this mass loss (Depoorter et al., 2013; Rignot et al., 2013), but is not yet accurately represented within current coupled ice-ocean models. Efforts to improve coupled models focus on improving current parameterizations of melt rates at the ice-ocean interface (Kimura et al., 2015; Middleton et al., 2022; Zhao et al., 2024; Washam et al., 2023), collecting in-situ data (Stanton et al., 2013; Christianson et al., 2016; Jackson et al., 2017; Washam et al., 2020; Davis et al., 2023; Schmidt et al., 2023), and uncovering novel melt mechanisms not included in current coupled models (Rosevear et al., 2021, 2022). In this study, we focus on fluid processes that mediate one such novel melt mechanism under grounded ice.

Grounding lines are junctions between the grounded and floating portions of ice sheets. The grounding line has historically been considered to be a hydraulic barrier between the cold, fresh subglacial hydrologic system and the relatively warm, saline ocean. Models have suggested that tidal forcing may push seawater upstream of the grounding line (Sayag and Worster, 2013; Walker et al., 2013), causing tidally asymmetric melt in this "grounding zone" region (Gadi et al., 2023). Such asymmetry results in stronger melting during the ascent of high tide and weaker melting during the transition to low tide. Recent field observations have suggested that this zone is hydraulically active, with mixing occurring between the ocean and subglacial hydrology upstream of the grounding line (Macgregor et al., 2011; Horgan et al., 2013; Whiteford et al., 2022; Kim et al., 2024). Satellite observations find evidence for elevated rates of basal melt at and beyond the grounding line relative to very low values typically expected for grounded ice, contributing to retreat (Milillo et al., 2019; Ciracì et al., 2023).

More recently, seawater intrusion within and beyond grounding zones has been hypothesized to behave similarly to flow in estuaries, developing a wedge-shaped density front at which fresh glacier melt discharge flows over saline seawater. Wilson et al. (2020) and Robel et al. (2022) adapted a theoretical model for layered shallow water flows in estuaries (e.g., Krvavica et al., 2016) to demonstrate mathematically that freshwater velocity, the geometry of the subglacial environment, and the wall drag acting on the fluid all potentially exert important controls on the extent of seawater intrusion in subglacial hydrological systems. Due to the stratified nature of a salt wedge and lower fluxes, Robel et al. (2022) has hypothesized that ice loss from seawater intrusion is driven by double-diffusive convection. More recent work suggests that shear-driven melting beneath grounded ice can enlarge the subglacial cavity, enhancing seawater intrusion and potentially triggering a runaway positive feedback (Bradley and Hewitt, 2024). This feedback arises when melting exceeds ice advection, preventing upstream ice from replenishing the ablated region and allowing the conduit to grow unchecked. Bradley and Hewitt (2024) identified a regime in which seawater intrusions could become unbounded, effectively "intruding infinitely."

Prior theories of ocean-driven melt have emphasized the importance of turbulent mixing in driving heat and salt transport towards the ice-ocean interface (Mcphee, 2008) and creating fully mixed boundary layers adjacent to the ice. Traditional melt parameterizations rest on this assumption of an ice-adjacent fully mixed boundary layer (Jenkins, 2011). However, in regions of seawater intrusion such assumptions may not hold, as a layer of buoyant freshwater lies above a dense salt wedge in a stratified environment (Wilson et al., 2020), with a boundary layer that evolves along the length of the seawater intrusion. This means the near-ice salinity goes to zero and the equation of state changes, and is therefore a distinctly different regime than the assumptions inherent in current melt parameterizations. Turbulent mixing can be a mechanism that reduces stratification by enhancing interfacial mixing between the salt wedge and freshwater layer, transporting heat and salt upward into the ice-adjacent freshwater layer. Conversely, excess buoyant forcing from basal melting induced by seawater intrusion may inhibit intrusion by increasing total freshwater discharge and strengthening stratification. Prior studies of seawater intrusion have omitted turbulent mixing in the interest of obtaining simple mathematical theories and have not considered feedbacks between intrusion-induced melting and intrusion persistence. In this study, we investigate these effects with the aid of a computational fluid dynamics solver, with three aims: (1) to test previously proposed controls on seawater intrusion distance, (2) to determine the effects of turbulent mixing on seawater intrusion, and (3) to investigate the dynamics of intrusion-induced basal melting.

## 2   Methods

To study the dynamics of seawater intrusion beneath grounded ice, we utilize ANSYS Fluent (ANSYS, 2022), a computational
fluid dynamics (CFD) solver of the Reynolds-Averaged Navier Stokes (RANS) equations using the finite volume method. More
details on the RANS equations and ANSYS Fluent are provided in the appendix section B1. Using a CFD solver allows the
simulations to have model resolutions on the scale of millimeters. Unlike previous studies, by using this CFD solver at such fine
resolution, we are able to resolve heat and mass transfer through the entire water column, and appropriate turbulence closure
schemes allow for the boundary layer to be resolved. Interfacial shear instabilities that might be expected of highly stratified
flows such as the ones simulated in this study are not explicitly resolved in RANS models, however the averaged mixing effect
of such instabilities should be captured. Furthermore, a recent study using direct numerical simulations has identified that such
instabilities do not arise at the lower Reynolds numbers we are simulating here (Zhu et al., 2023). ANSYS Fluent has been
extensively validated across a wide range of flow geometries and conditions. The $\kappa$-$\epsilon$ turbulence closure model, widely used in
computational fluid mechanics and implemented in ANSYS Fluent, has proven effective for simulating flow structures around
complex bathymetry (Zangiabadi et al., 2015; Al-Zubaidy and Hilo, 2022) and sediment-laden plumes (Nguyen et al., 2020).
Its applicability to multiphase modeling in geometries similar to those studied here has also been validated against experimental
results (Sultan et al., 2019). Most relevant to this study, Chalá et al. (2024) demonstrated ANSYS Fluent's capability to simulate
seawater intrusion in porous aquifers, showing strong agreement between experimental data and model predictions for intrusion
length and shape. Additionally, ANSYS Fluent has been successfully applied to freeze desalination processes, using the volume
of fluid method to model saltwater mixtures cooled at the base (Jayakody et al., 2017).

In this study, we test the effects of freshwater velocity ($u_f$), turbulent mixing, and subglacial geometry on seawater intrusion
distance and vertical structure. The freshwater velocity is varied over three orders of magnitude (0.05-5 cm s$^{-1}$) to mimic a
range of likely subglacial discharge velocities in Antarctica and Greenland in non-summer months (Carter et al., 2017; Davis
et al., 2023; Washam et al., 2020) and previous experimentally tested speeds (Wilson et al., 2020). The corresponding Reynolds
numbers for the given geometry and freshwater flux are 25, 250, and 2500 for the low, medium, and fastest freshwater cases
presented. In the experimental set up in this study, we only consider quasi laminar flow in order to facilitate comparison
to previous studies (Wilson et al., 2020; Robel et al., 2022) which find that subcriticality ($Fr < 1$) is required for intrusion
development. To test the role of turbulent mixing, we varied a mixing parameter across three values along with simulating
laminar flow cases. The simulations in which we range over subglacial discharge velocities with medium turbulent mixing (e.g.
$C_\mu = 0.09$) are our reference simulations to which we will compare all other results. The purpose of the reference simulations is
to compare with previous theory that considers quasi-laminar flow and no melting effects. Further testing on potential intrusion
control variables is also considered and explored in detail. More discussion on turbulence modeling is given in section 2.3.

In evaluating the geometric effect on intrusion dynamics, we tested both a retrograde slope with an angle of 0.5° relative to
the horizontal, and a thicker subglacial environment. We also simulated melting induced by seawater intrusion to investigate
how this secondary source of buoyancy affects intrusion persistence and structure. Each simulation is initialized with a warm,
salty ocean basin ($S$ = 30 ppt and $T$ = 0.5 ° C) and a fresh, cold subglacial environment ($S$ = 0 ppt and $T$ = -0.73 ° C) as

shown in Figure B1. The thermal driving associated with the initial conditions is approximately 1.2° and the reduced gravity is 0.23 m s$^{-2}$. The transient solver is then run for 12 hrs at 5 s time steps, with data exported every 20 time steps (100 s). A quasi-steady-state is reached by all simulations, as evaluated by the time change of the average density of the subglacial space being less than 10$^{-4}$ kg m$^{-3}$. All results presented are time-averaged values from when the quasi-steady-state is reached. A summary of the experimental setup, parameters, and key results are given in Tables 1, 2, and 3.

## 2.1 Domain and Boundary Conditions

We consider a two-dimensional subglacial domain, encompassing one vertical and one horizontal (orthogonal to the local grounding line) dimension. The domain is akin to an unbounded freshwater sheet that meets the ocean at a specified discharge point, representing the grounding line. Since we do not resolve ice dynamics, we prescribe the grounding line as a vertical ice face in the domain boundary geometry, instead of including an ice shelf to reduce the domain size needed in these simulations and limit geometric constraints to an idealized subglacial water sheet. The geometry of the bounding surfaces in this configuration does not change in time, with the vertical ice front chosen to limit the geometric influences on intrusion distance to only within the subglacial environment. Tides are not considered in this study, which would temporally alter the geometry of the subglacial environment and therefore be another factor influencing intrusion distance. The underlying bedrock is impermeable and the subglacial environment between the ice and the bedrock is not obstructed by obstacles, both of which would introduce additional controls on the intrusion distance (Robel et al., 2022).

Figure 1 depicts the standard model configuration we use in this study. There are two velocity inlets common to all simulations: a seawater source at the inlet boundary of the tall ocean basin (dark blue arrow in Figure 1) and a freshwater source at the inlet boundary of the subglacial environment (light blue arrow in Figure 1). A pressure outlet boundary (red arrow in Figure 1) is prescribed in the ocean basin employing a zero gradient flux at the boundary and ensuring mass conservation in the model. In describing the results, we utilize the convention of downstream being towards the ocean basin and the upstream being towards the freshwater inlet. The seawater inlet velocity is prescribed as $u_o$ = 0.5 cm s$^{-1}$ across all cases, acting as a sustaining source of saline warm water to the model domain. Simulations with varied $u_o$ (Table B1) indicate that the seawater inlet speed does not have a qualitative influence on the seawater intrusion distance or vertical structure over a range of relatively weak ocean current speeds that we consider appropriate for the constrained ocean cavity near the grounding line (0.05-5 cm s$^{-1}$), so we set $u_o$ = 0.5 cm s$^{-1}$ for all simulations going forward.

Finally, vertical and horizontal ice wall boundaries (where the ice is in contact with the fluid domain) are defined with characteristics that mimic a grounding line environment. The ice wall boundaries have a pressure and salinity-dependent thermal boundary condition represented by the liquidus condition:

$$T_b = S_b\lambda_1 + \lambda_2 + z_b\lambda_3. \tag{1}$$

Here, $\lambda_1$, $\lambda_2$, and $\lambda_3$ are constants, and the boundary salinity is $S_b$. The depth of the ice is equal to $z_b$, in these simulations we set this to be 1000 m. The boundary salinity, $S_b$, is the salinity of the cell filled with water nearest to the ice face and is permitted to evolve during the simulation via the evolution equation B12. For the reference simulations, the boundary salinity

evolves in time and space due to advection and diffusion of the subglacial discharge and seawater intrusion. In the simulations with melting, an additional source of freshwater is injected into the near-boundary grid cells, actively freshening the boundary salinity. Both the vertical and horizontal ice boundaries have a no-slip kinematic condition in the non-melting cases, forcing the freestream fluid velocity to be zero at the ice wall. The vertical ice front in this configuration resembles a tidewater glacier and is utilized to restrict the geometric controls on intrusion distance to those within the subglacial environment. Having a

low-sloping ice-shelf bottom would introduce further constraints on the ability for seawater to intrude beyond the grounding line since intrusion distance is a function of the height of its environment (Robel et al., 2022; Wilson et al., 2020). Such a configuration would also require computational domains much larger than are feasible to simulate at the resolution needed to properly resolve the ice-water boundary layer. Despite the tidewater-like geometric configuration, the low subglacial discharge fluxes make the fluid domain appropriate for simulating conditions expected in Antarctica year-round or Greenland in non-

summer months.

The "subglacial environment" is the domain upstream of the grounding line (x > 0 in Figure 1). Since the bedrock and ice-adjacent boundary layer thicknesses depend on the freestream fluid velocities, the mesh resolution changes for each freshwater velocity to accurately model near-wall processes with the chosen turbulence closure scheme. For the tested freshwater velocities, the vertical domain size hinders the development of a full boundary layer. Instead, everywhere in the domain, the

fluid feels the effects of the wall boundary. Further discussion on domain and meshing is included in sections B4 and B5.

## 2.2 Salt and Heat Transport

In addition to solving the RANS equation for fluid velocities, we configure the CFD solver to calculate the concentration of salt with a "species transport model" (advection-diffusion equations) (ANSYS, 2009). Salt is therefore transported as an active tracer within the fluid domain. To ensure mass transport within the computation domain is realistic and physical, there are

two velocity (hence mass) inlets (i.e the subglacial discharge and ocean inflow) in the non-melting case and three velocity inlets (subglacial discharge, the melting horizontal ice face, and the ocean inflow) in the melting case. Mass is conserved via a pressure outlet (red arrow in Figure 1) which employs a zero-gradient flux boundary condition. At this boundary, the mass outflow rate is not specified and is determined as part of the numerical solution based on the requirement that all flow variables have zero gradients in the direction normal to the boundary. This kind of arrangement is typically used to emulate

fluid flows in an infinite domain as in our case where subglacial channel discharge is released at the grounding line into an ocean with infinite extent. Energy, and therefore fluid temperature, is evolved via an energy conservation equation employed by the CFD solver resolving advection, conduction, and salt diffusion (ANSYS, 2009). Conduction represents heat transfer due to horizontal thermal gradients. Since salt is tracked as an active tracer, it transports enthalpy associated with its specific heat and concentration gradients. This must be accounted for in the energy equation Fluent solves. Further details of heat and salt

transport in the model can be found in appendix section B3 alongside discussion of the RANS formulation in appendix section B1.

The seawater inlet is prescribed with 30 ppt salinity and $T_o = 0.5°$ C while the freshwater inlet is prescribed with zero salinity and the corresponding pressure-dependent freezing point from equation 1. Seawater temperature is chosen to represent warm

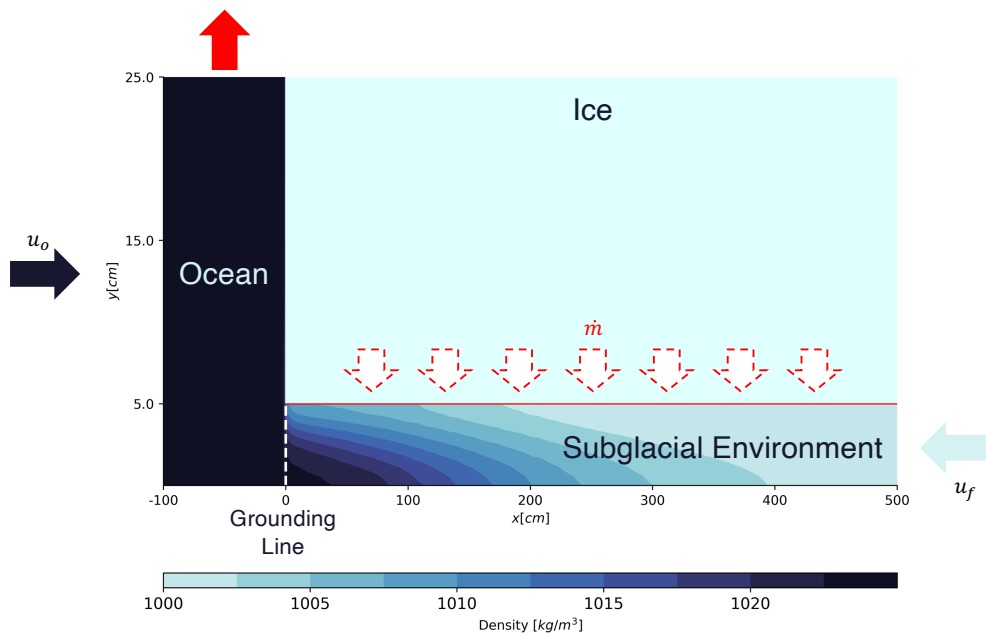

**Figure 1.** Schematic of the domain used in ANSYS Fluent. Ice block is not simulated and used to graphically depict the vertical and horizontal (red solid line) ice-ocean interfaces. The length and height of the subglacial environment vary between simulations. The ocean basin is 2 m wide by 5 m tall for all simulations. The dark blue arrow on the left represents seawater input ($u_o$), while the light blue arrow on the right represents freshwater input ($u_f$). The solid red arrow represents the zero-gradient flux boundary outlet. The red dashed arrows represent the meltwater input ($\dot{m}$) that is turned on for the simulations with melting as a function of near-wall temperature (eq. 4).

cavity Antarctic conditions (Middleton et al., 2022; Kimura et al., 2015). Seawater salinity is set to generate a water mass with density characteristic of the Southern Ocean. The species transport model also calculates water density ($\rho_w$) with a prescribed linear equation of state from Roquet et al. (2015):

$$\rho_w = 1000 + S_w * (0.7718 * 1000) + (T_w - 273.15) * (-0.1775). \tag{2}$$

The salinity, $S_w$, and temperature, $T_w$, of the fluid are found at the center of each cell. This simplified linear equation of state was chosen for ease of implementation as a user-defined function within ANSYS Fluent and with the assumption that

cabelling or thermobaricity do not play a role in this environment. A higher-order equation of state was tested to compare the linear formulation used here. While the fluid density is different, the salinity and temperature of the fluid remain unchanged since they have their respective transport equations. Further discussion on choice and justification of the linear equation of state is presented in appendix section B8.

## 2.3 Turbulence Closure Scheme

Turbulent mixing can increase the exchange of heat and salt between seawater and fresh subglacial discharge, homogenizing the water column and potentially reducing the ability for seawater to intrude. The theory of Wilson et al. (2020) assumes no mixing between the two water masses, treating the interfacial drag (which dictates shear) as a free parameter dependent on fluid flow. Robel et al. (2022) treats interfacial drag as negligible to derive a closed-form prediction for intrusion distance. However, under scenarios of fast subglacial discharge, the experiments described in Wilson et al. (2020) exhibited interfacial mixing with the

formation of wave crests at the top of the salt wedge. To understand the role turbulent mixing has in a seawater intrusion regime, we enable the commonly used $\kappa$-$\epsilon$ two-equation turbulence closure scheme (Mansour et al., 1989; Launder and Spalding, 1983). A closure scheme is necessary because averaging the Navier-Stokes equations introduces Reynolds stresses due to turbulent motion within the fluid. Reynolds stresses take the form $(\overline{u_i' u_j'})$, the averaged product of turbulent velocity fluctuations. The closure scheme used here employs the turbulent-viscosity hypothesis, which relates the deviatoric Reynolds stresses to the

mean strain rate via a positive scalar eddy viscosity (Pope, 2000). The $\kappa$-$\epsilon$ closure scheme solves for the eddy viscosity with

$$\mu_t = \rho_w C_\mu \frac{\kappa^2}{\epsilon}, \tag{3}$$

where $\kappa$ represents the turbulent kinetic energy, $\epsilon$ represents dissipation, and $\rho_w$ is the fluid density. This form of turbulence closure provides an avenue to manipulate the degree of turbulent mixing by modulating the parameter $C_\mu$. This is a model parameter that dictates the amount of turbulent transport (mixing) given some $\kappa^2/\epsilon$. Larger values of $C_\mu$ imply that for a

given level of turbulent kinetic energy and dissipation, turbulent stresses will be increased due to increased eddy viscosity. We vary this parameter across a factor of four (0.045, 0.09, 0.18) which allows us to explore a range of possible turbulent mixing while maintaining model stability. The middle value, 0.09, is commonly adopted as the "standard value" and is derived from experiments with equilibrium shear flows e.g., the log-law region and above in pipe flows. We employ the Yang-Shih low-Reynolds formulation of the $\kappa$-$\epsilon$ closure scheme (Yang and Shih, 1993) which uses damping equations near wall bound-

aries to adequately resolve the viscous sublayer, allowing for finer resolution near the ice boundaries and complete boundary layer resolution (Hrenya et al., 1995; ANSYS, 2009). Further discussion on the ANSYS Fluent model and formulation of the turbulence closure model can be found in appendix section B2.

## 2.4 Basal Melting

In some simulations, we also simulate the added buoyancy flux resulting from intrusion-forced melting. Using the last time

step from the corresponding non-melting simulation as the initial condition, we enable melting at the horizontal ice wall by

**Table 1.** Simulations with varied freshwater discharge velocity and degree of turbulent mixing for flat geometry with H = 5 cm.

| $u_f$ (cm s$^{-1}$) | $C_\mu$ | L$_{sge}$ (m) | L$_i$ (m) |
|---|---|---|---|
| 5 | 0.18 | 20 | 0.52 |
| 5 | 0.09 | 20 | 0.51 |
| 5 | 0.045 | 20 | 0.52 |
| 5 | laminar | 20 | 0.49 |
| 0.5 | 0.18 | 20 | 4.72 |
| 0.5 | 0.09 | 20 | 4.79 |
| 0.5 | 0.045 | 20 | 4.80 |
| 0.5 | laminar | 20 | 4.76 |
| 0.05 | 0.18 | 40 | 25.31 |
| 0.05 | 0.09 | 40 | 23.09 |
| 0.05 | 0.045 | 40 | 24.19 |
| 0.05 | laminar | 40 | 20.63 |

changing it from a wall boundary condition to a velocity inlet (red dashed arrows in Figure 1). In ANSYS Fluent, a wall boundary condition holds a constant temperature while providing a shearing force on the fluid in accordance with the no-slip wall boundary condition. When changed to a velocity inlet, the inflowing melt can have a prescribed temperature and salinity but does not provide any source of shear. Therefore, the melting cases have a free shear kinematic boundary condition. To address this model limitation, we also conducted experiments without melting and with a free-slip kinematic condition, for comparison with the intrusions in the melting experiments. The temperature of the inflowing meltwater is set by equation 1. Melting is only turned on for the horizontal ice face (red boundary in Figure 1) and not the vertical ice face to isolate the added buoyancy effects from seawater intrusion forced melt only. The vertical ice face is susceptible to plume-driven melt from the buoyant subglacial discharge and is distinctly different than the seawater intrusion melt domain. The downward fluid velocity prescribed at the horizontal ice face is set by the melt rate, $\dot{m}$, and is a function of the difference between the near-wall cell's centroid temperature $T_w$, and ice-ocean interfacial temperature $T_b$, thermal conductivity $\kappa_T$, and density of the ice $\rho_i$

$$\dot{m} = \frac{\kappa_T}{\rho_i L_i} \frac{T_w - T_b}{0.5 H_c}. \tag{4}$$

The thermal forcing is divided by half of the near-wall cell height, $H_c$, to obtain the near-wall thermal gradient, $\partial T/\partial y$. $L_i$ is the latent heat of ice. The values of all constants are presented in Tables A1 and A2. This framework represents the conservation of heat at the ice-ocean interface, which varies along the horizontal ice face as the near-wall thermal gradient changes due to seawater intrusion and vertical mixing. We can use equation 4 to calculate melt rates instead of a parameterization because we resolve the boundary layer directly and do not need to make assumptions about how heat and salt are transported through the boundary layer. Inherent in this is the assumption ANSYS Fluent accurately simulates all appropriate boundary layer transport

processes necessary to get heat from the freestream fluid flow to the cell grid next to the ice face, and any melting experienced is due to the conservation of heat at the ice-adjacent cell. In setting a vertical velocity to mimic a moving interface, we introduce additional sources of momentum to the fluid. The vertical velocity arising from the meltwater inlet is small relative to the main flow and is therefore negligible, but the buoyancy that the meltwater brings into the fluid domain is an integral part of the heat-driven melting process. The input of fresh, cold water due to melting introduces a buoyancy flux to the domain, due to density differences between the meltwater and intrusion. An alternative to this formulation would be to replace $\kappa_T$ with the product of thermal diffusivity ($K_T$), seawater density ($\rho_w$), and seawater heat capacity ($c_w$), allowing for varying seawater density to affect heat transfer to the boundary. However, back-of-envelope calculations show small variances in density ($\sim 20$ kg/m$^3$) lead to small changes ($< 5\%$) in the melt rate. Therefore, using a constant thermal conductivity is appropriate. Note the boundary does not move over time (as in Bradley and Hewitt, 2024) and an evolving geometry of the subglacial space is not tested in this work. This choice greatly simplifies the computational domain and considerations of meshing with turbulent closures.

**Table 2.** Simulations with melting and without melting, free slip kinematic boundary condition.

| $u_f$ (cm s$^{-1}$) | Boundary Condition | $C_\mu$ | H (cm) | $\Theta$ ° | $L_{sge}(m)$ | $L_i$ (m) |
|---|---|---|---|---|---|---|
| 5 | melt-enabled | 0.09 | 5 | 0 | 20 | 0.27 |
| 5 | free slip | 0.09 | 5 | 0 | 20 | 0.67 |
| 0.5 | melt-enabled | 0.09 | 5 | 0 | 20 | 5.36 |
| 0.5 | free slip | 0.09 | 5 | 0 | 20 | 11.17 |
| 0.05 | melt-enabled | 0.09 | 5 | 0 | 40 | 23.17 |
| 0.05 | free slip | 0.09 | 5 | 0 | 40 | 40.0 |
| 0.5 | melt-enabled | 0.09 | 5 | 0.5 | 20 | 6.75 |
| 0.5 | free slip | 0.09 | 5 | 0.5 | 20 | 20.0 |
| 0.5 | melt-enabled | 0.09 | 7.5 | 0 | 40 | 13.25 |
| 0.5 | free slip | 0.09 | 7.5 | 0 | 40 | 13.5 |

## 3 Results

### 3.1 Characteristics of the Seawater Intrusion

In all simulations, warm seawater intrudes some distance beyond the defined grounding line (x = 0 m in Figure 2). A strong control on seawater intrusion distance is the freshwater discharge velocity, in line with previous work (Wilson et al., 2020; Robel et al., 2022; Krvavica et al., 2016). The simulation with the lowest flux of freshwater (Figure 2C) has approximately a 25 m intrusion, while the fastest flux experiences only about 0.5 m of intrusion (Figure 2B). This range of intrusion distances demonstrates a weaker dependence on freshwater velocity than suggested by Robel et al. (2022) where intrusion distance has

an inverse quadratic dependence on freshwater velocity and therefore should vary by a factor of 1000 in response to the range of input velocities tested here. Turbulent mixing, as modulated by $C_\mu$, affects intrusion distance to a lesser degree than freshwater

discharge velocity when varied over a wide range encompassing likely values on the lower-end for realistic estuarine-like mixing rates (Geyer et al., 2000, 2008). For the middle freshwater velocity (Figure 2A), increased turbulent mixing slightly reduces the intrusion distance. To contrast the effects of turbulent mixing, we tested laminar flow cases with no turbulent mixing (green transects Figure 2) and saw no meaningful difference in intrusion distance for $u_f = 0.5$ cm s$^{-1}$. For the low and high freshwater velocity cases, including turbulent mixing (blue transects in Figure 2B,C) increases intrusion distance when

compared to the laminar test case (green transects in Figure 2B,C). For the slowest freshwater case, the intrusion is reduced in the absence of turbulent mixing due to the lack of entrainment between the seawater intrusion and the buoyant subglacial discharge. When modeled, this entrainment extends the intrusion by generating a tail of relatively low-density water. This is not to say that turbulent mixing is unimportant in the dynamics of seawater intrusion, but rather that intrusion distance is not strongly sensitive to the strength of turbulent mixing (over the range of discharge velocities and $C_\mu$ values considered

realistic for subglacial and estuarine environments), particularly when compared to other factors such as the geometry of the subglacial environment and freshwater discharge flux. In the flow regimes simulated here, the turbulent viscosity does not get large enough to greatly affect the flow dynamics as shown in Figure D1. It may be that for much higher discharge velocities ($\mathcal{O}$(m s$^{-1}$)) encountered at times of high subglacial discharge, turbulent mixing plays a more important role than the cases considered here. However, under those conditions, the freshwater flux is likely to be supercritical and prohibit any intrusion

development as predicted by theory in Wilson et al. (2020) and Robel et al. (2022).

**Table 3.** Simulations with varied geometry

| $u_f$ (cm s$^{-1}$) | $C_\mu$ | H (cm) | $\Theta$ $^\circ$ | $L_{sge}$ (m) | $L_i$ (m) |
|---|---|---|---|---|---|
| 0.5 | 0.09 | 5 | 0.5 | 20 | 7.87 |
| 0.5 | laminar | 5 | 0.5 | 20 | 7.63 |
| 0.5 | 0.09 | 7.5 | 0 | 40 | 12.95 |
| 0.5 | laminar | 7.5 | 0 | 40 | 13.5 |

Changing subglacial geometry has a large effect on intrusion distance as demonstrated by the experiment with a taller subglacial channel (H = 7.5 cm) plotted as an orange transect in Figure 2A. Increasing the channel height by 50% increased the intrusion distance by nearly a factor of 3. This indicates an even stronger sensitivity to the height of the subglacial opening in these more realistic simulations than predicted by Robel et al. (2022), which finds a quadratic dependence on the subglacial

conduit height. We also tested the effects of a retrograde bed slope ($\theta = 0.5°$) on intrusion distance, which increased the intrusion distance by about a factor of 1.5 relative to the flat cases tested (magenta transect Figure 2A). We did not find any evidence for an unbounded increase in the intrusion distance under these retrograde slopes, as predicted by Wilson et al. (2020) and Robel et al. (2022). That being said, our finite domain length limits the intrusion distances achievable in this model configuration.

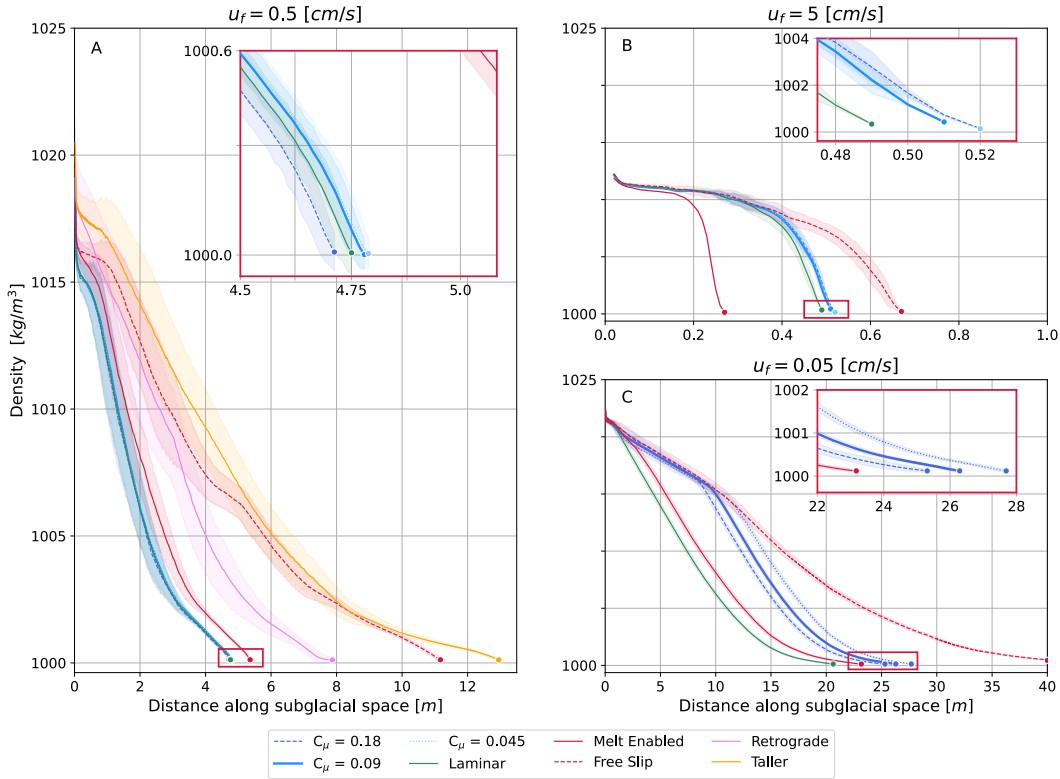

**Figure 2.** Depth-averaged density transects along the the subglacial environment. The transect termini represent the intrusion distance or the point at which density is less than or equal to the density of freshwater at the pressure-dependent freezing point. Blue lines represent simulations without melting. Dashed lines represent higher turbulent mixing ($C_\mu = 0.18$), dotted lines are low turbulent mixing ($C_\mu = 0.045$), and solid lines are medium turbulent mixing ($C_\mu = 0.09$). Dashed red transects are for simulations without melting and have free-slip kinematic boundary conditions on the horizontal ice face. The solid red transects are simulations with melting and medium turbulent mixing. The orange and magenta transects in panel A are scenarios with different geometries, a taller subglacial environment (H = 7.5 cm), and a retrograde slope ($\theta = 0.5°$) respectively. The green transects represent laminar flow cases with no turbulent mixing. Shading depicts the first temporal standard deviation or the maximum/minimum value depending on the smallest absolute difference with the average value. Note the varying x-axis across the panels.

The time-averaged vertical profiles of temperature, salinity, and velocity along the intrusion for simulations without basal melting (Figure 3) depict a strongly stratified two-layered exchange flow, with variable thermohaline gradients for each freshwater flux tested. The fastest freshwater flux (Figure 3A-D) has the strongest stratification and fastest exchange flow of all the cases tested. For all cases, the height and slope of the thermocline and halocline decrease upstream into the subglacial space. The strength in stratification and exchange flow decreases with decreased freshwater flux. This trend holds for the retrograde

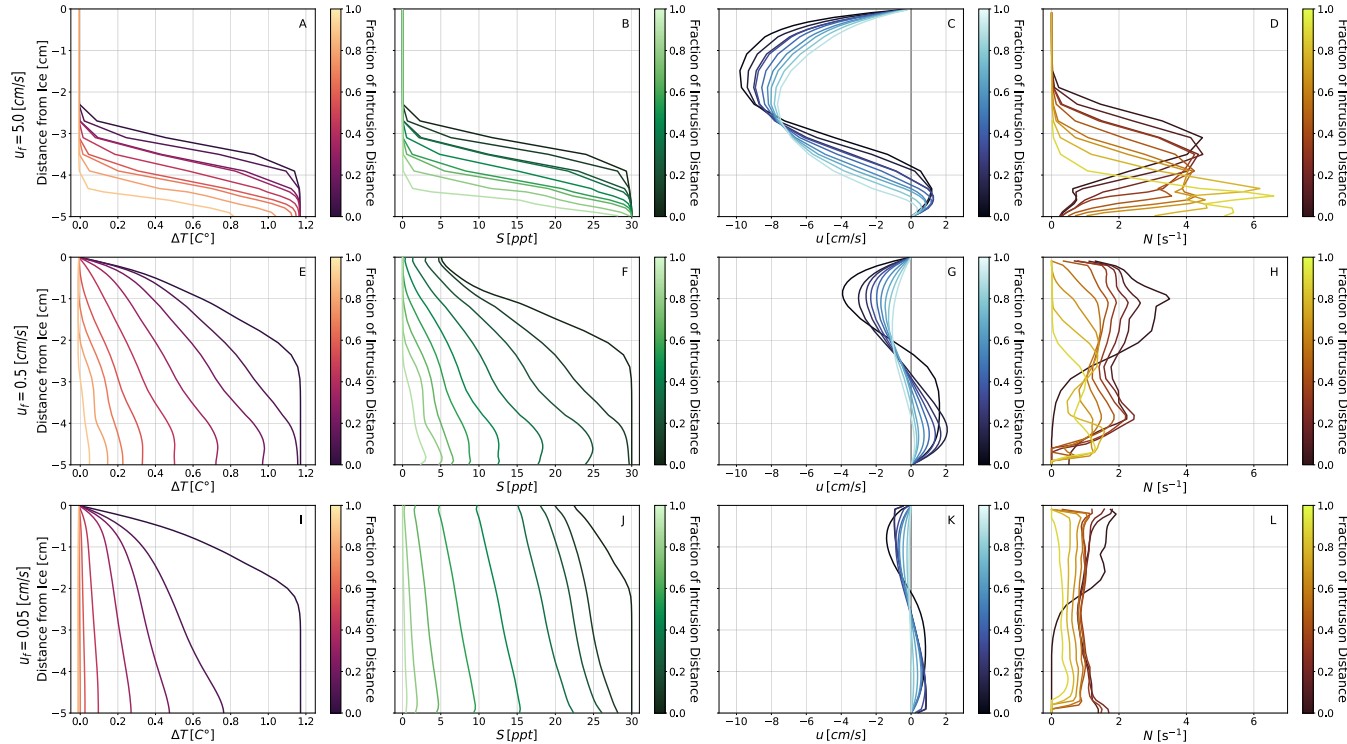

**Figure 3.** Time-averaged vertical profiles of thermal forcing (A, E, I), salinity (B, F, J), x-component of velocity (C, G, K), and buoyancy frequency (D, H, L) along the seawater intrusion for each freshwater velocity tested in the standard geometry with medium turbulence $C_\mu = 0.09$. The fraction of intrusion distance represents evenly spaced vertical slices at relative distances along each intrusion. The top row (A, B, C, D) is for the fastest freshwater velocity, $u_f = 5$ cm s$^{-1}$. The middle row (E, F, G, H) is for the middle freshwater velocity, $u_f = 0.5$ cm s$^{-1}$. Lastly, the third row (I, J, K, L) is for the slowest freshwater velocity, $u_f = 0.05$ cm s$^{-1}$.

slope case and increased channel height case (Figure E1). The retrograde slope has a nearly identical vertical structure to the
similar flat channel test case, while the simulation with increased channel height sees a faster exchange flow due to an increased flux of freshwater. The system decays to a classic pipe flow further upstream into the subglacial space, out of the regime of intrusion. The degree of vertical mixing increases with decreasing freshwater velocity, which is similar to observed mixing dynamics in estuaries (Montagna et al., 2013). For high freshwater fluxes, a well-developed salt-wedge forms and high stratification persists (Figure 3A-D). For the lowest freshwater flux, the diffusive and advective timescales are of the same order,
which leads to homogeneity in the vertical distribution of heat and salt (Figure 3I-L). In estuaries, strong tides generate shear at the bed, increasing turbulence and vertical mixing (Montagna et al., 2013). Tides may play a similar role in subglacial seawater intrusion regimes, however, that is not simulated here.

Buoyancy frequency, $N$ is a measure of the degree of stratification in the water column:

$$N = \sqrt{\frac{-g}{\rho_0}\frac{d\rho}{dy}}. \qquad (5)$$

Higher $N$ indicates stronger density gradients and stratification, tracking the presence of seawater intrusion. Buoyant forcing, as represented by $N$, suppresses mixing in the presence of a flat horizontal ice boundary (as we have here), competing with the velocity shear and turbulence that drive mixing over the length of the intrusion. In regimes of seawater intrusion, buoyancy frequency is large due to the large vertical density gradients and small vertical length scale (Figure 3D,H,L). Buoyancy frequency is largest at the grounding line, and decays to zero upstream of the intrusion. This strong stratification can act to "shield" the ice from melting, generating a near-ice layer of fresh, cold water. However, the horizontal density gradient introduced by the characteristic wedge shape of seawater intrusion will drive vertical baroclinic convective motion to flatten isopycnals. Such baroclinic adjustment may be an important source of interfacial mixing, working in tandem with turbulence and double-diffusive convection to reduce stratification within the subglacial environment. This convective-driven mixing mechanism differs from convective mixing caused by a sloping ice boundary, in which a buoyant plume may form. For the retrograde slope geometry tested here, there is a slight reduction in stratification strength relative to the comparable flat geometry simulation (Figure E1D) likely attributed to natural convective mixing. Where subglacial openings have complex geometry, we anticipate buoyant-driven convection from sloping ice boundaries to aid in driving mixing on small scales which will reduce the stratification from the intrusion.

Drag from the ice shears fluid flow in the subglacial environment to have zero velocity at the ice, per the no-slip kinematic boundary condition. This shearing determines boundary layer thickness and therefore influences heat and salt transport towards the ice. Within the model framework here, the wall drag coefficient is not a free parameter to be set, but rather diagnosed from the simulations via the relationship (Pope, 2000)

$$C_d = \frac{2\nu}{u^2}\left(\frac{\partial u}{\partial y}\right), \qquad (6)$$

where $\partial u/\partial y$ is the velocity gradient normal to the wall and is evaluated with a linear fit above the peak in velocity of the upper freshwater layer. The free stream current speed, $u$, is obtained from the local centerline flow. The kinematic viscosity is represented by $\nu$. The full derivation of calculating the drag coefficient is given in appendix section C. In previous work (Wilson et al., 2020; Robel et al., 2022), $C_d$ is set to values around $\sim 10^{-3}$ derived from observed drag coefficients under sea ice, however here we diagnose $C_d$'s in the range of $10^{-2}$ to $10^1$ (Figure 4) when a no-slip kinematic boundary condition is applied. The high drag coefficients simulated here upstream of the intrusion regime are in line with the expected values for laminar flows with these Reynolds numbers. However, over the regime of intrusion, the drag coefficient increases by nearly an order of magnitude in all cases tested. The increased drag coefficients over the intrusion are more difficult to estimate and likely arise from enhanced turbulence in the interfacial shear layer between the intrusion and buoyant subglacial outflow. Where intrusion occurs, the near-wall velocity gradient grows (darker profiles in Figure 3C, G, K), increasing the shear velocity and

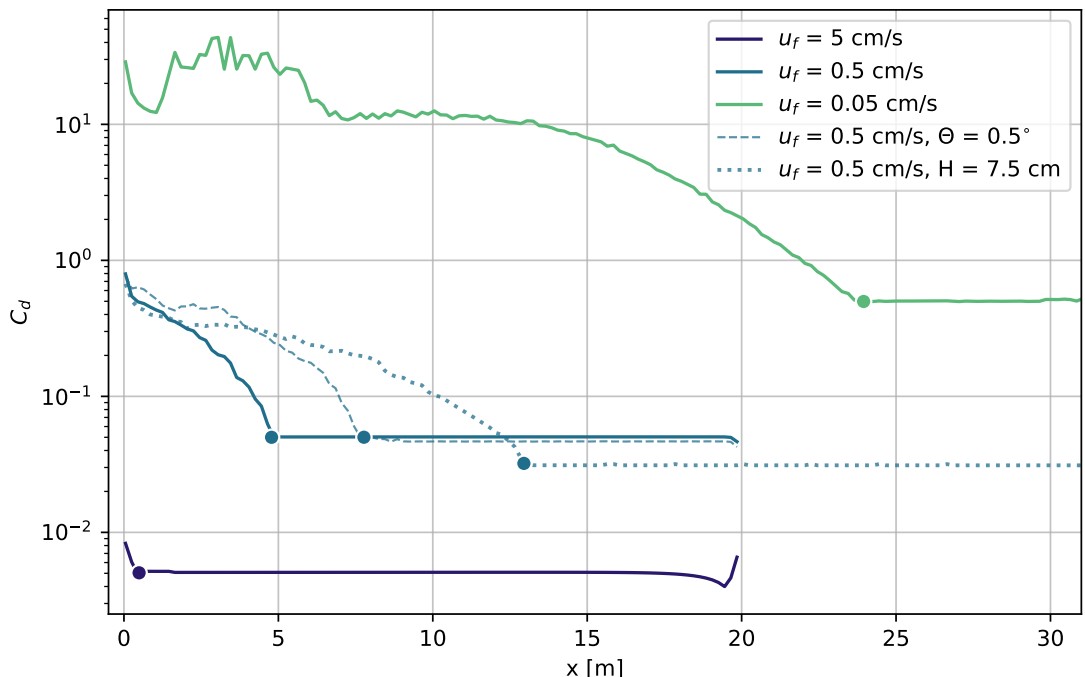

**Figure 4.** Drag coefficient for each freshwater velocity tested with medium turbulence ($C_\mu = 0.09$). The dashed line represents the middle freshwater velocity with a retrograde slope, and the dotted line is for the same $u_f$ but with a taller subglacial environment. The scattered points represent the intrusion distance for each case.

drag. This velocity gradient is an important driver of heat flux to the ice, as discussed in later sections. We also simulated a free-
slip ice boundary condition, meaning no drag at the ice boundary (i.e., $C_d = 0$). This resulted in the largest change in intrusion distance (red dashed transects in Figure 2), and in some cases allowed for seawater to fill the whole simulation domain. This indicates incredible sensitivity of intrusion distance to the applied kinematic boundary conditions. The high drag coefficients simulated here likely explain the much smaller intrusion distances simulated here compared to prior studies (Wilson et al., 2020; Robel et al., 2022), and are discussed in more detail in section 4.

**3.2  Dynamics of intrusion-induced melt**

Intrusion distance does not vary significantly in simulations where basal melt of ice is included (red lines in Figures 2), when compared to simulations with no-slip kinematic boundary cases (blue transects in Figure 2). However, in simulating an interface that allows for an added fresh water source from "melting", we lose the shearing effect of the boundary. Therefore, we find it appropriate to compare the intrusion distances of the simulations with melting to those without melting that have a free slip

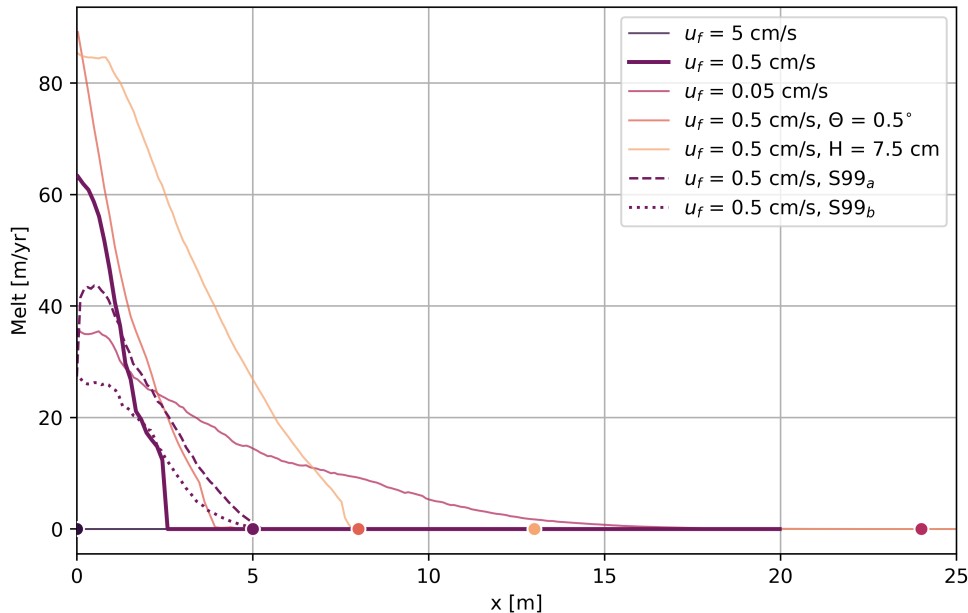

**Figure 5.** Simulated melt rates for all freshwater velocities with medium turbulent mixing alongside the retrograde slope simulation and taller subglacial geometry simulation. The dots indicate the respective intrusion distance. Two versions of parameterized melt rates are plotted for the middle freshwater velocity. The dashed line is the melt rate using a constant "best-fit" turbulent transfer coefficient in the S99 parameterization (S99$_a$). Alternatively, the dotted line uses a variable turbulent transfer coefficient calculated in equation 8 (S99$_b$).

boundary imposed (dashed red transects in Figure 2). Under this comparison, nearly all intrusion distances simulated with melting are reduced by over 50% relative to their corresponding without melting free slip simulations (Table 2).

     The highest simulated melt rate occurs in the sloped subglacial environment with $u_f$ = 0.5 cm s$^{-1}$, peaking at approximately 90 m yr$^{-1}$. The taller subglacial environment under the same $u_f$ follows closely, exhibiting a slightly lower peak melt rate but sustaining elevated melting over an extended region. By contrast, the standard flat geometry with $u_f$ = 0.5 cm s$^{-1}$ experiences

a peak melt rate of about 60 m yr$^{-1}$. In scenarios with retrograde slopes and thicker subglacial environments, the area of ice experiencing significant melt is considerably larger than in the standard flat configuration. These results indicate that the sloped and taller geometries not only expand the region affected by seawater intrusion-induced melting but also increase the peak melt rate by nearly 50%.

     In simulations with the fastest freshwater discharge ($u_f$ = 5.0 cm s$^{-1}$) and standard flat geometry, melting is negligible due

to minimal intrusion and strong stratification. Conversely, for the slowest discharge ($u_f$ = 0.05 cm s$^{-1}$), the melt rate is not the highest but extends over the largest region, suggesting a trade-off between peak melt intensity and spatial extent.

The melt distribution aligns with the intrusion wedge, with maximum melting concentrated near the grounding line and tapering to negligible rates of millimeters per year at approximately half the intrusion distance. This secondary source of freshwater discharge from melting at the horizontal ice-water interface does not substantially alter the vertical structure of the flow regime. The stratification strength and exchange flow within the intrusion are primarily governed by the magnitude of freshwater discharge, which outweighs the momentum contribution from meltwater input.

## 3.3 Parameterization of basal melting in regions of seawater intrusion

Large-scale ocean and ice sheet models typically have grid resolutions much coarser than what is necessary to resolve the ice-ocean boundary layer and directly calculate heat and salt fluxes from the ocean towards the ice. In practice, parameterizations are used in large-scale models to approximate the melt rate based on ocean temperatures and salinities modeled outside the boundary layer ($10^1$-$10^4$ m) from the ice. However, such parameterizations have not previously been tested in the flow regimes relevant to seawater intrusion below grounded ice. Here, we simulate intrusion-induced basal melt by employing the conservation of energy (eq. 4) at the horizontal ice boundary and compare these model results directly to a traditionally used parameterization scheme. In our simulations, we can accurately capture relevant boundary layer heat and salt transport processes and calculate basal melt rates because our model resolution of millimeters directly resolves the near-ice boundary layer.

The most widely used parameterization for modeling ocean-induced ice melt under ice shelves is the three-equation parameterization of Holland and Jenkins (1999) (referred to as S99), which assumes heat and salt transport occurs via shear-driven mixing. Here, we utilize model output to obtain the boundary temperature, $T_b$, which is constrained by the boundary salinity as described in the methods section. The water temperature, $T_w$, is set to the temperature at the center of the subglacial space. The melt rate, as parameterized by S99, is therefore given by:

$$\dot{m}_T L_i \rho_i = \rho_w c_w \gamma_T (T_w - T_b) - \rho_i c_i \kappa_i \frac{dT_i}{dy}. \tag{7}$$

Here, $L_i$ is the latent heat of ice, $\rho_w$ is the seawater density, and $c_w$ is the heat capacity of seawater. The product of the vertical temperature gradient of ice $dT_i/dy$, ice density $\rho_i$, the heat capacity of ice $c_i$, and the thermal conductivity of ice $\kappa_i$ represents heat conduction into the ice. All values are presented in Table A2. We set the ice vertical temperature gradient to zero since we do not model any heat conduction within the ice and instead assume all energy is used to cause melting. The turbulent transfer velocity, $\gamma_T$, describes the transfer of heat across the outer portion of the boundary layer and into the viscous sublayer adjacent to the ice. This transfer velocity is further parameterized via

$$\gamma_T = \frac{u_*}{2.12 \ln(u_* h \nu^{-1}) + 12.5 Pr^{2/3} - 9}, \tag{8}$$

where $h$ is the distance from the ice of the chosen reference for $T_w$, usually taken to be the thickness of the boundary layer (Holland and Jenkins, 1999). The Prandtl number ($Pr$) is the ratio of viscous forces to diffusive forces for temperature (Kader

and Yaglom, 1972; McPhee et al., 1987). The kinematic viscosity is represented by $\nu$ and the shear velocity, $u_*$, is defined as

$$u_* = \sqrt{\nu \frac{\partial u}{\partial y}} \qquad (9)$$

(Schlichting and Gersten, 2016). However, the shear velocity is generally solved as a quadratic function of the wall drag coefficient using a form of equation 6. This means shear-driven parameterized melt rates are sensitive to the choice of drag

coefficient ($C_d$). In this study, $C_d$ is found to be $10^{-2}$ to $10^{1}$ with variability of over magnitude in the regime of seawater intrusion, which is larger than the observed range for sea ice of $10^{-3}$ to $10^{-2}$ (Mcphee, 1980; Randelhoff et al., 2014; Brenner et al., 2021). In using shear velocity to drive the parameterization, this framework assumes the system has enough momentum to dictate boundary layer transport processes. When momentum is low, other processes like diffusion or natural convection may contribute to setting the complex boundary layer structure and vertical thermohaline gradients. For the weak flow scenarios

tested here, we anticipate shear-driven turbulence to be relatively unimportant (Rosevear et al., 2022).

In this study, we assessed the S99 parameterization for a simulation with melting with $u_f = 0.5$ cm s$^{-1}$ using two approaches. In the first approach, we evaluated the turbulent transfer coefficient ($\gamma_T$) using the relationship defined in equation 8 (dotted line in Figure 5). With these values of $\gamma_T$, the S99 parameterization yielded peak melt rates approximately half those simulated directly in our study, underestimating melt in the downstream region of the intrusion. The $\gamma_T$ values derived from this approach

ranged from $10^{-5}$ m s$^{-1}$ within the intrusion to a background value of $10^{-7}$ m s$^{-1}$ upstream of the intrusion. In the second approach, we computed the value of $\gamma_T$ that minimized the root mean squared error (RMSE) between parameterized and simulated melt rates. The optimal $\gamma_T$, corresponding to the best agreement with simulated melt rates, was approximately 3 $\times$ $10^{-5}$ m s$^{-1}$ (dashed line in Figure 5). This optimized turbulent transfer coefficient underestimated peak melt rates in the downstream region of the intrusion while overestimating melt rates upstream near the intrusion nose. Both the tuned and

calculated versions of $\gamma_T$ produced melting along the entire length of the intrusion. In contrast, the simulated melt rates were concentrated along the first half of the intrusion, with a marked drop-off in melting further downstream before the intrusion terminated. Neither parameterization approach was able to replicate this downstream reduction in melt rates in the simulations.

Values of $\gamma_T$ from our simulations generally fall on the lower end of the range reported in the literature, reflecting the low velocities and thin geometries tested (i.e., small Reynolds numbers). However, the corresponding thermal Stanton numbers

($\sqrt{C_d}\Gamma_T$) are in closer agreement within the range found in the literature (Holland and Jenkins, 1999; Jenkins et al., 2010; Washam et al., 2023). Thermal Stanton numbers represent the ratio of heat transfer to the thermal capacity of a fluid and indicate the balance of these two processes (Jenkins et al., 2010), whereas transfer velocities represent the efficacy of heat transport via turbulent and molecular processes throughout the boundary layer. Stanton numbers are often used as tuning coefficients when the boundary layer can't be resolved. The thermal Stanton number is equivalent to $\gamma_T/u$, and back of the

envelope calculations for the middle freshwater velocity where $\gamma_T$ is $\sim 10^{-5}$ m s$^{-1}$ and upper layer velocity within the intrusion is $\sim 4$ cm s$^{-1}$ gives $\sqrt{C_d}\Gamma_T \sim 2.5$ x $10^{-4}$.

Three distinct issues make it difficult to apply existing parameterizations in regions of seawater intrusion: (1) stratification, (2) the interaction of two boundary layers, and (3) change in water flow direction near the ice boundary. Most melt parameter-izations assume a well-mixed and fully developed boundary layer, with reference temperature and salinity taken beyond or at

the boundary layer's edge. However, in the simulations presented here, some degree of stratification exists due to insufficient boundary shear mixing between the upper layer of subglacial discharge and the lower layer of seawater intrusion. Intrusion beyond a grounding line entails fluid flow between two boundaries, which is intrinsically different than the geometries considered for other ocean-induced melting regimes (flow bounded by a singular wall on one side). The upper fresh glacial water will have a boundary layer with the ice, and the lower saline ocean water will generate a boundary layer with the bed. In the subglacial environment, the opposing fresh and saline flows meet and create an interfacial boundary with zero velocity. The sharp transition in opposing flows and strong interfacial shear is associated with two accelerating fluid layers that are increasing drag on their respective boundaries. Here, where subglacial environments have thicknesses of 5 or 7.5 cm, and freshwater velocities ranging from 5 cm s$^{-1}$ to 0.05 cm s$^{-1}$, a complete and stable boundary layer never develops indicating the entire subglacial domain feels the effect of at least one boundary. This means multiple transport processes (turbulent and viscous) operate at the same relative importance, rather than one mechanism dominating the other. These characteristics of seawater intrusion within narrow gaps violate the assumptions inherent in traditionally used parameterizations which rely on well-mixed and fully developed boundary layers (e.g. Holland and Jenkins, 1999).

## 4   Discussion

In the context of previous work, these simulations confirm that freshwater flux and the geometry of the subglacial environment are both strong controls on seawater intrusion. Our simulated intrusions follow the general trend and scale sensitivity to those identified in previous laboratory experiments (Figure 6) (grey markers; Wilson et al., 2020) which are within a factor of 10 to the theoretical prediction (dashed line) from Robel et al. (2022). Previous studies estimate possible seawater intrusion distances of kilometers to 10s of kilometers (Wilson et al., 2020; Robel et al., 2022), which give many orders of magnitude difference than any of the intrusions simulated in this study. This difference is likely due to the drag coefficient at the ice wall ($C_d$), which field studies of ocean flow under sea ice (Mcphee, 1980; Randelhoff et al., 2014; Brenner et al., 2021) and geometric parameterizations (Lu et al., 2011) estimate to be of order 10$^{-3}$ to 10$^{-2}$. In this study, the drag coefficient of the wall cannot be prescribed but rather is an emergent property arising from the no-slip kinematic boundary condition and momentum dissipation within the model from the $\kappa$-$\epsilon$ closure scheme. Calculating the drag coefficient using model output gives $C_d$ with values of order 10$^{-2}$ to 10$^1$ with variability of over a magnitude within the intrusion regime. The analytical theory of intrusion distance ($L$) for an unobstructed water sheet from Robel et al. (2022) is,

$$L = \frac{H^2 g'}{4 C_d^2 u^2},\tag{10}$$

where H = 0.05 m is the height of the subglacial environment and $g'$ = 0.20 m s$^2$ is the reduced gravity. The drag coefficient, $C_d$, and the fluid velocity, $u$, are both set to the average values within the intrusion. The reduced gravity is referenced to the density difference between the prescribed pure freshwater and pure seawater. Assuming an unobstructed sheet with negligible drag at the salt wedge interface gives a predicted intrusion distance on the order of 10$^0$ (i.e., comparable to the blue transects in Figure 2 A). The theory thus gets the magnitude of intrusion distance correct in the simulations of this study. In some cases with

free-slip kinematic boundary conditions, seawater fills the entire subglacial portion of the model domain. Extending the model domain to capture the full seawater intrusion distance is computationally prohibitive, hence why we have mainly focused on no-slip cases in this study. Indeed, equation 10 predicts unbounded intrusions for $C_d = 0$. Since both the drag coefficient and fluid velocity exhibit great variability along the intrusion regime, and the theory is sensitive to changes in either, it could make choosing which values to use difficult. Another explanation for the disagreement is the assumption of negligible interfacial drag. Robel et al. (2022) finds that including interfacial drag of order $10^{-4}$ reduces predicted intrusion distance by about a factor of 2. Other factors to consider in future development of more realistic theories of seawater intrusion are the potential role of melt feedbacks which we have shown above are important to setting the intrusion dynamics and basal melt rates, and drag from the bottom of the channel which will have different mechanical properties than the ice base.

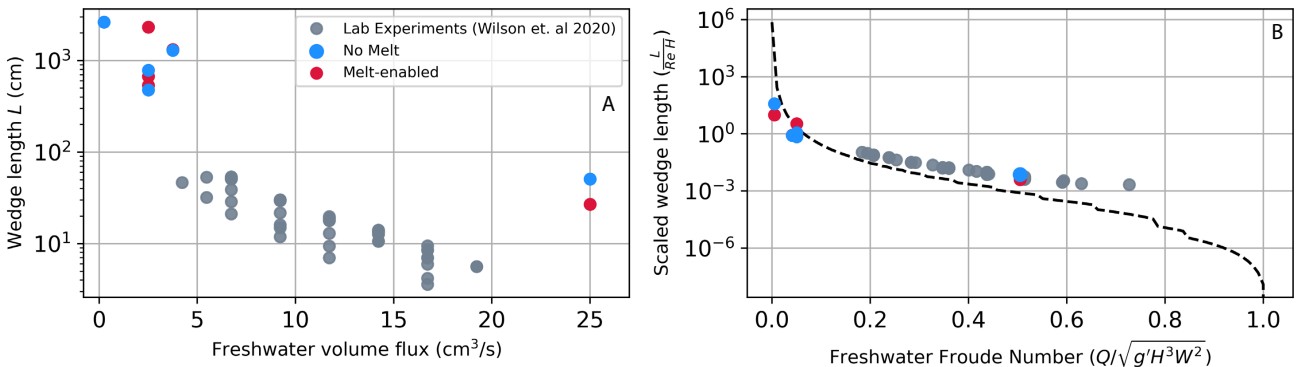

**Figure 6.** Comparison between Wilson et al. (2020) experimental data (gray markers) and intrusion characteristics found in this study (red and blue markers). The red markers represent simulations with melting, and the blue markers represent simulations without melting. The black dashed line is the numerical solution to Robel et al. (2022) with $\gamma = 2$.

There are two limitations to consider within ANSYS Fluent when melting is enabled: (1) the geometry of the subglacial environment does not change, providing no geometric feedback like that considered in Bradley and Hewitt (2024) and (2) changing the ice wall to a velocity inlet means there is no boundary drag acting on the fluid. Both of these limitations will influence overall intrusion distance, however, we can conclude that the addition of buoyant melt feedback does not greatly affect intrusion distance or set vertical mixing dynamics, which is primarily controlled by the upstream freshwater fluxes, similar to estuaries.

Figure 5 demonstrates the disagreements between the melt calculated directly in our high-resolution simulations and the predicted basal melt rates from parameterizations. The S99 parameterization assumes transport of heat and salt is dictated by transfer velocities. The thermal turbulent transfer velocity calculated directly from our simulations is on the lower end of those previously documented due to the low velocities tested here and strong stratification within the intrusion regime. Turbulent transfer velocities are derived from theory and experimental results of pipe flow that assume the fluid is steady and parallel

to the wall. In seawater intrusions flow is spatially varying, with opposing directions of flow parallel to the wall and some flow perpendicular to the wall. The combined effect of low velocities and narrow gaps under grounded ice (we test subglacial domains with 5, 7.5 cm gap height in this study) means that the viscous sublayer and buffer layer together occupy the entire domain considered, and drag from either boundary affects our entire subglacial domain. The turbulent transfer velocities in the S99 parameterization are intended to be used with ocean properties at the edge or beyond the outer boundary layer - if the domain of interest does not develop the full boundary layer then this theory of heat and mass transfer is not appropriate. Since the entire subglacial environment is within the viscous sublayer (where viscous effects dominate) or the buffer layer (where viscous effects and turbulence both occur), the S99 parameterization is not appropriate for described melt from seawater intrusions. This is expected since S99 was not formulated considering domains of seawater intrusion and strong stratification.

If we anticipate viscous effects to dominate in seawater intrusions under grounded ice, then using the thermal and haline molecular diffusivities as so-called "transport velocities" would be appropriate. However, this would require knowing the width of the diffusive sublayer which, given computational constraints for coupled ice-ocean models, cannot be resolved. In polar settings where seawater is warmer than subglacial discharge, double-diffusive convection could be the fitting heat transfer mechanism, acting against the stratification that persists from seawater intrusion. However, this requirement limits this mechanism's relative importance in Antarctica where seawater can be colder than the pressure-dependent freezing point for subglacial discharge (Davis et al., 2023). Double-diffusive convection is dominant where shear is weak, traditionally in low-flow, highly stratified environments, where the geometry does not favor strong natural convection (Rosevear et al., 2021). Observations beneath Ross Ice Shelf near the grounding zone hint at the potential for double-diffusive convection to contribute to vertical mixing when the water column is structured by an upper layer of cold, fresher adjacent to the ice and a mixed homogenized layer at depth (Begeman et al., 2018). Conceptually, we may expect double-diffusive convection to be a good description of the source of mixing and basal melt in these simulations (as hypothesized by Robel et al., 2022), given the right oceanographic settings. Due to model limitations, the results presented here do not include melt from salt diffusion, hindering a direct comparison to a double-diffusive convection parameterization (e.g. eq. B1 in Rosevear et al., 2022). Future work should prioritize parameterizing this form of ice loss in regions of seawater intrusion using high-resolution models that can resolve the boundary layer structure.

Currently, only Bradley and Hewitt (2024) have considered the problem of intrusion-induced melting and allows the subglacial gap to grow in height due to melting in the region of seawater intrusion. Since a taller subglacial gap leads to further seawater intrusion (as predicted in Wilson et al., 2020; Robel et al., 2022) this geometric effect constitutes a positive feedback on seawater intrusion. Our study includes feedbacks from basal melting on flow within the subglacial environment, but not this geometric feedback. We find that intrusion-induced melt does not significantly change the structure of the intrusion, but may act to reduce the regime experiencing intrusion-induced melt. Further, we find that the region of ice experiencing intrusion-induced melt is approximately half of the total intrusion length. Here, we represented the ocean thermal forcing to the ice by the ice adjacent thermal gradient, as opposed to an average of the water column's temperature as done in Bradley and Hewitt (2024). In this study, there is non-negligible melt for $\sim 50\%$ of the intrusion distance, whereas averaging the water column temperature results in melt across the entire intrusion. This latter approach potentially enhances the strength of the geometric

feedback over a wider area of the ice. The widening of the subglacial space will compete with the excess buoyant forcing to influence seawater intrusion development. Furthermore, tidal pumping of intrusions will add another scale of complexity in resolving these feedbacks, since they provide a temporally asymmetric melt rate as demonstrated in Gadi et al. (2023). Further work is needed to consider both feedbacks within a single modeling framework to determine which ultimately dominates, and under what conditions since they operate on different temporal and spatial scales.

Due to computational limitations, our simulations cannot span the full range of discharge velocities and geometric sizes of potential seawater intrusion domains. However, the cases considered here are comparable to several cases where subglacial properties are constrained from observations (Carter et al., 2017; Davis et al., 2023; Washam et al., 2020), indicating the need for a parameterization that operates as a function of boundary layer processes, near-ice (order of cms) seawater properties, and buoyant stability of the water column. Scaling the simulations presented here to ice-shelf cavity size domains will limit our ability to resolve boundary layer processes. This scaling issue is not independent of the constraints within large-scale coupled models, where the coarse resolution will not allow for seawater intrusions to form beyond grounding lines. For such coupled models, further experimentation is needed to identify a sub-grid basal melt parameterization (i.e., similar to the purely numerical schemes identified in Seroussi et al., 2019) that can be applied to grounded ice as a function of local bed characteristics, freshwater flux, and thermal forcing. Complex-sloping ice wall boundaries likely couple the seawater intrusion dynamics to the proglacial plume, providing an avenue to combine these processes in a single framework for coupled models with resolution $\mathcal{O}(\mathrm{kms})$.

The challenges associated with complex fluid dynamics and computational limitations are not limited to domains of seawater intrusion. These challenges also apply to the 100's of meters downstream of the grounding line where a small vertical gap exists between the ice and seafloor. The transition from a two-wall bounded flow to a single-wall bounded flow influenced by buoyant, shear, and diffusive instabilities is currently not represented in coupled models. It is likely this transition couples plume dynamics to intrusion dynamics which may provide a baseline framework to unify the transition. This regime transition, which may be akin to those described in Rosevear et al. (2022), is critical to understanding what sets grounding zone melt rates and therefore what drives retreat.

A sophisticated implementation of seawater intrusion in coarse-grid coupled ice-ocean models is possible. It would require simulating subglacial hydrology, estimating discharge fluxes, and deducing the required scale at which the ocean grid should extend under "grounded" glacier ice. From there, a modified basal melt parameterization could be added to calculate an intrusion melt rate. If a closed-form parameterization based on first-principles theory is not possible in the complex case of seawater intrusion, a modified parameterization based on a comprehensive suite of high-resolution CFD simulations, similar to this study, could be used to constrain an effective parameterization. Alternatively, intrusion-induced melt could be included in a more ad-hoc manner by extending the ocean grid to some prescribed distance inland set by a modified form of the theory proposed by Robel et al. (2022). A prior ice sheet model intercomparison (Seroussi et al., 2019) study found that when sub-grid melt schemes are applied to partially floating grid points (cells that cross the grounding line), projected sea level contributions from the Antarctic ice sheet can be up to twice as high. Such sub-grid melt schemes have a similar effect to intrusion-induced basal melt.

# 5   Conclusions

Seawater intrusion beneath grounded portions of ice sheets leads to complex interactions between the subglacial hydrologic system and the ocean. The ability for seawater to intrude under grounded ice is dependent on the subglacial discharge velocity and geometry of the subglacial environment. Both of these controls are consistent with studies of estuaries (Krvavica et al., 2016), bolstering the idea that grounding zones are subglacial estuaries (Horgan et al., 2013). Turbulent mixing has a neglible effect on intrusion distance, since turbulent viscosity is minimal for the subcritical freshwater velocities required for an intrusion to develop. Here, we found that intrusion-induced basal melting does not drastically alter the intrusion mixing dynamics, which is set by the freshwater velocity. Excess buoyancy forcing from basal melting may constitute a negative feedback that will compete with the geometric positive feedback identified in another study (Bradley and Hewitt, 2024). Our simulations show that seawater intrusion can produce enhanced melting at, and just upstream of, the grounding line where warm seawater occupies most of the subglacial environment, with melt rates on the order of 10's of meters per year for the configurations considered in this study. Current basal melt parameterizations that assume a fully-developed boundary layer under predict the peak in basal melt rates due to seawater intrusion and fail to capture the drop off in melting due to stratification at the nose of the intrusion. Diffusive melting frameworks may do a better job of estimating intrusion-induced basal melt near the grounding line where melt rates are high, however, for this to occur in reality, the seawater has to be warmer than the subglacial discharge, which we expect to limit its applicability in Antarctica. Theories of heat and salt transfer that consider the complex overlap of transport phenomena occurring in the buffer layer should be further considered to better incorporate basal melt from seawater intrusion in coarse models.

Coupled ice-ocean models do a poor job of reproducing observed patterns of enhanced basal melt near grounding lines due to resolution and the assumptions inherent in melt parameterizations (Adusumilli, 2021; Ciracì et al., 2023). Furthermore, the assumption of a hydraulic barrier between subglacial hydrology and the ocean leads to an incomplete consideration of the fluid dynamics at grounding zones. Improving projections of the ice sheet contributions to sea level rise requires a more complete representation of grounding zones as dynamic estuaries where subglacial hydrology, the ocean, and glacier ice all interact.

 **Appendix A: Variable Descriptions**

**Table A1.** Variable descriptions and values for ANSYS Fluent Simulations

| Variable | Description | Value | Units | Equation |
|---|---|---|---|---|
| $\rho_w$ | Seawater density | | kg m$^{-3}$ | 2 |
| $\lambda_1$ | Salinity coefficient in liquidus equation | -5.73 x 10$^{-2}$ | C | |
| $\lambda_2$ | Liquidus point intercept | 8.32 x 10$^{-2}$ | C | |
| $\lambda_3$ | Pressure coefficient in liquidus equation | -7.53 x 10$^{-4}$ | C m$^{-1}$ | |
| $S_w$ | Seawater salinity | | ppt | |
| $T_w$ | Seawater Temperature | | C | |
| $S_f$ | Subglacial discharge salinity | 0 | ppt | |
| $T_f$ | Subglacial discharge Temperature | -0.67 | C | |
| $S_o$ | Seawater inlet salinity | 30 | ppt | |
| $T_o$ | Seawater inlet Temperature | 0.5 | C | |
| $u$ | Freestream fluid velocity | | m s$^{-1}$ | |
| $u_f$ | Subglacial discharge velocity | | m s$^{-1}$ | |
| $u_o$ | Seawater inlet velocity | | m s$^{-1}$ | |
| $H$ | Height of subglacial space | 5, 7.5 | cm | |
| $H_c$ | Height of near-ice grid cell | | m | |
| $\theta$ | Slope of subglacial space | 0., 0.5 | ° | |
| $L_i$ | Simulated intrusion distance | | m | |
| $L_{sge}$ | Length of subglacial environment in simulations | | m | |
| $\mu_t$ | Eddy viscosity | | m$^2$ s$^{-1}$ | 3 |
| $C_\mu$ | Turbulence parameter | | | |
| $\kappa_T$ | Thermal Conductivity of seawater | 0.57 | W m$^{-1}$ C$^{-1}$ | |
| $\epsilon$ | Eddy dissipation | | m$^2$ s$^{-3}$ | |
| $\kappa$ | Turbulent kinetic energy | | m$^2$ s$^{-2}$ | |
| $\dot{m}$ | Modeled melt rate | | m s$^{-1}$ | 4 |

**Table A2.** Variable descriptions and values for processing simulation ouput

| Variable | Description | Value | Units | Equation |
|---|---|---|---|---|
| $C_d$ | Drag coefficient | | | 6 |
| $u_*$ | Shear velocity | | m s$^{-1}$ | 9 |
| $N$ | Buoyancy Frequency | | s$^{-1}$ | 5 |
| $\rho_i$ | Ice density | 918 | kg m$^{-3}$ | |
| $T_b$ | Ice-ocean interfacial temperature | | C | 1 |
| $S_b$ | Ice-ocean interfacial salinity | 0 | ppt | |
| $g$ | Gravitational Constant | 9.81 | m s$^{-2}$ | |
| $\rho_0$ | Reference density | 1000 | kg m$^{-3}$ | |
| $\dot{m}_T$ | Parameterized shear melt rate | | m s$^{-1}$ | 7 |
| $\gamma_T$ | Thermal Turbulent transfer velocity | | m s$^{-1}$ | 8 |
| $\Gamma_T \sqrt{C_d}$ | Thermal Stanton Number | | | |
| $\partial T_i / \partial y$ | Vertical ice thermal gradient | 0 | C m$^{-1}$ | |
| $\kappa_i$ | Thermal conductivity of ice | 2.22 | W m$^{-1}$ C$^{-1}$ | |
| $L_i$ | Latent heat | 334000 | J kg$^{-1}$ | |
| $c_w$ | Seawater heat capacity | 4011.4 | J kg$^{-1}$ C$^{-1}$ | |
| $c_i$ | Heat capacity of ice | 2009 | J kg$^{-1}$ C$^{-1}$ | |
| $Pr$ | Prandtl Number | 13.8 | | |
| $\nu$ | Kinematic viscosity of seawater | $1.95 * 10^{-6}$ | m$^2$ s$^1$ | |
| $K_T$ | Thermal molecular diffusivity | $1.4 * 10^{-7}$ | m$^2$ s$^{-1}$ | |
| $g'$ | Reduced gravitational constant | 10.22 | m s$^{-2}$ | |
| $L$ | Theoretical intrusion distance | | m | 10 |

## Appendix B:  Model Settings and Equations

### B1    Reynolds Averaged Navier Stokes Equations

ANSYS Fluent solves the Reynolds Averaged Navier Stokes (RANS) equations. This formulation of the Navier-Stokes equations decomposes the flow into a mean state and fluctuation about the mean:

$$u_i = \overline{u_i} + u_i'$$
(B1)

where $\overline{u_i}$ is the time-averaged velocity and $u_i'$ is perturbations about the mean velocity. Scalar quantities are also decomposed into their mean and fluctuations. Substituting these decomposed values into the momentum equations yields:

$$\frac{\partial \rho}{\partial t} + \frac{\partial}{\partial x_i}(\rho u_i) = 0$$
(B2)

$$\frac{\partial}{\partial t}(\rho u_i) + \frac{\partial}{\partial x_i}(\rho u_i u_j) = -\frac{\partial p}{\partial x_i} + \frac{\partial}{\partial x_j}\left[\mu\left(\frac{\partial u_i}{\partial x_j} + \frac{\partial u_j}{\partial x_i} - \frac{2}{3}\delta_{ij}\frac{u_l}{x_l}\right)\right] + \frac{\partial}{\partial x}(-\rho\overline{u_i' u_j'}) + \rho g_i$$
(B3)

Equations B2 and B3 are the RANS equations and have the same general form as the Navier-Stokes equations, however, the velocities and solution variables now represent the time-averaged values. In addition, to represent the effect of turbulence, new terms are added. These terms are the Reynolds stresses, $\overline{u_i' u_j'}$. To close this system of equations, the Reynolds stresses must be solved via a turbulence closure. More details on turbulence modeling and our closure choice are given in section B2.

### B2    Turbulence Modeling

The turbulent-viscosity hypothesis (also known as the Boussinesq Hypothesis) is used in turbulence modeling to solve for the Reynolds stresses. This hypothesis states the deviatoric Reynolds stresses (those deviating from the mean) are proportional to the mean strain rate tensor by a positive scalar. This scalar represents the eddy viscosity (also referred to as turbulent viscosity). This relationship is:

$$-\rho\overline{u_i' u_j'} + \frac{2}{3}\rho k \delta_{ij} = \mu_T\left(\frac{\partial \overline{u_i}}{\partial x_j} + \frac{\partial \overline{u_j}}{\partial x_i}\right)$$
(B4)

The only unknown in the system of equations is the eddy viscosity ($\mu_T$), which can be solved by a variety of different turbulence closure schemes. Here, we employ the two-equation $\kappa$-$\epsilon$ closure scheme which solves for eddy viscosity by relating it to the square of turbulent kinetic energy and inverse of turbulent dissipation by a positive scalar $C_\mu$ and fluid density (eq. 3). This closure scheme requires two additional equations to solve for turbulent kinetic energy and turbulent dissipation. These equations are:

$$\frac{\partial}{\partial t}(\rho k) + \frac{\partial}{\partial x_i}(\rho k u_i) = \frac{\partial}{\partial x_j}\left[\left(\mu + \frac{\mu_t}{\sigma_k}\right)\frac{\partial k}{\partial x_j}\right] + G_k + G_b - \rho\epsilon - Y_M + S_k$$
(B5)

for turbulent kinetic energy and:

$$\frac{\partial}{\partial t}(\rho \epsilon) + \frac{\partial}{\partial x_i}(\rho \epsilon u_i) = \frac{\partial}{\partial x_j}\left[\left(\mu + \frac{\mu_t}{\sigma_k}\right)\frac{\partial \epsilon}{\partial x_j}\right] + C_{1\epsilon}\frac{\epsilon}{k}(G_k + C_{3\epsilon}G_b) - C_{2\epsilon}\rho\frac{\epsilon^2}{k} + S_\epsilon \tag{B6}$$

for turbulent dissipation. Buoyancy effects are included in the evolution equation for turbulent kinetic energy by a source term, $G_b$, which can be solved for by:

$$G_b = g_i \frac{\mu_T}{\rho Pr_T}\frac{\partial \rho}{\partial x_i} \tag{B7}$$

where $g_i$ is the component of gravity in the i$^{th}$ direction, $\mu_T$ is the turbulent viscosity, and $Pr_T$ is the turbulent Prandtl number (0.85 default value). The density gradient, $\partial \rho/\partial x_i$ is taken over the i$^{th}$ direction. Buoyancy effects are not included in the dissipation equation due to a higher degree of uncertainty (ANSYS, 2022). Note, in the equation for turbulent dissipation, turbulent kinetic energy is in the denominator, resulting in issues when $\kappa$ approaches zero near wall boundaries. To resolve boundary layer dynamics with the $\kappa$-$\epsilon$ closure, we use the low-Reynolds formulation, which employs damping functions and fixes the singularity that arises with low values of $\kappa$. The low-Reynolds $\kappa$-$\epsilon$ closure we employ here is the Yang-Shih version (Yang and Shih, 1993). In this formulation, the near-wall turbulence timescale is set to the Kolmogorov timescale ($T_k$) which is proportional to $(\nu/\epsilon)^{1/2}$. In doing so, the equation for eddy viscosity near the wall becomes:

$$\mu_T = C_\mu f_\mu k \left(T_k + \frac{k}{\epsilon}\right) \tag{B8}$$

Where $f_\mu$ is the "damping function" and equal to:

$$f_\mu = [1 - exp(-a_1 R_y - a_3 R_y^3 - a_5 R_y^5)]^{(1/2)} \tag{B9}$$

and $R_y = k^{(1/2)}y/\nu$. The constants $a_1$, $a_3$, and $a_5$ are constrained from direct numerical simulation experiments for turbulent channel flow.

The final adjustment to the standard $\kappa$-$\epsilon$ formulation for near-wall flows is to add an additional source of dissipation, which results from inhomogeneity in the mean flow field. This takes the form:

$$E = \nu\nu_T\left(\frac{\partial u_i}{\partial u_j \partial u_z}\right)^2 \tag{B10}$$

This formulation of the low-Reynolds $\kappa$-$\epsilon$ turbulence closure allows for solving the free-stream portion of the flow regime as well as the near-wall region where viscous effects dominate since the added terms tend to zero when turbulence is high.

We employed the low-Reynolds $\kappa$-$\epsilon$ turbulence closure model to allow for increased resolution near the ice boundaries. We tested the Yang-Shih formulation used here against the Launder-Sharma low-Reynolds formulation and saw little disagreement between the intrusion distances simulated (Table B2). However, we note that intrusion distance can change based on the

choice of turbulence closure formulation. This is distinctly different than the amount of turbulence negligibly changing the intrusion distance as we discuss in the results section of the main text. Different closure schemes will resolve the flow structure differently, leading to a nominal change in intrusion distance, but changing the amount of turbulence in a given turbulence

closure scheme does not largely affect intrusion distance.

The $\kappa$-$\epsilon$ turbulence model provides a simple pathway to modulate the amount of turbulent mixing, by changing the $C_\mu$ value. The $C_\mu$ value controls the amount of relative turbulent kinetic energy to dissipation represented in the eddy viscosity value and thus directly affects the Reynolds stresses via the turbulent-viscosity hypothesis. The typical value assigned to $C_\mu$ is 0.09, which has been empirically found for simple wall-bounded flows (Pope, 2000). However, for complex stratified flows or highly

energetic jets, a standard constant value for the whole domain may not be appropriate (e.g. Lai and Socolofsky, 2019). Based on equation 3, it is clear the value of $C_\mu$ is dependent on turbulence dynamics. However, in a modeling framework, we have to prescribe $C_\mu$. Since a 'true' value for $C_\mu$ has not been found for the flow regime considered here (stratified and dynamically variable), we deemed it an appropriate approach to modulate this value to induce more or less turbulent mixing. In setting $C_\mu$, we found numerical instability when $C_\mu \to 1$, therefore limiting our upper-end choice to a factor of 2 of the standard value

used. For the lower-end case, we wanted to avoid choosing a too low of a value, since the solution relaxes to laminar flow when $C_\mu \to 0$. This constrained our lower-end choice to a factor of 0.5 of the standard value.

### B3   Species Transport and Energy Modeling

To simulate the effects salinity and temperature have on fluid density, we employed the species transport model with a user-defined equation based on a linear equation of state for seawater (eq. 2).

Fluent automatically turns on energy conservation when species transport is enabled and solves for water temperature via:

$$\frac{\partial}{\partial t}(\rho_w E) + \nabla(\overrightarrow{u}(\rho_w E + p)) = \nabla(k_{eff}\nabla T - \sum_{j=1} h_j J_j) + S_h \tag{B11}$$

The first two terms on the right-hand side represent energy transfer due to the conduction of heat ($\nabla(k_{eff}\nabla T)$) and species diffusion ($\nabla(\sum_{j=1} h_j J_j)$). Conduction represents heat transfer due to thermal gradients. As salt diffuses in the medium, it also transfers heat due to its unique thermal properties, and therefore must also be included. The contribution to heat transport

from salt has negligible impacts since it is not a leading order term in equation B11 and therefore tracking species contribution to energy transport does not significantly change the results. $S_h$ represents chemical sources of heat; here, there are none so this term is zero (ANSYS, 2009). The left-hand side represents the time evolution of energy and energy advection. This energy transport equation also represents the transport of temperature throughout the domain, with the relation $T_w c_w \rho_w V = E$, where $T$ is temperature, $c_w$ is the heat capacity, $\rho_w$ is the seawater density, and $V$ is the volume.

Salt is transported as an active tracer within the fluid, employing an advection-diffusion-reaction equation:

$$\frac{\partial}{\partial t}(\rho S) + \nabla(\rho_w \overrightarrow{u} S) = -\rho_w D_s \nabla S + R \tag{B12}$$

Where $\vec{u}$ is the velocity vector, $D_s$ is the diffusion coefficient for salt (set to $1.5 \times 10^{-9}$ m$^2$s$^{-1}$), and $R$ is the production rate from reactions. The production rate for salt in these simulations is zero. The boundary salinity, $S_b$, is set to the near-ice grid cell (scale of nanometers) which follows the evolution equation given by B12. This evolution equation accounts for advective and diffusive transport mechanisms. The boundary salinity is used in the liquidus condition (equation 1) to set the thermal boundary condition for the ice interfaces. In the reference simulations, $S_b$ only evolves due to advection and diffusion from the salt wedge entering the subglacial domain and the freshwater layer exiting the subglacial space as a plume. For the simulations with melting, $S_b$ is diluted as fresh, cold water enters the domain normal to the ice boundary. This dilution occurs because Fluent is a finite volume solver and so balances inflows and outflows across each control volume in the grid, while conserving overall mass balance in the domain via a pressure outlet (red arrow Figure 1). The thermal boundary condition (given by equation 1) adjusts based on this dilution. Similarly, the chilling of near-ice waters occurs due to local injection of meltwater at the freezing point.

## B4 Domain

The 2-dimensional domain is defined as a long and thin subglacial environment attached to a tall rectangular ocean basin. The length of the subglacial environment is 20 m for most cases. The slowest freshwater velocity simulations and the taller subglacial environment simulations required a longer domain to reach a quasi-steady-state and therefore their subglacial environment length is 40 m. The height of the subglacial domain is 5 cm for all cases except the taller scenario where the height is 7.5 cm. The ocean tank's height is 5 m and its width is 2 m for all cases. For analysis, we designate the 'grounding line' or the point where the subglacial environment meets the ocean basin to be at x = 0 m. However, in the model domain, this point exists at x = 2 m.

A freshwater inlet is defined at the rightmost boundary of the subglacial environment. A seawater inlet is defined at the leftmost boundary. A pressure outlet is set at the top of the ocean domain, enforcing a no-gradient flux across the boundary. The ice boundary walls are defined at the top of the subglacial environment and the right side of the ocean domain. For both melting and non-melting simulations, these ice walls have a thermal boundary condition dependent on near-wall salinity and pressure (eq. 1). Neither ice wall boundary allows for salinity diffusion, which would be another mechanism of melt to account for. The non-melting simulations employ a no-slip kinematic condition, which forces the fluid velocity to be zero at the wall. For cases with melting, the top boundary of the subglacial space is turned into a velocity inlet to simulate melt. In designating this boundary as a velocity inlet, a free-slip kinematic condition is required. The downward vertical velocity set at the melting ice boundary follows equation 4. We turn off melting for the vertical ice boundary to isolate the intrusion-induced melt from the vertical plume dynamics that would arise from the vertical ice boundary.

## B5 Meshing

For the given turbulence closure scheme employed here (Yang-Shih low Reynolds k-e turbulence model) a nondimensionalized distance, $y^+$, must be <= 5. This follows the law-of-the-wall principle, where mesh thickness near the wall must be coordinated

with the closure scheme for near-wall viscosity effects to be rendered correctly. This nondimensionalized distance $y^+$ is a function of dynamic viscosity, density, distance normal to the wall, and shear velocity.

$$y^+ = \frac{u_* y}{\nu} \tag{B13}$$

Given a value for $y^+$ and know the kinematic viscosity, $\nu$, all that is needed is the shear velocity, $u_*$, in order to find the necessary dimensional y distance of the first grid cell. To find $u_*$ we need the skin friction $C_f$. For laminar flows, the skin friction is:

$$C_f = \frac{16}{Re} \tag{B14}$$

Where $Re$ is the Reynolds number with the characteristic length being the channel length :

$$Re_L = \frac{uL}{\nu} \tag{B15}$$

To find $u_*$:

$$u_* = \sqrt{\frac{\tau}{\rho_w}} \tag{B16}$$

Where $\tau = \frac{1}{2}\rho C_f u^2$. Rearranging we then get:

$$u_* = \sqrt{\frac{1}{2} C_f u^2} \tag{B17}$$

We can then plug this value for $u_*$ back into B13 equation to calculate the near-wall mesh thickness. We generate a mesh for each freshwater velocity and geometric combination (5 total) following the steps above and use inflation layers to increase resolution near the wall boundary, but reduce it near the center of the domain. Generating the mesh as such greatly reduces total simulation run time and data storage requirements.

## B6   Run-Time

To keep the simulation time consistent, each simulation runs for 43,200 s (12 hrs).After this period of time, we evaluate if the solution has reached a quasi-steady-state by evaluating the change in density of the subglacial environment over 100 time steps. If this change is below $10^{-4}$ kg m$^{-3}$ then a quasi-steady-state has been achieved. A few runs required longer simulation times to reach such a steady state and were consequently run for another 12 hrs.

To set the time step we used the eddy turnover time. This is the largest time step we can take without causing non-physical effects and is represented by the:

$$t = \frac{h}{u_*} \tag{B18}$$

where $h$ is the height of the channel. For each simulation, a time-step of 5 s was chosen in accordance with the fastest

freshwater flow (in turn has the largest shear velocity, and therefore smallest eddy turnover time). Putting this together, for all

runs, a total of 8,640 time steps were taken at a time step of 5 s. For every 20th time step (100 s) data was exported along a

middle transect, the ice base, and the domain's entire fluid surface.

Each simulation is initialized with a fresh and cold subglacial environment ($S$ = 0 ppt and $T_w$ = -0.75 °C) and a warm and

salty ocean basin ($S$ = 30 ppt and $T_w$ = 0.5 °C) as shown in Figure B1. The simulation then runs for 12 hrs of simulation

time. Each simulation experiences a period of 'spin-up' time where the intrusion develops, and almost all simulations reach a

quasi-steady-state. We define a quasi-steady-state to occur once the time change in subglacial environment's average density is

less than 0.0001. The results presented in the main body of this work are the time-averaged results once a quasi-steady-state is

reached and the following 5000 s of simulation time.

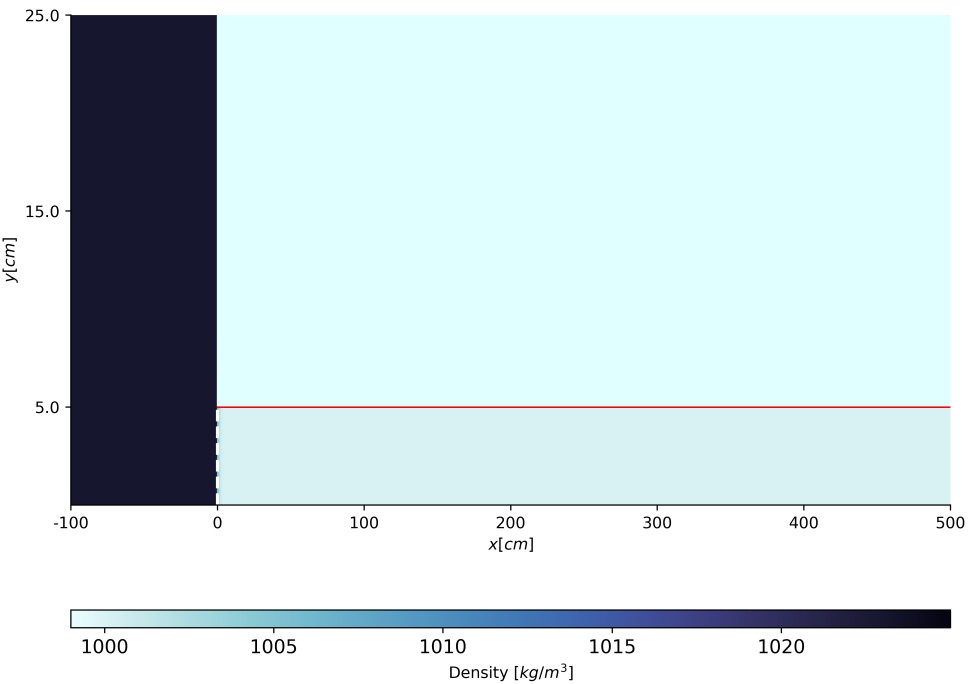

**Figure B1.** Initial condition for all model simulations.

## B7 Variable Ocean Forcing

**Table B1.** Simulations with varied ocean velocity.

| $u_f$ (cm s$^{-1}$) | $u_o$ (cm s$^{-1}$) | $C_\mu$ | H (cm) | $\Theta$ (°) | L$_{sge}$ (m) | L$_i$ (m) |
|---|---|---|---|---|---|---|
| 0.5 | $10^{-6}$ | 0.09 | 5 | 0 | 20 | 4.81 |
| 0.5 | 5 | 0.09 | 5 | 0 | 20 | 4.78 |
| 0.5 | 0.05 | 0.09 | 5 | 0 | 20 | 4.79 |

Previous theoretical work on seawater intrusion neglected a background ocean velocity in order to get a simple relationship between freshwater flux and intrusion distance. To test the appropriateness of this simplification, we varied the seawater inlet velocity across 3 magnitudes ($u_o$ = 5, 0.5, and 0.05 cm s$^{-1}$). In addition, we also tested the smallest ocean velocity the model would converge at, $u_o$ = $10^{-6}$ cm s$^{-1}$. Between the ocean velocities tested, there is essentially no difference in intrusion distance, mixing characteristics, or vertical structure. This supports the idea that the exchange flow that develops within the intrusion is

set by the flux of buoyant freshwater, and is independent of ocean currents, similar to estuaries. A summary of these results is given in Tables B1 and 1.

## B8 Equation of State

**Table B2.** Simulations with varied modeling choices for low-Reynolds turbulence closure and equation of state

| $u_f$ (cm s$^{-1}$) | Model Choice | $C_\mu$ | H (cm) | $\Theta$ (°) | L$_{sge}$ (m) | L$_i$ (m) |
|---|---|---|---|---|---|---|
| 0.5 | Launder-Sharma Low-Re $\kappa$-$\epsilon$ closure | 0.09 | 5 | 0 | 20 | 4.57 |
| 0.5 | Higher Order E.O.S. | 0.09 | 5 | 0 | 20 | 4.59 |

For all the simulations reported in the main body of this work a linear equation of state from Roquet et al. (2015) (eq. 2) is used to solve for fluid density. We also tested a higher-order equation of state from Roquet et al. (2015):

$$\rho_w = 1000 - \frac{C_b}{2}(T_w - \Theta_0)^2 - T_h Z T_w + b_0 S_A \tag{B19}$$

Here, $C_b$ = 0.011 kg m$^{-3}$ K$^{-2}$, $T_h$ = 2.5 x 10$^{-5}$ kg m$^{-4}$K$^{-1}$, $b_0$ = 0.77 kg m$^{-3}$ (kg g$^{-1}$), and $\Theta_0$ = -4.5° C. With the higher-order equation of state, the baseline density of the freshwater and the seawater were lower than the linear equation of state. However, the distribution of salt and heat remained the same because they are transported by their respective equations (eq. B12, B11). Because of this, the intrusion distance did not vary significantly between the two equation of state formulations.

## Appendix C: Calculating the Drag Coefficient

The drag coefficient cannot be prescribed for the simulations but is rather diagnosed from the model output. To do so, we use the relationship between the drag coefficient, $C_d$, and the wall shear stress, $\tau_w$ (Pope, 2000):

$$Cd = \frac{2\tau_w}{\rho u^2} \tag{C1}$$

The wall shear stress is equal to:

$$\tau_w = \mu \left( \frac{\partial u}{\partial y} \right) \tag{C2}$$

We can rearrange and solve for $C_d$ as a function of the near-wall velocity gradient which produces equation 6.

## Appendix D: Turbulent Viscosity Results

Turbulent viscosity is helpful to understand the strength of turbulence-induced mixing within a fluid. In the fluid regimes tested here, with low Reynolds numbers, turbulent viscosity is small and comparable to seawater's kinematic viscosity. It is for this reason that modulating the degree of turbulent mixing does not impact the intrusion distance. Turbulent viscosity is greatly reduced over the length of the intrusion, likely due to enhanced stratification suppressing mixing. For the middle and slowest freshwater velocities, the intrusion distance is near the transition point defined by the critical gradient Richardson number, which relates the relative importance of shear to buoyancy.

## Appendix E: Vertical Profiles for Retrograde and Taller Subglacial Channel Geometries

Changing the subglacial environment's geometry did not greatly impact the overall structure of the intrusion even if it did change the intrusion distance. The vertical distribution of heat and salt for both the retrograde and taller geometries are qualitatively similar to their comparable base case (Figure 3E,F and Figure E1A,B,E,F). Consequently, the degree of stratification, as measured by the buoyancy frequency, is similar (Figure 3H and Figure E1D,H). The structure of the exchange flow is quite similar as well, albeit a slighty stronger exchange flow for the taller geometry due to enhanced flux of freshwater at the subglacial inlet.

*Code and data availability.* The code and time-averaged data for all figures and data processing can be found at this repository: https://github.com/madiemamer/seawater-intrusion.git. If interested in the original data output by ANSYS Fluent please contact the corresponding author.

*Author contributions.* A.A.R., E.W., C.C.K.L, and M.M. conceived the presented work and methodology. M.M. conducted all simulations and analyses. A.A.R. aided in the simulation set-up and analysis. C.C.K.L. provided consultation on the methods and background theory.

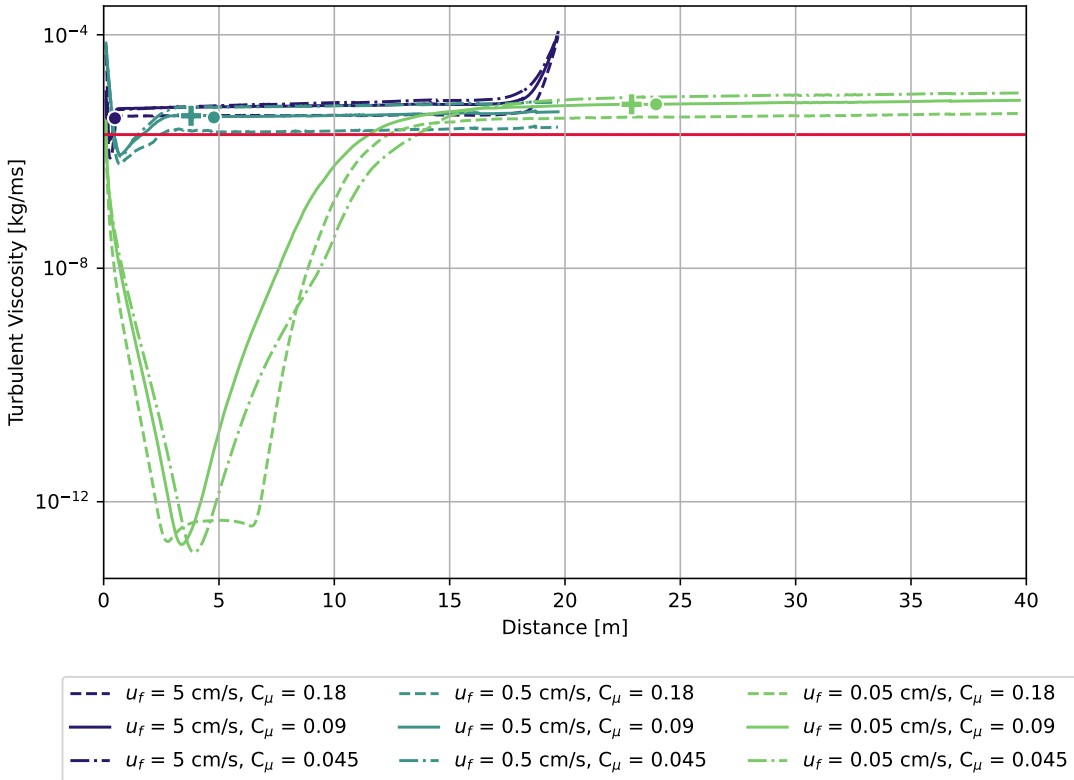

**Figure D1.** Turbulent viscosity for each freshwater velocity and an each turbulent mixing scenario. Solid lines are medium turbulent mixing ($C_\mu = 0.09$), dashed lines are for high turbulent mixing ($C_\mu = 0.18$), and dash-dot lines are for low turbulent mixing ($C_\mu = 0.045$). The red line represents seawater's kinematic viscosity. The dots represent the intrusion distance for the medium turbulent mixing case for the given freshwater velocity. The cross markers represent the point at which the critical gradient Richardson number is crossed.

P.W. and E.W. contributed to interpreting the analysis and choosing the appropriate melt parameterization schemes to test. A.A.R. supervised the findings of this work. M.M. wrote the manuscript with assistance from A.A.R. and consultation with C.C.K.L, E.W., and P.W. .

*Competing interests.* The authors declare that they have no conflict of interest.

*Acknowledgements.* We acknowledge the computing resources that made this work possible provided by the Partnership for an Advanced
Computing Environment (PACE) at Georgia Tech in Atlanta, GA. We would like to thank research scientist Fang (Cherry) Liu for her assistance on challenges related to PACE and HPC. Financial support for this work came from startup funding from Georgia Tech and the University System of Georgia.

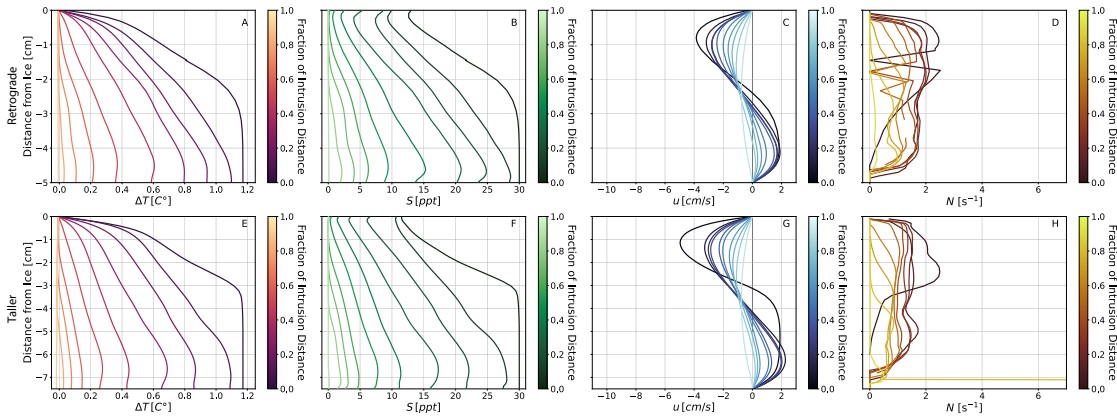

**Figure E1.** Time-averaged vertical profiles of thermal forcing (A,E), salinity (B,F), x-component of velocity (C,G), and buoyancy frequency (D,H) along the seawater intrusion for the retrograde (A-D) and taller (E-H) subglacial environment geometries with $u_f = 0.5$ cm s$^{-1}$ and medium turbulent mixing.

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
