# Peer review of "Modeling Mixing and Melting in Laminar Seawater Intrusions Under Grounded Ice"

_EGUsphere, 2024_

## Author Comment (AC1)

Response to Reviewer #2 for Manuscript "High-Fidelity Modeling of Turbulent Mixing and Basal Melting in Seawater Intrusion Under Grounded Ice" by Mamer, Robel, Lai, Wilson, and Washam

**Summary:**
This paper studies the estuarine-type dynamics of seawater intrusion/subglacial discharge upstream of an ice sheet grounding line in a CFD model (ANSYS fluent). The subglacial environment is two-dimensional with a large aspect ratio (5cm high and 3-5m long) and is forced by fresh subglacial discharge upstream and salty inflow from the ocean downstream. The authors investigate the structure and distance of the saltwater intrusion, and its sensitivity to various parameters (primarily freshwater discharge velocity). They find that that the height of the subglacial environment and the velocity of the freshwater discharge are strong controls on the intrusion distance, and the strength of turbulent mixing has a somewhat weaker/more ambiguous effect. The authors also investigate basal melt at the upper surface of the subglacial region and how it affects the intrusion (to do so they must change the boundary condition at the top surface away from the no-slip condition), finding that melt decreases the intrusion distance, offering a possible negative feedback.

This is an interesting study of an important process; however, I have several major concerns with the paper that may affect the conclusions. If they are addressed, I would be happy to review a revised manuscript.

We appreciate the thoughtful review and valuable feedback given by reviewer 2. We have identified the reviewer's main concerns to be:

1. The lack of turbulence statistics in describing differences between laminar and turbulent cases and in justifying the modeling choices to vary $C_\mu$.
2. The lack of a pressure and salinity dependent thermal boundary condition for the ice boundary. Furthermore, the lack of a free-slip case to compare with the melt-enabled simulations.
3. The incomplete discussion of a double-diffusive/diffusive mixing regime and inappropriate diffusive melting comparison to modeled melt.
4. Overall issues with clarity in describing the numerous simulations.

To address these concerns, we will re-run each of our simulations with a salinity and pressure-dependent thermal boundary condition and add additional cases with a free-slip ice kinematic boundary condition. In discussing model choices and comparing between cases of varying degrees of turbulent mixing, we will refer to model values of turbulent viscosity. The consideration of diffusive-convective mixing will be limited to the discussion section, and we will work to improve the clarity of the results and analysis.

Below, we address each concern in greater detail. The reviewer's comments are written in black, our responses are given in red, and the original manuscript text is in grey.

**First Comment:**

I am concerned about the turbulence modelling and the lack of inclusion of turbulence quantities in the paper. The fidelity (reported to be high!) of these simulations relies on the appropriateness of the turbulence closure, however, the reader is not given any context for the choice/s of C_mu,

We have added to line 563:

> "The typical value assigned to $C_\mu$ is 0.09, which has been empirically found for simple wall-bounded flows (Pope, 2000). However, for complex stratified flows or highly energetic jets, a standard constant value for the whole domain may not be appropriate (e.g. Lai et al. 2019). Based on equation 2, it is clear the value of $C_\mu$ is dependent on turbulence dynamics. However, in a modeling framework, we have to prescribe $C_\mu$. Since a 'true' value for $C_\mu$ has not been found for the flow regime considered here (stratified and dynamically variable), we deemed it an appropriate approach to modulate this value to induce more or less turbulent mixing. In setting $C_\mu$, we found numerical instability when $C_\mu \to 1$, therefore limiting our upper-end choice to a factor of 2 of the standard value used. For the lower-end case, we wanted to avoid choosing too low of a value, since the solution relaxes to laminar flow when $C_\mu \to 0$. This constrained our lower-end choice to a factor of 0.5 of the standard value."

Since $C_\mu$ away from the wall (near-wall would be set by the low-Reynolds formulation) can be found via the relationship:

$$C_\mu = \frac{\nu_t \epsilon}{k^2 \rho} \qquad\qquad (R1)$$

it is clear that the 'real' value of $C_\mu$ is highly dependent on turbulence dynamics. In a modeling framework, we have to prescribe $C_\mu$ to obtain eddy viscosity, it is not a post-processing derived value. Since a 'true' or 'real' value for $C_\mu$ has not been found for the flow regime considered here (stratified), we deemed it an appropriate approach to modulate this value in order to induce more or less turbulent mixing. The goal in doing this is to answer a specific scientific question, namely: "What is the role of turbulent mixing in seawater intrusion and melting?"

The empirically derived value for $C_\mu$ that is traditionally used, 0.09, is for standard wall-bounded simple flow regimes. Under scenarios of high energetics (like a jet-plume) $C_\mu$ can approach 0.3 (Lai et al., 2019). For our fastest freshwater flux (Re = 2400), a jet-plume does form, therefore making a relatively increased $C_\mu$ value potentially realistic. Our original intention in setting $C_\mu$ was to span three orders of magnitude, with the standard value of 0.09 as the 'middle case'. However, initial runs demonstrated instability within the model

when $C_\mu$ was set to 0.9. On the other end, when $C_\mu$ is very low ($\to$ 0) the flow regime becomes laminar, which would not be helpful to address the research question at hand. Based on this, we chose to range the value by a factor of ¼ .

nor are we given any more intuitive quantities (e.g. the resulting turbulent diffusivity) to better understand the effect of varying C_mu, and how increased turbulence affects the flow.

We agree that turbulent viscosity would be more intuitive in explaining what changing the $C_\mu$ value results in. Here, we have plotted the turbulent viscosity (Figure R1) across all flux cases tested with each $C_\mu$ value in a simplified pipe flow example. For the revised simulations in our simulated domain, we will produce a similar figure that will be referenced in the appendix/supplementary.

[Figure]

**Figure R1**

Turbulent Viscosity for each flux tested, with varied $C_\mu$. Transects are taken along the middle of the pipe domain. The solid lines represent $C_\mu$ = 0.09, the dashed are for $C_\mu$ = 0.045, and the dotted lines are $C_\mu$ = 0.18. This is for a simplified pipe domain with length = 1m, height = 0.05m, and a velocity inlet prescribed at x = 1m. A pressure outlet (zero gradient flux boundary condition) is defined at x = 0m.

Without this information and context, I have low confidence in the modelling overall, especially since the effect of varying C_mu is non-monotonic for some cases (Fig 2B) and counter to my expectations for others (Figs 2A & C), i.e. I would have expected that increased mixing would decrease intrusion distance, whereas the authors find the opposite result.

Note, that the eddy viscosity in Figure R1 is lower than the molecular viscosity of water (~1.8e-3 Pa s) in most cases tested here, which indicates that turbulence does not play a dominant role in heat/salt/momentum transport. This likely contributes to our conclusion that turbulence in this domain does not affect intrusion distance significantly.

My recommendations to address this are as follows:
- o   Present the turbulent diffusivity (or viscosity) alongside T, S, u. In main MS or in a supplement

This will be done with the updated experiments, as demonstrated in Figure R1.

- o Present the Reynolds numbers for the cases

This has been done. We have added to line 84:

"The corresponding Reynolds numbers for the given geometry and freshwater flux are 25, 250, and 2500 for the low, medium, and fastest freshwater cases presented."

- o Give some context for the choice of C_mu for similar flows – I presume the engineering literature can provide. Stratified plane couette flow may be a starting point in terms of a bounded, stratified flow.

We have modified line 563 as shown above.

**Second Comment:**

I don't understand why the temperature profiles have much sharper gradients than the salinity gradients (c.f. the darkest line in Fig A8A to the darkest line in Fig A8B). The authors mention the steep T gradients (not seen in S) in line 202 *(Such a steep thermocline is most likely due to the temperature boundary condition we imposed on the horizontal ice boundary)* however, I am concerned that the problem runs deeper than different boundary conditions. Why, for example, is temperature at 2cm depth at the GL entrance at 0.5 degrees (i.e. unmodified from ambient conditions), while salinity is <20ppt at the same location? There may be a serious issue with the mixing of these scalars which would significantly affect your results. Temperature-salinity plots may help determine if there is a problem.

We appreciate the reviewer bringing this to our attention. We have found a mis-assigned diffusion coefficient. We have re-run one non-melting scenario ($u_f$ = 0.005 m/s, $C_\mu$= 0.09) for the full model run-time (12 hrs at 5 s time steps). This updated diffusion coefficient does affect the horizontal gradient of salinity (Figure R2, panel B). Now, near the grounding line, both the temperature and salinity have been modified from ambient conditions. The quasi-steady-state intrusion distance is not greatly affected, with intrusion distance still at ~ 2m. In the updated model runs, we will use the correct haline diffusion coefficient.

[Figure]

**Figure R2**

Vertical profiles of temperature and salinity at various distances upstream of the grounding line for experiments with fixed values for salinity fickian diffusion coefficient. These profiles are from the time-averaged simulation results, with a total model run time of 12 hrs at 5 s time steps. Ocean inlet conditions are 30 ppt and 0.5 deg C. Freshwater inlet conditions are set to the pressure freezing point at a glaciostatic pressure of 1000m and zero salinity. Note, that the freshwater temperature is negative because of new thermal boundary conditions that are dependent on pressure and salinity effects, further discussed in comment 6.

**Third Comment**

By comparing the temperature and salinity profiles as u_f is increased (Figs 3 A,B, Figs A8 A,B,D,E), it appears to me that the greatest control on mixing is time spent in the subglacial channel. For example, at u_f=0.05 cm/s salinity is quite well-mixed over the full depth of the channel, and varies mostly with distance along-channel, implying that diffusion (rather than advection) is dominating transport. For u_f=5cm/s, the top, outflowing layer remains fresh, indicating that the transport dominated by advection. This is somewhat counter to my expectations that turbulent mixing should be stronger for the higher velocity cases. This result should be investigated and discussed. As for comment (1), plots of the turbulent diffusivities for each case may be enlightening.

This is an interesting takeaway that we have not yet explored. We agree that this should be discussed and is potentially similar to observed characteristics in estuaries. In addition to the turbulent diffusivities, a local evaluation of the Peclet number may provide insight. It may also require investigating if this relationship is only dependent on freshwater velocity, or if the same vertical mixing characteristics develop for variable ocean velocity – even if intrusion distance does not change.

Added to line 207:

> "The degree of vertical mixing increases with decreasing freshwater velocity, which is similar to observed mixing dynamics in estuaries (Montagna 2012). For high freshwater fluxes, a well-developed salt-wedge forms and high stratification persists (Figure 3 panels A and B, Figure A8 panels A and B). For low freshwater fluxes, vertical mixing is strong and the water column homogenizes (Figure A8 panels D and E). Similar behavior is observed in estuaries, where estuaries dominated by tides and wind-driven mixing generate enough boundary shear to overcome stratification and initiate vertical mixing (Montagna 2012)."

**Fourth Comment**

The different BC between the melt-enabled and no-melt cases makes it impossible to attribute changes in the intrusion to the effect of melting alone. An additional case is needed with free slip and no melting to better separate these effects.

We agree that an additional free-slip case would benefit the analysis. As additional simulations, we will run each freshwater flux scenario with $C_\mu$ = 0.09 and a free-slip boundary. In Figure R3c below, we show an example of such flow for a simplified pipe flow case.

**Fifth Comment**

There are numerous different simulations included in the paper, however, few where one variable is systematically changed. This makes the paper/results hard to follow at times. A results table or bar chart showing how the intrusion distance changes across simulations would go some way to addressing this.

Based on this suggestion, we will revise the experiments table in the appendix (Table A2) and add it to the main body of the text. Referencing this table early on will help orient the experiments and results. We will include Reynolds number, intrusion distances, drag coefficients, statistics of melt rate, and statistics of mixing as well. It should then be clear that there is a "baseline case" and a number of experiments with only a parameter varied. However, there are other cases where we cannot avoid changing more than one model aspect at a time due to model limitations.

**Sixth Comment**

No salinity effect on the melt. In the simulations, the interface is salty (Fig 3E) which will act to depress the freezing temperature and therefore the interface temperature, altering the melt rate. The authors should state why they have not included this extremely

important effect (in more detail than "model limitations" line 437). In addition, some discussion of the likely effect of this simplification on the results is needed.

The thermal boundary condition for the ice boundary was originally chosen to keep the model simple. We expect that melt rates would only increase with a salinity and pressure-dependent boundary condition due to an increase in the overall thermal driving.

Based on recommendations from both reviewers, we will modify the thermal boundary condition of the upper boundary ('the ice') to be a function of near-ice salinity and a reference glaciostatic pressure (e.g. at 1000m).

We specifically edited lines 87 - 88:

> "The ice wall boundaries have a pressure and salinity-dependent thermal boundary condition of:
>
> $$T_b = S_b \lambda_1 + \lambda_2 + z \lambda_3 \qquad\qquad (R2)$$
>
> Here, $\lambda_1$, $\lambda_2$, and $\lambda_3$ are constants, and the boundary salinity is $S_b$. The depth of the ice is equal to $z_b$, in these simulations we set this to be 1000 m. Both the vertical and horizontal ice boundaries have a no-slip kinematic condition in the non-melting cases, forcing the freestream fluid velocity to be zero at the ice wall."

In addition, we have added to line 147:

> "Therefore, the melting cases have a free shear kinematic boundary condition. The temperature of the inflowing meltwater is set by equation R2."

Where equation R2 is the pressure and salinity-dependent freezing point equation.

We have also edited lines 519 - 522 in the appendix:

> "For both melting and non-melting simulations, these ice walls have a thermal boundary condition dependent on near-wall salinity and pressure (eq. R2). Neither ice wall boundary allows for salinity diffusion, which would be another mechanism of melt to account for. The non-melting simulations employ a no-slip kinematic condition, which forces the fluid velocity to be zero at the wall. For melt-enabled cases, the top boundary of the subglacial space is turned into a velocity inlet to simulate melt. In designating this boundary as a velocity inlet, a free-slip kinematic condition is required. The downward vertical velocity set at the melting ice boundary follows equation 4. We turn off melting for the vertical ice boundary to

isolate the intrusion-induced melt from the vertical plume dynamics that would arise from the vertical ice boundary."

We will update the results and discussion section accordingly with the new simulations. Figure R3 is an example run of simplified pipe flow with such a thermal boundary condition.

[Figure]

**Figure R3**

Example run of a 2D pipe-like geometry with ice at y = 0.05 m and a velocity inlet prescribed at x = 1m. A zero-gradient flux outlet is prescribed at x = 0 m. This case has a free-slip condition at the ice base (along y =0.05) as well as a salinity and pressure dependent thermal boundary condition (here, we set the ice thickness to be 1000 m). Panel A is temperature, panel B is density, and panel C is velocity magnitude. In panel C, a classical boundary layer forms along the flow direction at the bottom boundary where there is a no-slip condition. In panel A and B, a thin thermal boundary layer forms due to the salinity and pressure dependent thermal boundary condition cooling the near-ice water.

**Seventh Comment**

Double diffusive framework. Equation 9 is the same as Equation 4 (with marginally different thermal diffusivity) and is therefore not a parameterization to be tested, rather a re-stating of your melt BC. I propose removing Fig 6B and the "diffusive melt" in Fig 5. Notation m_DC and m_DDC (which are interchangeably used) are inappropriate here.

We recognize this and attempt to clarify that this comparison was not a parameterization to be tested, but rather a different 'transfer mechanism' (i.e. thermal diffusion instead of thermal conduction). However, we understand this may be redundant and not scientifically important and have removed it.

**Eighth Comment**

Transfer velocities/Stanton numbers (Figure 6 )– I don't really understand the purpose of C and D. All D shows is the weak dependence of the Stanton number on ustar at low ustar (see the denominator of (7)), and all C shows is the same but multiplied by ustar. Since the S99 parameterization can't accurately predict melting for your simulations, this (trivial) result becomes misleading, as readers may think that it lends support to a certain Stanton number being useful more broadly for modelling ice-ocean interactions.

The intention of Figures 6C and 6D is to measure the turbulent transfer velocity ($\gamma_T$ ) and Stanton number using the physical flow-field values from our experiments. In doing so, we want to understand how 'efficient' this fluid domain is at transporting heat, relative to other published/observed values. Furthermore, we aim to explain how the value obtained for a Stanton number or turbulent transfer velocity is grid-dependent, meaning in a stratified regime with an interfacial shear layer you could have an order of magnitude difference depending on where the reference point is chosen. We recognize this result might be trivial, based on the turbulent transfer velocity's dependence on shear velocity. However, to think about parameterizing such a complex fluid regime, we may try to visualize how useful values (i.e. the Stanton number) vary vertically.

If you want to compare the Stanton number of your simulations to those found in the literature, you need to rearrange equation (6) less the conductive ice flux to solve for the Stanton number (and the (unitless) transfer coefficient \$Gamma_T$) using your model output melt rate, temperature etc.

We agree that there are benefits to both approaches and will also calculate this 'tuned' Stanton value with the new model results.

**Other Comments**

1. Line 83 – move references (carter…) after "non-summer months"
   - Fixed

2. Line 97 - "Our vertical domain size is at the upper bound of the viscous sublayer length scale that could exist between a well-mixed boundary layer and the ice" I don't know what this means, please clarify.
   - The domain size and velocities (Reynolds number) used in these simulations constrain the development of the boundary layer and freestream flow. In this Reynolds number regime, we don't see the development of a complete boundary layer with flow unaffected by the boundary (freestream flow)
   - To clarify, we have made these edits in the main text at lines 97 - 98:

      For the tested freshwater velocities, the vertical domain size hinders the development of a full boundary layer. Instead, everywhere in the domain, the fluid feels the effects of the wall boundary."

3. Line 154 – should be dT/dy in your coordinates
   - Fixed

4. Line 177 – "vary by a factor of 1000 **in response to the range of input velocities tested here**".
   - Fixed

5. Line 179 – insert comma before "increased"
   - Fixed

6. Line 180 – "For the middle freshwater velocity (Figure 2 C)," - should this be 2A?
   - Fixed. We appreciate you finding this error.

7. Line 182 – "To contrast the effects of turbulent mixing, we tested a laminar flow case with no turbulent mixing (green line Figure 2 A) and saw no meaningful difference in intrusion distance." It would be good to compare and contrast this case more, i.e. how different is the T/S structure? If it's not different, then presumably turbulent mixing is not occurring in the channel.
   - We appreciate this suggestion and will include a more thorough exploration of the difference between turbulence viscosity, vertical structure (e.g. buoyancy frequency), and T/S space between the laminar and turbulent cases with the new simulation results.

8. Line 197/8 – refs should not be in parentheses.
    o Fixed.

9. Line 212 – How is Cd calculated? i.e. at what value of y are ustar and u evaluated? Also, as mentioned earlier I think Cd is a main result and this figure should come to the main text.

    o We will refine the drag figure in the appendix (Figure A5) and the methodology to calculate drag. We will add more discussion about our simulated drag coefficients to the results section with the new model results.

    o We have edited lines 140 - 141 in the main body:

        "Within the model framework here, the wall drag coefficient is not a free parameter to be set, but rather diagnosed from the simulations via the relationship (Pope, 2000):

        $$C_d = \frac{2v}{\overline{u^2}}\left(\frac{\partial \overline{u}}{\partial y}\right)$$
        (R3)

        where $\partial \overline{u}/\partial y$ is the mean shear velocity and is evaluated over the upper 0.5 cm of the domain. The mean free stream current speed, $\overline{u}$, is obtained from the centerline flow. The kinematic viscosity is represented by $v$. The full derivation of calculating the drag coefficient is given in section A4."

    We also added to section A4:

        "The drag coefficient cannot be prescribed for the simulations but is rather diagnosed from the model output. To do so, we use the relationship between the drag coefficient, $C_d$, and the wall shear stress, $\tau_w$ (Pope, 2000):

        $$C_d = 2\tau_w / (\rho \overline{u^2})$$
        (R4)

        The wall shear stress is equal to:

        $$\tau_w = \mu \left(\frac{\partial \overline{u}}{\partial y}\right)$$
        (R5)

        and $\mu$ is the dynamic viscosity. We can rearrange and solve for $C_d$ as a function of the near-wall velocity gradient as shown in equation R3."

10. Line 241 – "Reduction in velocity gradients arises from an increase in stratification, suppressing turbulence, and the kinematic boundary condition being a velocity inlet and not a no-slip wall." – as per major comment 4, actually you can't isolate these effects currently and a no-slip no-melt case is needed.
    - We appreciate this insight and agree with this recommendation. We will conduct a free-slip experiment to compare with the melt-enabled scenarios, per the suggestion in major comment 4. The comparison between melt-enabled cases and free-slip cases will replace lines 229-241.

11. Line 256 –Predominantly horizontal motion?
    - The motion is predominately horizontal from how the kinematic boundaries are prescribed. However, the horizontal density gradient that arises due to the characteristic 'wedge' shape of the seawater intrusion could introduce vertical convective motion to 'flatten' out the intrusion. This vertical motion may drive interfacial mixing and is an important mechanism to reduce the strength of stratification.
    - To further clarify, we have edited lines 252-258:

      "However, the horizontal density gradient introduced by the characteristic wedge shape of seawater intrusion will drive vertical baroclinic convective motion to flatten isopycnals. Such baroclinic adjustment may be an important source of interfacial mixing, working in tandem with turbulence and double-diffusive convection to reduce stratification within the subglacial environment. This convective-driven mixing mechanism differs from convective mixing caused by a sloping ice boundary, in which a buoyant plume may form. For the idealized scenarios in this study, buoyant convection via ice geometry will not drive mixing and thus melt since the ice is perfectly horizontal."

12. Line 261 – again, you can't attribute this to increased stratification (which MAY decrease shear/drag) because you haven't isolated the effect of the no-slip BC (which WILL decrease shear/drag)
    - We appreciate this insight. Similar to comment 10, we will adjust any comparison in near-wall kinematics to be between a free-slip boundary and the melt-enabled cases.

13. Line 286 – Diffusive convective melting also involves convection driven by cooling and is different from diffusive melting. See Martin & Kauffman (1977) section 3 for diffusive melting and Martin & Kauffman (1977) section 4 for diffusive-convective melting.
    - We appreciate this insight. The results comparing melt rate parameterizations will be reworked in light of major comment 7. With this, we will clarify our intent to describe diffusive-convective melting and not diffusive melting.

14. Fig 3 – Is any averaging (time or space) done to obtain these profiles?
    - Yes, these are vertical profiles from the time-averaged domain. In order to clarify, we have edited the caption:

        "Figure 3. Time-averaged vertical profiles of temperature(A, D), salinity(B, E), and x-component of velocity (C, F) along the seawater intrusion for uf =0.5 cm/s and medium turbulence Cμ = 0.09. The distance beyond the grounding line represents the distance (m) upstream of the fixed grounding line. The top row (A, B, C) is for the non-melt enabled case, i.e. the horizontal ice boundary is a wall boundary with a fixed temperature. The bottom row (D, E, F) is for the melt-enabled case where the ice boundary becomes a velocity inlet with freshwater inflow as a function of near-wall temperature."

        And figure discussion at line 200:

        "The time-averaged vertical profiles of temperature, salinity, and velocity along the intrusion for non-melt-enabled cases (Figure 3) depict a two- layered flow in opposing directions, with a relatively uniform low-sloping vertical gradient in salinity, and a strong thermocline in the 2 cm directly below the ice."

15. Fig 3 - It would be great to see the systematic change in the intrusion as u_f is increased with a series of side-by-side plots, rather than having to move back and forward from figs 3 to A8.
    - We will take this into consideration and present the vertical profiles for the other freshwater velocity cases within the main text with results from the new simulations.

16. Line 319 – I would not say that the flow is weak at 5,10 cm/s. However, the height of the channel is very small, so the Reynolds number will be small. Again, Re or other turbulence metrics are needed.
    - All cases tested here have low Reynolds numbers ranging from 25 to 2500. We have added this to line 84:

        "The corresponding Reynolds numbers for the given geometry and freshwater flux are 25, 250, and 2500 for the low, medium, and fastest freshwater cases presented."

    - In addition, the earlier suggestion to discuss turbulent viscosity metrics has been illuminating and we will use this variable to describe the energetics within the flow field for the new simulations.

17. Line 397 – how is the reduced gravity calculated here? Based on the density difference between the freshwater and saltwater?

   - The reduced gravity is calculated based on the density difference between pure freshwater and pure saltwater. However, a point can be made that the reduced gravity should change along the intrusion and amongst the freshwater cases based on the observation made in major comment 3. In the discussion, we will explore the sensitivity of the length scale proposed by Robel et al. 2022 to variations in reduced gravity.
   - To improve clarity, we have edited line 399:

      "Calculating the drag coefficient using model output gives Cd with values of order $10^{-2}$ to $100$. The analytical theory of intrusion distance ($L$) for an unobstructed water sheet from Robel et al. (2022) is,

      $$L = \frac{H^2 g'}{4 C_d^2 u_f^2}$$

      where $H$ = 0.05 m is the height of the subglacial environment, $g'$ = 0.20 m/s$^2$, and Cd is set to the maximum value within the intrusion. Reduced gravity is referenced to the density difference between the prescribed pure freshwater and pure seawater."

18. Line 427 – "If we anticipate viscous effects to dominate in seawater intrusions under grounded ice, then using the thermal and haline molecular diffusivities as so-called "transport velocities" would be appropriate, similar to the diffusive-convective framework presented above" – The units (m^2/s vs m/s) are not consistent. To turn the molecular diffusivity into a transport velocity, you need a lengthscale, i.e. the width of the diffusive sublayer. That's the hard part which is not addressed here!

   - We appreciate the reviewer's insight on this. With the model used here, we have a high enough domain resolution to identify the width of the diffusive sublayer. We can define this width to be where the turbulent viscosity is equal to the molecular viscosity. This works because the diffusive sublayer is where molecular diffusivity dominates. Therefore, there should be a point of transition where turbulent viscosity becomes weaker than molecular viscosity. This transition point (i.e. where they are equivalent) will represent the diffusive sublayer's thickness. We will present these results in tandem with the discussion of appropriate transfer mechanisms on line 427, highlighting the limitation of applying this method to large-scale coupled models.

   - We have edited line 427:

      "If we anticipate viscous effects to dominate in seawater intrusions under grounded ice, then using the thermal and haline molecular

diffusivities as so-called "transport velocities" would be appropriate. However, this would require knowing the width of the diffusive sublayer which, given computational constraints for coupled ice-ocean models, cannot be resolved."

19. Figure 6 caption – The Stanton number should be $\gamma T /U$, not $\gamma T /Cd$
    o Fixed.

20. Figure 6 caption last line – are both the dashed and two solid lines from Washam et al 2023?
    o Both the dashed and solid lines are reported in Table 1 in Washam et al. 2023. However, the dashed line is reported first in Washam et al. (2020) and is cited as such in Table 1 of Washam et al. (2023).

21. Line 490 – "bolstering the idea that grounding zones are subglacial estuaries (Horgan et al., 2013).
    o No change is indicated in the comment.

22. Line 566 – seems like a note-to-self.
    o We appreciate you catching this. It has been removed.

23. Line 594 – either "given by (3)" or "given by equation 3"
    o Fixed.

24. The appendix is quite bloated. I think some of the figures could be relegated to SI and some should go to the main text. For example, I think Fig A5 is a key result and should go in the main text. Fig A1 could go in SI.
    o Based on recommendations from both reviewers, we will shorten the appendix to contain only descriptions of the model (equations, meshing, run-time settings). We will transfer the discussion of the drag coefficient to the main body under the results section. Sensitivity tests and model evaluation will go in Supplementary material.

25. Table A1 – inconsistent unit formatting (italics/roman). Look to TC style guide, or use roman which is typical. Units for theta should just be degrees.
    o We have made all units in Roman format and followed the TC style guide in handling denominator values.

26. Table A2 – again, unit formatting. H [cm]
    o We have made all the units in Roman format, following the TC style guide.

27. Figure A5 – the colours are too hard to tell apart.
    o We will refine this figure and include it in the main text.

28. Figures A9 & A10. The figures are labelled "law of the wall" but no interpretation is offered. What is the black vertical line and what does it mean? What would the profiles be expected to look like if a log layer was present? What portion of the flow is being shown? Is y+=0 at the top or bottom of the domain? Do we see a viscous boundary layer, i.e. u+=y+? in addition, it would be much more helpful if y+ was on the y axis, since that's how the model is set up and how all your other profile plots are oriented.

- We have combined A9 and A10 into one figure with the following caption: "Law of the wall for standard geometry cases with medium turbulence ($C_\mu$ = 0.09) for all velocities (Panels A-C). Law of the wall for the alternate geometry cases with medium turbulence($C_\mu$ = 0.09) and freshwater velocity u_f = 0.5 cm/s (Panels C and D). In these figures, the vertical profiles are taken over the top half of the domain (2.5 cm from the ice face), to avoid effects from the intrusion interface. Therefore, y+ = 0 is at the top of the domain, where the ice boundary exists. The black line here represents y+ = 30, which would represent the point at which the log-law region of flow would develop and where u+ =y+. This point represents the transition to the fully turbulent outer boundary layer. Note how none of the profiles presented here cross that point, and therefore do not have a fully developed boundary layer."

- This figure will move to supplementary information and we will add a characteristic log law profile to demonstrate what the flow would look like for a fully developed boundary layer.

29. Reference 1 (Adusumilli) seems incomplete
- Fixed.

Works Cited

Pope, S.B. (2000). Turbulent Flows. *Cambridge University Press*, Cambridge, 305-308.

Lai, C.C.K., Socolofsky, S.A. (2019). Budgets of turbulent kinetic energy, Reynolds stresses, and dissipation in a turbulent round jet discharged into a stagnant ambient. *Environ Fluid Mech* **19**, 349–377. https://doi.org/10.1007/s10652-018-9627-3

Montagna, P. A., Palmer, T. A., & Beseres Pollack, J. (2013). *Hydrological changes and estuarine dynamics* (1st ed. 2013.). Springer. https://doi.org/10.1007/978-1-4614-5833-3

Washam, P., Nicholls, K. W., Münchow, A., and Padman, L. (2020). Tidal Modulation of Buoyant Flow and Basal Melt Beneath Petermann Gletscher Ice Shelf, Greenland, J. Geophys. Res.-Oceans, 125, https://doi.org/10.1029/2020JC016427

Washam, P., Lawrence, J. D., Stevens, C. L., Hulbe, C. L., Horgan, H. J., Robinson, N. J., Stewart, C. L., Spears, A., Quartini, E., Hurwitz, B., Meister, M. R., Mullen, A. D., Dichek, D. J., Bryson, F., and Schmidt, B. E. (2023). Direct observations of melting, freezing, and ocean circulation in an ice shelf basal crevasse, Sci. Adv., 9, DOI:10.1126/sciadv.adi7638

---

## Author Comment (AC2)

Response to Reviewer #1 for Manuscript "High-Fidelity Modeling of Turbulent Mixing and Basal Melting in Seawater Intrusion Under Grounded Ice" by Mamer, Robel, Lai, Wilson, and Washam

**General comments:**
The authors aim to tackle a fascinating problem of practical importance. Melting near grounding lines is thought to have much more impact on glacier dynamics than melting anywhere else, yet observations are limited, and high-fidelity simulations are lacking. Thus, the work is highly novel and has significant potential for improving our understanding of ice-ocean interactions that have a high impact on climate dynamics. However, I am not sure that the ANSYS Fluent RANS solver used is appropriate for this problem. In fact, I have never seen the ANSYS Fluent RANS solver applied to environmental flows as complex as in this work. This does not mean that it cannot work, but that significant effort should be devoted to validating the code. As the governing equations solved by the codes are not clearly presented or discussed (in particular, the boundary conditions, which are yet key to evaluating the melting dynamics), I have found it difficult to assess the appropriateness of the mathematical/numerical model. I am particularly concerned with the modelling of the ice-ocean boundary: there are no salt constraints and melt is said to be activated in a way that I did not find correctly physically motivated. In the unfortunate case that the code cannot in fact solve the exact problem at hand (with temperature and salt stratification coupled through a phase change boundary) I would suggest that the authors reformulate the problem as a list of hypotheses--inspired from the full problem--and testable with simulations of a modified/simpler model (but mathematically transparant), which could be the reduced thermal driving model discussed below (point 2).

We thank the reviewer for their thorough comments on this manuscript. We have identified this reviewer's primary concerns to be:
1. The lack of model description and presentation of main equations used by ANSYS Fluent. Along with this, the reviewer also identified the lack of citations supporting the choice of ANSYS Fluent (e.g. validation studies).
2. The lack of a salinity and pressure-dependent thermal boundary condition for the ice boundary.
3. Overall issues with clarity in describing model equations and model initialization.

To address these concerns, we have revised our appendix to present the full model framework, including governing equations, domain and meshing, and validation studies. In addition, we will re-run each of our simulations with a salinity and pressure-dependent ice thermal boundary condition.

We address each comment below. The reviewer's comments are written in black, our responses are given in red, and the original manuscript text is in grey.

**First Comment:**

The governing equations that the code solves should be clearly presented, as I do not think that TC readers are familiar with RANS models (and because I have never seen RANS models used in this context). This requires presenting the Reynolds decomposition of the flow (which would help you justify the fact that 2D dynamics is expected since turbulence is not resolved) and the governing equations for the ensemble-average variables. The closure for the Reynolds stresses and turbulent fluxes should then be discussed in greater details than in the current manuscript. The kappa-eps scheme is mentioned (Eq. (2)) but important details are lacking: for instance, how are kappa and epsilon related to the resolved variables?

We have added a subsection to the appendix describing the governing equations of the model and further discussed the turbulence closure scheme. We have revised appendix section A4 to have these subsections:

A 4.1 Reynolds Averaged Navier Stokes Equations

"ANSYS Fluent solves the Reynolds Averaged Navier Stokes (RANS) equations. This formulation of the Navier Stokes equations decomposes the flow into a mean state and fluctuation about the mean:

$$u_i = \overline{u_i} + u_i' \tag{R1}$$

where $\overline{u_i}$ is the time-avearged velocity and $u_i'$ is perturbations about the mean velocity. Scalar quantities are also decomposed into their mean and fluctuations about the mean. Substituting these decomposed values into the momentum equations yields:

$$\frac{\partial \rho}{\partial t} + \frac{\partial}{\partial x_i}(\rho u_i) = 0 \tag{R2}$$

$$\frac{\partial}{\partial t}(\rho u_i) + \frac{\partial}{\partial x_j}(\rho u_i u_j) = -\frac{\partial p}{\partial x_i} + \frac{\partial}{\partial x_j}\left[\mu\left(\frac{\partial u_i}{\partial x_j} + \frac{\partial u_j}{\partial x_i}\right)\right] + \frac{\partial}{\partial x_j}\left(-\rho\overline{u'_i u'_j}\right) - \frac{\Delta\rho g}{\rho\epsilon^3} \tag{R3}$$

Equations R2 and R3 are the RANS equations and have the same general form as the Navier-Stokes equations, however, the velocities and solution variables now represent the time-averaged values. In addition, new terms have been added to incorporate the effect of turbulence. These terms are the Reynolds stresses $(\rho\overline{u'_i u'_j})$. To close this system of equations, the Reynolds stresses must be solved via a turbulence closure. More details on turbulence modeling and our closure choice are given in section A 4.2."

I would like also to see more details on the so-called damping functions of the low-Re formulation that supposedly enable accurate diffusive boundary layer representation. Is it like in a wall-resolved large-eddy simulation model?

The standard $k$-$\epsilon$ turbulence model relates eddy viscosity to turbulent kinetic energy and turbulent dissipation to close the system of equations. This requires solving the transport equations for turbulent kinetic energy and dissipation. Turbulent kinetic energy vanishes near the wall, producing a singularity in these transport equations. To fix this, a modification must be made to the timescale set by the ratio of turbulent kinetic energy and dissipation. This is where low-Reynolds formulations of the $k$-$\epsilon$ model are helpful, which resolves the transport of turbulent kinetic energy and turbulent dissipation in the low-Reynolds number regions of flow (i.e near no-slip boundaries).

In large eddy simulations (LES), only large eddies of a system are resolved, with small-scale (typically sub-grid) eddies being filtered out. For wall-resolved large-eddy simulations, the boundary layer is directly resolved by having a very fine mesh to resolve the large gradients near the wall. The damping functions and adjustments made to the $k$-$\epsilon$ model in order to resolve the boundary layer are similar to wall-resolved LES since they both require a finer mesh and do not rely on wall functions to parameterize the near-boundary gradients. However, because of the averaging done to obtain the RANS equations, a singularity is introduced in the turbulent dissipation equation via the turbulence timescale. The low-Reynolds formulation of the $k$-$\epsilon$ model fixes this (as explained and edited in minor comment 6) and uses the damping function to ensure a smooth transition between the freestream flow and the boundary flow.

We have added a description of one version of a low-Reynolds formulation to the appendix (minor comment 6). Other formulations of low-Reynolds models are also available within ANSYS Fluent, and a sensitivity analysis will be done to determine the most appropriate version. Within the appendix, we will include an explanation of the exact version we use, alongside a description of how it works.

**Second Comment:**

The decoupling of the salt dynamics from the ice-ocean boundary dynamics is not justified and a priori seems wrong. I assume that this decoupling is due to code limitations. However, if you cannot justify the decoupling, I am afraid this just means that the code is not suited for environmental flows with temperature and salt stratification and melting. In the worst case scenario you might consider reformulating the problem in terms of a single scalar variable, namely thermal driving (ref: Adrian Jenkins' papers and other people's related works). The collapse of the full dynamics onto a reduced thermal driving model is thought to be accurate in highly-turbulent environments (for which kappa-epsilon applies anyway) and small salinity variations. This latter condition is obviously problematic (which should be discussed) with regards to your problem of interest. However, I would rather see thermal driving model simulations transparently solved by ANSYS than results from a non-transparant full temperature-salinity model.

In the original manuscript, the only way salt is not considered is within the melting dynamics. We did not model phase change (regardless of whether this is melting or dissolution) in the simulations. Dissolution-driven melting (where salt would get into the crystal matrix, suppress the freezing point to below the ice temperature, and therefore cause it to melt) would only dominate ice loss if the seawater is subfreezing (below the local pressure and salinity dependent freezing point) and therefore mass transfer into the ice matrix would exceed heat transfer.

In the original simulations, we set the boundary temperature to $0°C$. However, based on suggestions from both reviewers, we will re-configure the simulations to have a pressure and salinity dependent thermal boundary condition for the ice boundaries using equation R5. In Figure R1, we demonstrate what this thermal boundary condition would look like for a free-slip case in a simplified pipe flow example.

We have added to line 573:

> "Salt is transported as an active tracer within the fluid, employing an advection-diffusion-reaction equation:

$$\frac{\partial}{\partial t}(\rho S) + \nabla \cdot (\rho \vec{u} S) = -\nabla \cdot \vec{J} + R_i \tag{R4}$$

> Where $\vec{u}$ is the velocity vector, $\vec{J}$ is diffusion flux, and $R_i$ is the production rate from reactions. The production rate for salt in these simulations is zero."

Further edits regarding salinity sinks and dissolution-driven melting are given in major comment 3.

[Figure]

**Figure R1**

Example run of a 2D pipe-like geometry with ice at y = 0.05 m and a velocity inlet prescribed at x = 1m. This case has a free-slip condition at the ice base (along y =0.05) as well as a salinity and pressure dependent thermal boundary condition (here, we set the ice thickness to be 1000 m). Panel A is temperature, panel B is density, and panel C is velocity magnitude. In panel C, a classical boundary layer forms along the flow direction at the bottom boundary where there is a no-slip condition. In panel A and B, a thin thermal boundary layer forms due to the salinity and pressure dependent thermal boundary condition cooling the near-ice water.

**Third Comment**

The boundary conditions, especially at the ice-ocean interface, should be clearly presented and discussed.

We have elaborated on the boundary conditions (thermal and kinematic) in both the main text and the appendix where the model formulation is discussed.

We specifically edited lines 87 - 88:

"The ice wall boundaries have a pressure and salinity-dependent thermal boundary condition of:

$$T_b = S_b \lambda_1 + \lambda_2 + z \lambda_3 \tag{R5}$$

Here, $\lambda_1$, $\lambda_2$, and $\lambda_3$ are constants, and the boundary salinity is $S_b$. The depth of the ice is equal to $z_b$, in these simulations we set this to be 1000 m. Both the vertical

and horizontal ice boundaries have a no-slip kinematic condition in the non-melting cases, forcing the freestream fluid velocity to be zero at the ice wall."

In addition, we have added to line 147:

"Therefore, the melting cases have a free slip kinematic boundary condition. The temperature of the inflowing meltwater is set by equation R5."

Where equation R5 is the pressure and salinity-dependent freezing point equation.

We have also edited lines 519 - 522 in the appendix:

"For both melting and non-melting simulations, the ice walls have a thermal boundary condition dependent on near-wall salinity and pressure (eq. R5). Neither ice wall boundary allows for salinity diffusion, which would be another mechanism of melt to account for. The non-melting simulations employ a no-slip kinematic condition, which forces the fluid velocity to be zero at the wall. For melt-enabled cases, the top boundary of the subglacial space is turned into a velocity inlet to simulate melt. In designating this boundary as a velocity inlet, a free-slip kinematic condition is required. The downward vertical velocity set at the melting ice boundary follows equation 4. We turn off melting for the vertical ice boundary to isolate the intrusion-induced melt from the vertical plume dynamics that would arise from the vertical ice boundary."

The physical motivation for the so-called melt-activated formulation is lacking. Melting produces buoyant flows even when the boundary is no slip, simply because melting acts like a sink for salinity (though this is lacking in your model).

Since salt transport is included in the model (i.e. salt is transported as an active tracer) and we are not considering dissolution-driven melting, there does not need to be a sink for salinity in the melting framework. The model solves for the displacement and movement of 'saltier' waters via the salt transport equation (equation R4). Mass is conserved via a pressure outlet (zero gradient flux boundary condition) at the top of the ocean domain.

We have edited lines 104 - 105 to improve clarity:

"Salt is therefore transported as an active tracer within the fluid domain. To ensure mass transport within the computational domain is realistic and physical, there are two velocity (hence mass) inlets (i.e the subglacial discharge and ocean inflow) in the non-melting case and three velocity inlets (subglacial discharge, the melting horizontal ice face, and the ocean inflow) in the melting case. A pressure outlet is defined at the upper portion of the ocean domain, meaning the mass outflow rate along this boundary is not specified and is determined as part of the numerical

solution based on the requirement that all flow variables have zero gradients in the direction normal to the boundary. This kind of arrangement is typically used to emulate fluid flows in an infinite domain as in our case where subglacial channel discharge is released at the ground line into the ocean with infinite extent. "

Thus, should we envision your melt-activated formulation like a velocity compensation for the lack of salt sink in the model? If so, it was not clear to me whether the velocity is prescribed vertically or horizontally, and I am not sure it is the correct way to compensate the salt sink.

Since we are neglecting melting from dissolution, due to above-freezing fluid conditions, a salt-sink at this melting boundary is not needed. These simulations strictly focus on the heat-driven melting and associated buoyancy effects from added fresh water to the model domain.

The meltwater velocity is the rate at which buoyant freshwater is input into the domain to mimic heat-driven melting. The velocity is prescribed normal to the boundary where it is sourced, in this case vertical.

We have edited lines 143 - 144:

"In some simulations, we also simulate the added buoyancy flux resulting from the heat-limited melting scenario. Here, we neglect melting driven by dissolution, instead focusing on melting driven by thermal equilibrium at the ice boundary. Since the thermohaline conditions of the fluid domain are non-sub-freezing, the neglect of dissolution-induced melting is justified."

We have also edited line 150:

"The downward fluid velocity prescribed at the horizontal ice face is set by the melt rate..."

Estimating the velocity to enforce at the boundary to mimic the salinity sink from Eq. (4) is also not justified, i.e. why should the movement of the interface (assuming there is no immediate hydrostatic equilibrium of the ice shelf at such small scales) be directly used as a buoyancy-driven velocity input?

We acknowledge that adding a vertical velocity to mimic the vertical movement of a solid is a bit crude, however, because this vertical velocity (the melt rate) is small relative to the main flow, its direct contribution to the momentum flux is small or negligible. The buoyancy it brings to the fluid domain is significant, however, and is therefore an integral part of the heat-driven melting process. In addition, the vertical velocity that injects meltwater into the domain is not constant across the horizontal ice face but rather varies according to the underlying thermal equilibrium prescribed by equation 4.

We have added to line 160:

> "In setting a vertical velocity to mimic a moving interface, we introduce additional sources of momentum to the fluid. The vertical velocity arising from the meltwater inlet is small relative to the main flow and is therefore negligible, but the buoyancy that the meltwater brings into the fluid domain is an integral part of the heat-driven melting process."

We have also edited lines 154 - 156:

> "This framework represents the conservation of heat at the ice-ocean interface, which varies along the horizontal ice face as the near-wall thermal gradient changes due to seawater intrusion and vertical mixing."

**Fourth Comment**

In would like to see a validation of the code, which includes the choice of turbulence closure. A simple benchmark case should be set-up, for which turbulence-resolving simulation data exist (either from direct numerical simulation (DNS) or large-eddy simulation (LES)). Several groups (with Catherine Vreugdenhil, John Taylor, Ken Zhao etc) have published such DNS and/or LES data over the past 5 years or so, making it practical. Typical configurations are channel flow configurations, which should be accessible to ANSYS. Validation could be based on quantitative comparisons of mean variable profiles (e.g. temperature, TKE) normal to the ice-ocean interface.

ANSYS Fluent was chosen to address this research question because of its extensive validation and history of practical use within the engineering fluid mechanics community.

We have added to line 63:

> "ANSYS Fluent has been extensively validated across diverse flow geometries and conditions. Zangiabadi et al. (2015) validated ANSYS Fluent's suitability in simulating flow structures around realistic bathymetric highs in a study evaluating coastal tidal turbine deployment. Here, they found the RANS $k$-$\epsilon$ method to be more precise than the large eddy simulation (Zangiabadi et al., 2015). Chan et al. (2020) demonstrated good agreement between ANSYS Fluent's simulation of multi-phase sediment-laden plumes and prior experiments by Hall et al. (2010) and Virdung and Ramuson (2007), using the $k$-$\epsilon$ turbulence closure. Similarly, Sultan et al. (2018) achieved results consistent with experimental data for both multi-phase and single-phase flow in pipe geometries. Al-Zubaidy and Hilo (2022) found strong agreement between their model of lateral intakes for engineered flow channels with field measurements."

We believe these existing studies validate a wide range of cases similar to our setup, and any additional validation would be outside the scope of this paper.

**Fifth Comment**

Successful applications of ANSYS Fluent RANS solver to environmental flow configurations with temperature and salinity stratification should be cited and discussed (in particular how they validated the code).

We appreciate the reviewer highlighting the lack of justification for choosing ANSYS Fluent within the paper. We have incorporated examples of validation and applications of ANSYS Fluent to stratified environmental flow scenarios.

We have added to lines 63 - 64:

> "Most relevant to this study, Chala et al. (2024) validated Ansys Fluent's capability to model seawater intrusion in porous aquifers, achieving close alignment between experimental data and model predictions for intrusion length and shape. Furthermore, ANSYS Fluent has recently been used to model freeze desalination processes, where the volume of fluid method is applied to a species mixture of salt and water with a cooling base. Jayakody et al. (2017) demonstrated ANSYS Fluent's applicability to conducting parametric studies for freeze desalination processes by validating it with experimental studies."

**Sixth Comment**

The discussion of the steady state and transient runs is really confusing. You should distinguish the existence (or non-existence) of a steady state from your strategy of successive runs to achieve it. The key point that should be in the main text is that the problem has a natural steady state (for the ensemble-average variables) as there is no external variability and the ensemble-average variables do not exhibit temporal fluctuations once equilibrated. The strategy to reach it should then be discussed in an appendix.

The purpose of the steady-state solver is to generate an initial condition that is reminiscent of our expected solution since Fluent's steady-state solver is sensitive to initial conditions. We then ran the transient solver because the initial steady-state solver did not reach the seawater's full distance of intrusion. At the end of the transient run, a final steady-state solver run was conducted with the final transient run as the initial condition. This final step was to ensure the transient intrusion distances were not a product of numerical noise. Figure A1 in the appendix compares the time-averaged intrusion distances from the transient runs to the final steady-state run.

We have edited lines 166-169 in the main body:

"Each simulation is initialized with the steady-state solver employed by ANSYS Fluent. This step to initialize the domain is necessary because the model is sensitive to initial conditions and it cuts down on the total run time needed to achieve a quasi-steady-state. The transient solver is then run for 12 hrs at 5 s time steps. A quasi-steady-state is reached at 6000 s. All results presented are from the transient simulations and a comparison to the steady-state solver solutions is shown in section A4 (Figure A1). Note that the steady-state solver is different than the quasi-steady-state achieved in the transient runs. The steady-state solver employed by ANSYS Fluent drops the time derivative, while the quasi-steady-state achieved during the transient run is a stable intrusion state. A list of all simulations is presented in section A4 (Table A2)."

As well as lines 574 to 582 in the appendix:

"The results presented in the main body of this work are the time-averaged results from the transient run of 12 hrs. A series of tests were conducted to evaluate ANSYS Fluent's steady-state solver to the transient solver and their corresponding solutions. ANSYS Fluent's steady state solver drops the time derivative from the RANS equations and is sensitive to initial conditions, which is different than achieving a quasi-steady-state. The domain is initialized with a saline, warm ocean tank, and a fresh, cold subglacial environment with no intrusion. We run the steady-state solver with this initialization (named steady state pre), and in most cases, a small intrusion develops and occasionally no intrusion. We then run the transient solver for 12 hrs at 5s time steps where a quasi-steady-state intrusion develops. A secondary run is conducted with the steady-state solver, using the quasi-steady-state transient solution as the initial condition (named steady-state post). In the post-steady-state run, the intrusion developed during the transient simulation persists. The comparison of results between the pre and post steady-state solver runs demonstrates the steady-state solver's sensitivity to initial conditions. We thus disregard the steady state pre-results. To compare the steady state post and transient solutions, we plot the intrusion distances against each other, in which they nearly collapse on a 1 to 1 line (Figure A1). This comparison reinforces the development of a quasi-steady-state in the 12 hr transient runs."

**Seventh Comment**

I have found many typos in the appendices ("Need to list reference values, materials info, the methods and controls" found line 566), suggesting that these were not carefully reviewed by the authors. At the moment the appendices seem primarily like a draft list of comments and figures that did not make it into the main text (some in fact repeating what is already in the main text).

A careful review of the appendix will be conducted. In addition, we have added further descriptions of the governing equations (major comment 1, minor comments 4, 5, 6 and 7),

model boundary conditions (major comment 3), the use of transient and steady-state solver (major comment 6), and analysis of turbulence parameters. We have moved sensitivity tests and extra simulations to the supplementary material to make the appendix more concise.

**Eighth Comment**

The figures are lisible but could be improved(e.g. the x-axis label of Fig. 6 is quite small).

We will standardize the figures' text sizes and shapes for all figure axes and titles.

**Ninth Comment**

Because of the many questions I had/have with respect to the simulation code, I was not able to appreciate the comparison of the simulation results with the parameterized predictions of melt rates. If the authors can validate their code, I agree that this comparison would be an important addition to the paper, but it would have to be a fair comparison. That is, if the authors end up solving a model that is distinct from the model that the parameterizations (necessarily approximate) aim to mimic (arguably the real exact model), they should discuss result differences in light of model differences.

We understand the concerns of the simulation code to be: salinity sinks, ice-ocean boundary temperature independence of salinity, and vertical velocity as a proxy for a moving ice interface.

We reiterate that salt transport is included in these simulations and the only way it is not represented in the melting framework is by having a suppressed boundary temperature.We can safely neglect dissolution-driven melting due to the above-freezing fluid conditions. Including the depressed freezing point would only increase our simulated melt rates, by increasing the thermal driving (i.e., the current simulated melt rates are a lower bound). Thus, because the parameterized melt rates are already lower than the simulated melt rates, including a boundary temperature dependent on near-ice salinity would further increase the disagreement between simulated melt rates and parameterized melt rates. Our discussion of the simulation-parameterization disagreement attempts to highlight the differences in light of the assumptions of the fluid-structure that are inherent in the parameterization. i.e. well-mixed, higher Reynolds number, etc.

However, considering comments from both reviewers, we will re-run each simulation with a thermal boundary condition that is dependent on near-wall salinity and a reference glaciostatic pressure (i.e. at 1000 m). A proof-of-concept model run is demonstrated in Figure R1 for a simplified pipe flow geometry.

Since salt transport is represented in the model by its advection-diffusion equation, a sink does not need to be provided where meltwater enters the domain. If we consider melting

from dissolution processes, the ice boundary would need to remove salt. However, because the fluid domain simulated here is non-sub-freezing, heat-driven melting will dominate and therefore the neglect of diffusion in the model presented here is justified.

We acknowledge inputting a vertical velocity as a proxy for a moving ice interface is crude, however, because this vertical velocity is O(-6) and the freestream flow is O(-1) – O(-3) we believe this choice is justified.

**Tenth Comment**

Simulation snapshots and movies would really help visualize the flow.

In the supplementary material, we will include simulation snapshots for the initial steady-state runs (i.e. the initial condition) and movies of the transient runs with updated simulations. These will ideally aid Figure 1 from the original manuscript in describing what the flow regime looks like.

**Additional comments, including technical corrections: typing errors, etc.**

1.  line 29: could you clarify the idea of "tidally asymmetric"?

    o   Added to line 29:

        "Such asymmetry results in stronger melting during the ascent of high tide and weaker melting during the tidal ebb."

2.  line 87: writing "The ice wall boundaries have a temperature boundary condition of 0∘C" really felt like a bomb! And the lack of justification or description of the full mathematical model did not help disarm it. It is really not expected that prescribing 0∘C at the ice-water interface is reasonable, especially when the salinity goes from 0 to 30 psu.

    o   Based on suggestions from both reviewers, we will re-run all simulations with the pressure and salinity dependent freezing point boundary condition. Figure 1 above demonstrates how this will be done.

    o   We have edited the manuscript at lines 87-88, 147, and 519-522 as written in major comment 3.

3.  line 108: this equation of state may be suited for small salinity variations, but with a range of 0 to 30 psu, it may not be appropriate, unless the grounding line is beneath a lot of ice. I would recommend that you use a higher-order approximation of the true equation of state, or at least acknowledge that you know of the anomaly of the equation of state at low salinity and pressure but that you discard its effects to simplify the problem. Ideally you could discuss/speculate on how the results might change should you consider an accurate equation of state.

    o   We appreciate you bringing this concern to our attention. We will run a sensitivity test comparing our linear E.O.S. to the higher order approximation in Roquet et al. (2015):

$$\rho' = -\frac{C_b}{2}(\Theta - \Theta_0)^2 - T_h Z\Theta + b_0 S_A \qquad \text{(R6)}$$

Where:

$$C_b = 0.011 \ kg/m^3 K^2$$

$$T_h = 2.5 \ x \ 10^{-5} kg/m^4 K$$

$$b_0 = 0.77 \ kg/m^3 (kg/g)$$

$$\Theta_0 = -4.5°C$$

The results of this sensitivity test will be discussed in the appendix.

4. line 113: conduction, diffusion, and molecular dissipation all sound like the same thing to me. Could you explain how they differ? (reading your appendices it looks like conduction might be turbulent conduction)

   o Conduction represents the transfer of heat due to the thermal gradient. Within ANSYS Fluent, the conductive value used is the 'effective conductivity' which is the summation of the thermal conductivity of the fluid and turbulent conductivity. The diffusive term represents heat transfer due to species diffusion – in this case, salt. In equation A7, this term is the sum of every species' diffusive flux multiplied by their enthalpy. Molecular dissipation is another way of phrasing viscous dissipation, which is the heat transfer due to viscous forces, represented by the effective shear stress and kinematic viscosity.

   o Edited line 113 :

      "Energy, and therefore fluid temperature, is evolved via an energy conservation equation employed by the CFD solver resolving advection, conduction, salt diffusion, and viscous dissipation molecular dissipation. (ANSYS Inc., 2024). Conduction represents heat transfer due to thermal gradients, and viscous dissipation is the transformation of kinetic energy into thermal energy due to shear forces. As salt diffuses in the medium, it also transfers heat due to its unique thermal properties, and therefore must also be included."

5. line 124: the quadratic quantity indicated is one among many, such that the sentence does not read well. Please reformulate.

   o To clarify, we rewrote the quadratic quantity to have i,j notation.

   o Edited lines 123 - 127 are shown below.

6. line 126: could you recall what the Boussinesq hypothesis is?

   o Edited lines 123-127:

      "A closure scheme is necessary because averaging the RANS equations introduces Reynolds stresses due to turbulent motion within the fluid. Reynold's stresses take the form $\overline{u_i' u_j'}$ , the averaged product of turbulent velocity fluctuations. One class of closure models employs the turbulent-viscosity hypothesis, which relates the deviatoric Reynolds Stresses to the mean strain rate via a positive scalar eddy viscosity (Pope, 2000). Here, we utilize the two-equation $k$-$\epsilon$ closure scheme, which solves for the eddy viscosity by:

$$\mu_T = \rho_w C_\mu \, k^2/\epsilon \qquad\qquad\qquad (R7) \; "$$

o Added to Section A4, starting at line 551:

"The turbulent-viscosity hypothesis (also known as the Boussinesq Hypothesis) is used in turbulence modeling to solve for the Reynolds stresses. This hypothesis states the deviatoric Reynolds stresses (those deviating from the mean) are proportional to the mean strain rate tensor by a positive scalar. This scalar represents the eddy viscosity (also referred to as turbulent viscosity). This relationship is:

$$-\rho \overline{u_i' u_j'} + \frac{2}{3}\,\rho k\,\delta_{ij} = \rho \mu_T \left(\frac{\partial \overline{U_i}}{\partial x_j} + \frac{\partial \overline{U_j}}{\partial x_i}\right) \qquad\qquad (R8)$$

The only unknown left in the system of equations is the eddy viscosity, which can be solved for by a variety of different turbulence closure schemes. Here, we employ the two-equation $k$-$\epsilon$ closure scheme which solves for eddy viscosity by relating it to the square of turbulent kinetic energy and inverse of turbulent dissipation by a positive scalar $C_\mu$. This closure scheme requires two additional equations to solve for turbulent kinetic energy and turbulent dissipation. These equations are:

$$\frac{\partial}{\partial t}(\rho k) + \frac{\partial}{\partial x_i}(\rho k \, u_i) = \frac{\partial}{\partial x_j}\left[\left(\mu + \frac{\mu_t}{\sigma_k}\right)\frac{\partial k}{\partial x_j}\right]$$

$$+ G_k + G_b - \rho\epsilon - Y_M + S_k \qquad (R9)$$

For turbulent kinetic energy and:

$$\frac{\partial}{\partial t}(\rho \epsilon) + \frac{\partial}{\partial x_i}(\rho \epsilon u_i) = \frac{\partial}{\partial x_j}\left[\left(\mu + \frac{\mu_t}{\sigma_k}\right)\frac{\partial \epsilon}{\partial x_j}\right]$$

$$+ C_{1\epsilon}\frac{\epsilon}{k}(G_k + C_{3\epsilon}G_b) - C_{2\epsilon}\rho\frac{\epsilon^2}{k} + S_\epsilon \quad (R10)$$

For turbulent dissipation. Note, in the equation for turbulent dissipation, turbulent kinetic energy is in the denominator which results in issues when $k$ approaches zero near wall boundaries. To resolve boundary layer dynamics with the $k$-$\epsilon$ closure, we use the low-Reynolds formulation which employs damping functions and fixes the singularity that arises with low values of $k$.

A version of the low-Reynolds $k$-$\epsilon$ closure we employ here is the Yang-Shih version (Yang and Shih 1993). In this formulation, the authors set the near-wall turbulence timescale to be the Kolmogorov timescale $(T_k \propto (\nu/\epsilon)^{1/2})$. In doing so, the equation for eddy viscosity near the wall becomes:

$$\mu_T = C_\mu\, f_\mu\, k\, \left(T_k + \frac{k}{\epsilon}\right) \tag{R11}$$

Where $f_\mu$ is the "damping function" and equal to:

$$f_\mu = \left[1 - exp\left(-a_1\, R_y - a_3\, R_y^3 - a_5\, R_y^5\right)\right]^{1/2} \tag{R12}$$

and $R_y = \frac{k^{\frac{1}{2}}y}{\nu}$. The constants $a_1$, $a_3$, and $a_5$ are constrained from DNS experiments for turbulent channel flow.

The final adjustment to the standard $k$-$\epsilon$ formulation for near-wall flows is to add an additional source of dissipation, which results from inhomogeneity in the mean flow field. This takes the form:

$$E = \nu\mu_T \frac{\partial U_i}{\partial x_j \partial x_k}\frac{\partial U_i}{\partial x_j \partial x_k} \tag{R13}$$

This formulation of the low-Reynolds $k$-$\epsilon$ turbulence closure allows for solving the free-stream portion of the flow regime as well as the near-wall region where viscous effects dominate, since the added terms tend to zero when turbulence is high."

7. Eq. (2) P2: what are the equations for kappa and epsilon? I expect that they involve the resolved ensemble-average variables.

   o We have added these equations to the appendix in the model description as written above in comment 6.

8. Section 2.4: this section is really confusing as I already mentioned earlier, because buoyancy isn't due to some interfacial velocities but to changes in salinity and temperature at the phase-change boundary. Please reformulate.

   o We have emphasized that any buoyancy effects are exclusively due to the difference in density between the input meltwater and the ambient water, and not due to some prescribed velocity.

   o We have added to lines 160-161:

     "In setting a vertical velocity to mimic a moving interface, we introduce additional sources of momentum to the fluid. However, the vertical velocity arising from the meltwater inlet is small relative to the main flow and is therefore negligible. The input of fresh, cold water due to melting introduces a buoyancy flux to the domain, due to density differences between the meltwater and intrusion."

9. line 150: I don't see the physical justification for prescribing a "horizontal ice (?!) velocity" at the horizontal ice-water interface.

   o A fluid velocity-inlet boundary condition is applied to the horizontal ice base when melting is enabled. The fluid velocity prescribed here is normal to the boundary and therefore is downward, not horizontal.

   o Edited line 150:

     "The downward fluid velocity prescribed at the horizontal ice face is set by the melt rate, $\dot{m}$, and is a function of the difference between the near-wall cel''s centroid temperature $T_w$ and the ice-ocean interfacial temperature $T_b$, thermal conductivity $k_T$, and density of ice $\rho_i$:..."

10. line 179: can you provide a reference for "realistic estuarine-like mixing rates"?

    o We appreciate the reviewer bringing the lack of citations to our attention. We have amended line 179 to:

      "Turbulent mixing, as modulated by $C_\mu$, affects intrusion distance to a lesser degree than freshwater discharge velocity when varied over a wide range encompassing likely values on the lower-end for realistic estuarine-like mixing rates (Geyer et al. 2000, Geyer et al. 2008)."

11. line 193: rewrite "retrograde bed slope".

    o We have done this.

12. You refer to figure 5 before figure 4 so the two should be swapped.

o   We have done this.

13. line 253: Could we say "vertical baroclinic convective motion"?

o   We appreciate this suggestion and have made this edit.

14. Eq. (9) line 325: this equation doesn't look like a parameterization but rather like the exact expression for the melt rate as a function of the conductive heat flux at the interface.

o   Based on both reviewers' suggestions, we have removed this section from the results.

15. Fig. 5-7: it was not clear to me whether the results you plotted were for the melt-enabled model or not.

o   In Figure 5, the dashed lines represent melt-enabled scenarios as denoted in the caption. To address confusion, we have edited line 248:

> "When melt is enabled the horizontal extent of stratification in the subglacial environment is reduced, but where stratification occurs, it is stronger (e.g. dashed lines in Figure 4)."

o   For Figure 6, all data comes from melt-enabled cases, as denoted in the caption. We have added to line 341 to improve clarity:

> "Here, we tested the sensitivity of equations 6-9 to various choices of ice distance to obtain $T_w$, $S_w$, $\rho_w$, and $u$ for the melt-enabled cases."

o   For Figure 7, both melt-enabled and no-melting cases are represented as denoted in the legend. We have edited the figure caption to improve clarity:

> "Wilson et al. (2020) experimental data (gray markers) and intrusion characteristics found in this study (red and blue markers). The red markers represent simulations with melting enabled, and blue markers represent non-melting simulations. The black dashed line is the numerical solution to Robel et al. (2022) with $\gamma = 2$."

In addition, we have added to lines 387-389:

> "Our simulated intrusions for both non-melting and melt-enabled scenarios follow the general trend and scale sensitivity to those identified in previous laboratory experiments (Figure. 7) (Wilson et al. 2020) which are within a factor of 10 to the theoretical prediction (dashed line) from Robel et al. (2022)."

16. Eq. (A3): should it be Re_L? Or change L into x?

o We appreciate you pointing out this inconsistency. We have amended the equation to be:

$$Re_L = \frac{uL\rho_w}{\mu} \tag{R14}$$

17. Appendix A4: I have found many typos, you need to use capital letters for Reynolds number, kappa has become k etc Eq (A7) has signs/symbols displaced. Please discuss Eq (A7) more carefully. What is tau_eff? What is species diffusion in your case?

    o In combination with previous comments (e.g. 3, 4,5, 6, 7, and major comments 1 and 5) we have revised the appendix to encapsulate every part of the model and more thoroughly justify model choices. In this rewrite, we standardized the symbols used and refined appendix equations.

    o $T_{eff}$ is the effective shear stress, which includes viscous shearing effects as well as shear from the no-slip boundary conditions.

    o Species Diffusion in our case represents salt diffusion. Since we are transporting salt as an active tracer throughout the fluid, we must include its heat transport in the energy equation.

    o We have edited lines 569-571:

> "The first three terms on the right-hand side represent energy transfer due to the conduction of heat ($\nabla \cdot (k_{eff} \nabla T)$), species diffusion ($\nabla \cdot (-\sum_j h_j J_j)$), and molecular dissipation ($\nabla \cdot (\tau_{eff} v)$), respectively. Conduction represents heat transfer due to thermal gradients, and viscous dissipation is the transformation of kinetic energy into thermal energy due to shear forces from viscous effects and wall boundary effects ($\tau_{eff}$). As salt diffuses in the medium, it also transfers heat due to its unique thermal properties, and therefore must also be included."

Works Cited

Pope, S.B. (2000). Turbulent Flows. *Cambridge University Press*, Cambridge, 305-308.

Geyer, W.R., Trowbridge, J. H., & Bowen, M.M. (2000) The Dynamics of a Partially Mixed Estuary. *J. Phys. Oceanogr*. **30**, Iss 8, 2035-2048. https://doi.org/10.1175/1520-0485(2000)030<2035:TDOAPM>2.0.CO;2

Geyer, W.R., Scully, M.E. & Ralston, D.K. (2008). Quantifying vertical mixing in estuaries. *Environ Fluid Mech* **8**, 495–509. https://doi.org/10.1007/s10652-008-9107-2

Roquet, F., Madex, G., Brodeau, L., Nycander, J. (2015). Defining a Simplified Yet "Realistic" Equation of State for Seawater. *J. Phys. Oceanogr*. **45**, Iss 10, 2564-2579. https://doi.org/10.1175/JPO-D-15-0080.1

Chalá, D.C.; Castro-Faccetti, C.; Quiñones-Bolaños, E.; Mehrvar, M. Salinity Intrusion Modeling Using Boundary Conditions on a Laboratory Setup: Experimental Analysis and CFD Simulations. *Water* **2024**, *16*, 1970. https://doi.org/10.3390/w16141970

Zangiabadi, E.; Edmunds, M.; Fairley, I.A.; Togneri, M.; Williams, A.J.; Masters, I.; Croft, N. Computational Fluid Dynamics and Visualisation of Coastal Flows in Tidal Channels Supporting Ocean Energy Development. *Energies* **2015**, *8*, 5997-6012. https://doi.org/10.3390/en8065997

Chan, S.N., Lai, A.C.H., Law, A.W.K., Eric Adams, E. (2020). Two-Phase CFD Modeling of Sediment Plumes for Dredge Disposal in Stagnant Water. In: Nguyen, K., Guillou, S., Gourbesville, P., Thiébot, J. (eds) Estuaries and Coastal Zones in Times of Global Change. Springer Water. Springer, Singapore. https://doi.org/10.1007/978-981-15-2081-5_24

Rana A. Al-Zubaidy, Ali N. Hilo, (2022). Numerical investigation of flow behavior at the lateral intake using Computational Fluid Dynamics (CFD). Materials Today: Proceedings. Volume 56, Part 4. Pages 1914-1926. ISSN 2214-7853. https://doi.org/10.1016/j.matpr.2021.11.172.

Sultan, R. A., Rahman, M. A., Rushd, S., Zendehboudi, S., & Kelessidis, V. C. (2018). Validation of CFD model of multiphase flow through pipeline and annular geometries. *Particulate Science and Technology*, *37*(6), 685–697. https://doi.org/10.1080/02726351.2018.1435594

Jayakdoy, H., Al-Dadah, R., Mahmoud, S. (2017). Computational fluid dynamics investigation on indirect contact freeze desalination. Desalination. Volume 420, Pages 21-33. https://doi.org/10.1016/j.desal.2017.06.023

---

## Author Response (AR1)

Response to Reviewer #1 for Manuscript "High-Fidelity Modeling of Turbulent Mixing and Basal Melting in Seawater Intrusion Under Grounded Ice" by Mamer, Robel, Lai, Wilson, and Washam

**General comments:**

The authors aim to tackle a fascinating problem of practical importance. Melting near grounding lines is thought to have much more impact on glacier dynamics than melting anywhere else, yet observations are limited, and high-fidelity simulations are lacking. Thus, the work is highly novel and has significant potential for improving our understanding of ice-ocean interactions that have a high impact on climate dynamics. However, I am not sure that the ANSYS Fluent RANS solver used is appropriate for this problem. In fact, I have never seen the ANSYS Fluent RANS solver applied to environmental flows as complex as in this work. This does not mean that it cannot work, but that significant effort should be devoted to validating the code. As the governing equations solved by the codes are not clearly presented or discussed (in particular, the boundary conditions, which are yet key to evaluating the melting dynamics), I have found it difficult to assess the appropriateness of the mathematical/numerical model. I am particularly concerned with the modelling of the ice-ocean boundary: there are no salt constraints and melt is said to be activated in a way that I did not find correctly physically motivated. In the unfortunate case that the code cannot in fact solve the exact problem at hand (with temperature and salt stratification coupled through a phase change boundary) I would suggest that the authors reformulate the problem as a list of hypotheses--inspired from the full problem--and testable with simulations of a modified/simpler model (but mathematically transparant), which could be the reduced thermal driving model discussed below (point 2).

We thank the reviewer for their thorough comments on this manuscript. We have identified this reviewer's primary concerns to be:

1. The lack of model description and presentation of main equations used by ANSYS Fluent. Along with this, the reviewer also identified the lack of citations supporting the choice of ANSYS Fluent (e.g. validation studies).
2. The lack of a salinity and pressure-dependent thermal boundary condition for the ice boundary.
3. Overall issues with clarity in describing model equations and model initialization.

To address these concerns, we have revised our appendix to present the full model framework, including governing equations, and domain and meshing. We have referenced validation studies in the main text. In addition, we have re-run each of our simulations with a salinity and pressure-dependent ice thermal boundary condition.

We address each comment below. The reviewer's comments are written in black, our responses are given in red, and the original manuscript text is in grey.

**First Comment:**

The governing equations that the code solves should be clearly presented, as I do not think that TC readers are familiar with RANS models (and because I have never seen RANS models used in this context). This requires presenting the Reynolds decomposition of the flow (which would help you justify the fact that 2D dynamics is expected since turbulence is not resolved) and the governing equations for the ensemble-average variables. The closure for the Reynolds stresses and turbulent fluxes should then be discussed in greater details than in the current manuscript. The kappa-eps scheme is mentioned (Eq. (2)) but important details are lacking: for instance, how are kappa and epsilon related to the resolved variables?

We have added a section to the appendix describing the governing equations of the model and further discussion on the turbulence closure scheme. This includes the RANS equations, a description of the Reynolds stresses and closure problem, as well as the choice of turbulence closure used here. Within this section, we have also described the choice to modulate $C_\mu$, as brought up by Reviewer 2, and explained what such modulation does to the flow regime.

I would like also to see more details on the so-called damping functions of the low-Re formulation that supposedly enable accurate diffusive boundary layer representation. Is it like in a wall-resolved large-eddy simulation model?

The standard k-e turbulence model relates eddy viscosity to turbulent kinetic energy and turbulent dissipation to close the system of equations. This requires solving the transport equations for turbulent kinetic energy and dissipation. Near the wall, turbulent kinetic energy vanishes, which produces a singularity in these equations. In order to fix this, a modification must be made to the timescale set by the ratio of turbulent kinetic energy and dissipation. This is where low-Reynolds formulations of the k-e model are helpful, which resolves the transport of turbulent kinetic energy and turbulent dissipation in the low-Reynolds number regions of flow (i.e near no-slip boundaries).

Other formulations of low-Reynolds models are also available within ANSYS Fluent, and we have conducted a sensitivity analysis to determine the most appropriate version. Within the appendix, we have included an explanation of the exact version we use, alongside a description of how it works. We have edited parts of the appendix as shown in the in-line comment #6 below.

**Second Comment:**

The decoupling of the salt dynamics from the ice-ocean boundary dynamics is not justified and a priori seems wrong. I assume that this decoupling is due to code limitations. However, if you cannot justify the decoupling, I am afraid this just means that the code is not suited for environmental flows with temperature and salt stratification and melting. In

the worst case scenario you might consider reformulating the problem in terms of a single scalar variable, namely thermal driving (ref: Adrian Jenkins' papers and other people's related works). The collapse of the full dynamics onto a reduced thermal driving model is thought to be accurate in highly-turbulent environments (for which kappa-epsilon applies anyway) and small salinity variations. This latter condition is obviously problematic (which should be discussed) with regards to your problem of interest. However, I would rather see thermal driving model simulations transparently solved by ANSYS than results from a non-transparant full temperature-salinity model.

Salt dynamics within the fluid regime are resolved and salt is transported as a 'tracer' with an advection-diffusion equation:

$$\frac{\partial}{\partial t}(\rho S) + \nabla \cdot (\rho \vec{u} S) = -\nabla \cdot \vec{J} + R_i$$

Where $\rho$ is fluid density, $\vec{u}$ is the velocity vector, $\vec{J}$ is diffusion flux, and $R_i$ is the net production rate.

The only way salt is not considered is within the melting dynamics. This is in two ways, first, we do not resolve melting or freezing from dissolution – where salt would get into the crystal matrix, suppress the freezing point to below the ice temperature, and therefore cause it to melt. This process would only dominate ice loss if the seawater is subfreezing and therefore mass transfer into the ice matrix would exceed heat transfer.

The other way salinity could affect melting dynamics is by suppressing the ice's thermal boundary condition. In the original simulations, we set the boundary temperature to $0°\,C$. However, based on suggestions from both reviewers, we have re-configured the simulations to have this pressure and salinity-dependent thermal boundary condition for the ice boundaries.

**Third Comment**

The boundary conditions, especially at the ice-ocean interface, should be clearly presented and discussed.

We have further elaborated on the boundary conditions (thermal and kinematic) in both the main text and the appendix where the model formulation is discussed.

The physical motivation for the so-called melt-activated formulation is lacking. Melting produces buoyant flows even when the boundary is no slip, simply because melting acts like a sink for salinity (though this is lacking in your model).

Since salt transport is resolved in the model (i.e. salt is transported as a tracer) there does not need to be a sink for salinity in the melting framework. The model solves for the

displacement and movement of 'saltier' waters via the salt transport equations. Mass is conserved via a pressure outlet (zero gradient flux boundary condition) at the top of the 'ocean' domain.

Thus, should we envision your melt-activated formulation like a velocity compensation for the lack of salt sink in the model? If so, it was not clear to me whether the velocity is prescribed vertically or horizontally, and I am not sure it is the correct way to compensate the salt sink.

The velocity is not compensating for the salt sink, it is the rate at which buoyant freshwater is input into the domain to mimic melting. The velocity is prescribed normal to the boundary where it is sourced, in this case vertical.

Estimating the velocity to enforce at the boundary to mimick the salinity sink from Eq. (4) is also not justified, i.e. why should the movement of the interface (assuming there is no immediate hydrostatic equilibrium of the ice shelf at such small scales) be directly used as a buoyancy-driven velocity input?

We acknowledge that adding a vertical velocity to mimic the vertical movement of a solid is a bit crude, however, because this vertical velocity (the melt rate) is small relative to the main flow and is negligible in the momentum equations, we believe the decision is justified.

**Fourth Comment**

In would like to see a validation of the code, which includes the choice of turbulence closure. A simple benchmark case should be set-up, for which turbulence-resolving simulation data exist (either from direct numerical simulation (DNS) or large-eddy simulation (LES)). Several groups (with Catherine Vreugdenhil, John Taylor, Ken Zhao etc) have published such DNS and/or LES data over the past 5 years or so, making it practical. Typical configurations are channel flow configurations, which should be accessible to ANSYS. Validation could be based on quantitative comparisons of mean variable profiles (e.g. temperature, TKE) normal to the ice-ocean interface.

ANSYS Fluent was chosen to address this research question because of its extensive validation and history of practical use within the engineering community. Examples of validation exist for many different flow regimes, here are a few about environmental flows:

- Chalá, D.C.; Castro-Faccetti, C.; Quiñones-Bolaños, E.; Mehrvar, M. Salinity Intrusion Modeling Using Boundary Conditions on a Laboratory Setup: Experimental Analysis and CFD Simulations. *Water* **2024**, *16*, 1970. https://doi.org/10.3390/w16141970

- Zangiabadi, E.; Edmunds, M.; Fairley, I.A.; Togneri, M.; Williams, A.J.; Masters, I.; Croft, N. Computational Fluid Dynamics and Visualisation of Coastal Flows in Tidal Channels Supporting Ocean Energy Development. *Energies* **2015**, *8*, 5997-6012. https://doi.org/10.3390/en8065997

- Chan, S.N., Lai, A.C.H., Law, A.W.K., Eric Adams, E. (2020). Two-Phase CFD Modeling of Sediment Plumes for Dredge Disposal in Stagnant Water. In: Nguyen, K., Guillou, S., Gourbesville, P., Thiébot, J. (eds) Estuaries and Coastal Zones in Times of Global Change. Springer Water. Springer, Singapore. https://doi.org/10.1007/978-981-15-2081-5_24

- Rana A. Al-Zubaidy, Ali N. Hilo, (2022). Numerical investigation of flow behavior at the lateral intake using Computational Fluid Dynamics (CFD). Materials Today: Proceedings. Volume 56, Part 4. Pages 1914-1926. ISSN 2214-7853. https://doi.org/10.1016/j.matpr.2021.11.172.

- Sultan, R. A., Rahman, M. A., Rushd, S., Zendehboudi, S., & Kelessidis, V. C. (2018). Validation of CFD model of multiphase flow through pipeline and annular geometries. *Particulate Science and Technology*, *37*(6), 685–697. https://doi.org/10.1080/02726351.2018.1435594

- Oh, CH, & Kim, ES. "Validations of CFD Code for Density-Gradient Driven Air Ingress Stratified Flow." *Proceedings of the 18th International Conference on Nuclear Engineering*. *18th International Conference on Nuclear Engineering: Volume 6*. Xi'an, China. May 17–21, 2010. pp. 201-209. ASME. https://doi.org/10.1115/ICONE18-29807

We believe any additional validation would be outside the scope of this paper.

**Fifth Comment**

Successful applications of ANSYS Fluent RANS solver to environmental flow configurations with temperature and salinity stratification should be cited and discussed (in particular how they validated the code).

We appreciate the reviewer highlighting the lack of justification for choosing ANSYS Fluent within the paper. We have incorporated examples of validation and applications of ANSYS Fluent to stratified and/or complex environmental flow scenarios.

**Sixth Comment**

The discussion of the steady state and transient runs is really confusing. You should distinguish the existence (or non-existence) of a steady state from your strategy of

successive runs to achieve it. The key point that should be in the main text is that the problem has a natural steady state (for the ensemble-average variables) as there is no external variability and the ensemble-average variables do not exhibit temporal fluctuations once equilibrated. The strategy to reach it should then be discussed in an appendix.

We have revised our methodology to determine if a steady state has been reached. Instead of switching between solver types (transient vs. steady-state), we use only the transient solver. The domain is initialized with a salty, warm ocean tank and fresh, cold subglacial environment (e.g. Figure  B1 in the revised draft). The simulation then runs for 8640 time steps with a time step of 5s. A quasi steady-state for most simulations is reached during this time, as decided by a change of less than 0.0001 kg/m^3 over several timesteps in the spatially averaged subglacial environment density. The simulations that did not reach a quasi-steady state in the first 8640 time steps were run for another 8640 time steps. The results presented in the main text are time-averaged from once a quasi-steady-state is reached. We have elaborated on this methodology in the methods section and the appendix.

**Seventh Comment**

I have found many typos in the appendices ("Need to list reference values, materials info, the methods and controls" found line 566), suggesting that these were not carefully reviewed by the authors. At the moment the appendices seem primarily like a draft list of comments and figures that did not make it into the main text (some in fact repeating what is already in the main text).

A careful review of the appendix has been conducted. In addition, we have added further descriptions of the governing equations (major comment 1), model boundary conditions (major comment 3), designating a quasi-steady-state, and analysis of turbulence parameters. In addition, we have moved extra simulations and figures to the supplementary document to make the appendix more concise.

**Eighth Comment**

The figures are lisible but could be improved(e.g. the x-axis label of Fig. 6 is quite small).

We have standardized the figures' text sizes and shapes for all figure axes and titles.

**Ninth Comment**

Because of the many questions I had/have with respect to the simulation code, I was not able to appreciate the comparison of the simulation results with the parameterized predictions of melt rates. If the authors can validate their code, I agree that this comparison would be an important addition to the paper, but it would have to be a fair

comparison. That is, if the authors end up solving a model that is distinct from the model that the parameterizations (necessarily approximate) aim to mimic (arguably the real exact model), they should discuss result differences in light of model differences.

We understand the concerns of the simulation code to be: salinity sinks, ice-ocean boundary temperature independence of salinity, and vertical velocity as a proxy for a moving ice interface.

We reiterate that salt transport is resolved in these simulations and the only way it is not represented in the melting framework is by having a suppressed boundary temperature. Including this latter point would only increase our simulated melt rates, by increasing the thermal driving. Since the parameterized melt rates are already lower than the simulated melt rates, including a boundary temperature dependent on near-ice salinity would further increase the disagreement between simulated melt rates and parameterized melt rates. Our discussion of the simulation-parameterization disagreement attempts to highlight the differences in light of the assumptions of the fluid structure that are inherent in the parameterization. i.e. well-mixed, higher Reynolds number, etc.

However, considering comments from both reviewers, we have re-run each simulation with a thermal boundary condition that is dependent on near-wall salinity and a reference glaciostatic pressure (i.e. at 1000 m).

Since salt transport is represented in the model by its advection-diffusion equation, a sink does not need to be provided where meltwater enters the domain. Mass is conserved via a pressure outlet (zero gradient flux) downstream in the domain.

We acknowledge inputting a vertical velocity as a proxy for a moving ice interface is crude, however, because this vertical velocity is O(-6) and the freestream flow is O(-1) – O(-3) we think this choice is justifiable.

**Tenth Comment**

Simulation snapshots and movies would really help visualize the flow.

We have included a figure representing the initial state of the flow in the revised appendix (Figure B1).

**Additional comments, including technical corrections: typing errors, etc.**

1. line 29: could you clarify the idea of "tidally asymmetric"?

   o Added "Such asymmetry results in stronger melting during the ascent of high tide and weaker melting during the tidal ebb." To the next line.

2. line 87: writing "The ice wall boundaries have a temperature boundary condition of 0∘C" really felt like a bomb! And the lack of justification or description of the full mathematical model did not help disarm it. It is really not expected that prescribing 0∘C at the ice-water interface is reasonable, especially when the salinity goes from 0 to 30 psu.

   o Based on suggestions from both reviewers, we have revised this thermal boundary condition to be salinity and pressure dependent following :

$$T_b = S\lambda_1 + \lambda_2 + z\,\lambda_3$$

   We have set the pressure dependence to a reference state of 1000m thick ice column.

3. line 108: this equation of state may be suited for small salinity variations, but with a range of 0 to 30 psu, it may not be appropriate, unless the grounding line is beneath a lot of ice. I would recommend that you use a higher-order approximation of the true equation of state, or at least acknowledge that you know of the anomaly of the equation of state at low salinity and pressure but that you discard its effects to simplify the problem. Ideally you could discuss/speculate on how the results might change should you consider an accurate equation of state.

   o We appreciate you bringing this concern to our attention. We have run a sensitivity test comparing our linear E.O.S. to the higher-order approximation in Roquet et al. (2015):

   1. $\rho' = -\frac{C_b}{2}(\Theta - \Theta_0)^2 - T_h Z\Theta + b_0 S_A$
   2. $C_b = 0.011\ kg/m^3 K^2$
   3. $T_h = 2.5\ x\ 10^{-5} kg/m^4 K$
   4. $b_0 = 0.77\ kg/m^3 (kg/g)$
   5. $\Theta_0 = -4.5°C$

   The results of this sensitivity test are discussed in the revised appendix.

4. line 113: conduction, diffusion, and molecular dissipation all sound like the same thing to me. Could you explain how they differ? (reading your appendices it looks like conduction might be turbulent conduction)

- o Conduction represents the transfer of heat due to the thermal gradient. Within ANSYS Fluent, the conductive value used is the 'effective conductivity' which is the summation of the thermal conductivity of the fluid and turbulent conductivity. The diffusive term represents heat transfer due to species diffusion – in this case, salt. In equation A7, this term is the sum of every species' diffusive flux multiplied by their enthalpy. Molecular dissipation is another way of phrasing viscous dissipation, which is the heat transfer due to viscous forces, represented by the effective shear stress and kinematic viscosity.

- o Edited line 113 : Energy, and therefore fluid temperature, is evolved via an energy conservation equation employed by the CFD solver resolving advection, conduction, salt diffusion, and viscous dissipation . (ANSYS Inc., 2024). Conduction represents heat transfer due to thermal gradients, and viscous dissipation is the transformation of kinetic energy into thermal energy due to shear forces. As salt diffuses in the medium, it also transfers heat due to its unique thermal properties, and therefore must also be included.

5. line 124: the quadratic quantity indicated is one among many, such that the sentence does not read well. Please reformulate.

    - o Edited lines 123 - 127 shown below.

6. line 126: could you recall what the Boussinesq hypothesis is?

    - o Edited lines 123-127: A closure scheme is necessary because averaging the RANS equations introduces Reynolds stresses due to turbulent motion within the fluid. Reynold's stresses take the form $\overline{u_i' u_j'}$ , the averaged product of turbulent velocity fluctuations. One class of closure models employs the turbulent-viscosity hypothesis, which relates the deviatoric Reynolds Stresses to the mean strain rate via a positive scalar eddy viscosity (Pope, 2000). Here, we utilize the two-equation $k - \epsilon$ closure scheme, which solves for the eddy viscosity by:

$$\mu_T = \rho_w C_\mu \, k^2/\epsilon$$

    - o Added to the appendix: The turbulent-viscosity hypothesis (also known as the Boussinesq Hypothesis) is used in turbulence modeling to solve for the Reynolds stresses. This hypothesis states the deviatoric Reynolds stresses (those deviating from the mean) are proportional to the mean strain rate tensor by a positive scalar. This scalar represents the eddy viscosity (also referred to as turbulent viscosity). This relationship is:

$$-\rho \overline{u_i u_j} + \frac{2}{3} \, \rho k \, \delta_{ij} = \rho \mu_T \left( \frac{\partial \overline{U_i}}{\partial x_j} + \frac{\partial \overline{U_j}}{\partial x_i} \right)$$

The only unknown left in the system of equations is the eddy viscosity, which can be solved for by a variety of different turbulence closure schemes. Here, we employ the two-equation $k - \epsilon$ closure scheme which solves for eddy viscosity by relating it to the square of turbulent kinetic energy and inverse of turbulent dissipation by a positive scalar $C_\mu$. This closure scheme requires two additional equations to solve for turbulent kinetic energy and turbulent dissipation. These equations are:

$$\frac{\partial}{\partial t}(\rho k) + \frac{\partial}{\partial x_i}(\rho k\, u_i) = \frac{\partial}{\partial x_j}\left[\left(\mu + \frac{\mu_t}{\sigma_k}\right)\frac{\partial k}{\partial x_j}\right] + G_k + G_b - \rho\epsilon - Y_M + S_k$$

For turbulent kinetic energy and:

$$\frac{\partial}{\partial t}(\rho\epsilon) + \frac{\partial}{\partial x_i}(\partial\epsilon u_i) = \frac{\partial}{\partial x_j}\left[\left(\mu + \frac{\mu_t}{\sigma_k}\right)\frac{\partial\epsilon}{\partial x_j}\right] + C_{1\epsilon}\frac{\epsilon}{k}(G_k + C_{3\epsilon}G_b) - C_{2\epsilon}\rho\frac{\epsilon^2}{k} + S_\epsilon$$

For turbulent dissipation. Note, in the equation for turbulent dissipation, turbulent kinetic energy is in the denominator which results in issues when k approaches zero near wall boundaries. To resolve boundary layer dynamics with the k-e closure, we use the low-Reynolds number formulation which employs damping functions and fixes the singularity that arises with low values of k.

A version of the low-re k-e closure we employ here is the Yang-Shih version (Yang and Shih 1993). In this formulation, the authors set the near-wall turbulence timescale to be the Kolmogorov timescale ($T_k \propto (\nu/\epsilon)^{1/2}$). In doing so, the equation for eddy viscosity becomes:

$$\nu_T = C_\mu\, f_\mu\, k\, \left(T_k + \frac{k}{\epsilon}\right)$$

Where $f_\mu$ is the "damping function" and equal to:

$$f_\mu = \left[1 - exp\left(-a_1\, R_y - a_3\, R_y^3 - a_5\, R_y^5\right)\right]^{1/2}$$

and $R_y = \frac{k^{\frac{1}{2}}y}{\nu}$. The constants $a_1$, $a_3$, and $a_5$ are constrained from DNS experiments for turbulent channel flow.

The final adjustment to the standard k-e formulation for near-wall flows is to add an additional source of dissipation, which results from inhomogeneity in the mean flow field. This takes the form:

$$E = \nu\nu_T \frac{\partial U_i}{\partial U_j \partial U_z} \frac{\partial U_i}{\partial U_j \partial U_z}$$

This formulation of the low-re k-e turbulence closure allows for the free-stream portion of the flow regime as well as the near-wall region where viscous effects dominate since the added terms tend to zero when turbulence is high.

7. Eq. (2) P2: what are the equations for kappa and epsilon? I expect that they involve the resolved ensemble-average variables.

   o We have added these equations to the appendix in the model description as written above in comment 6.

8. Section 2.4: this section is really confusing as I already mentioned earlier, because buoyancy isn't due to some interfacial velocities but to changes in salinity and temperature at the phase-change boundary. Please reformulate.

   o We have reformulated how we discuss the buoyancy effects that arise due to the input of fresh, cold water. We have emphasized that any buoyancy effects are exclusively due to the difference in density between the input meltwater and the ambient water, and not due to some prescribed velocity (which wouldn't be a buoyancy effect).

9. line 150: I don't see the physical justification for prescribing a "horizontal ice (?!) velocity" at the horizontal ice-water interface.

   o A fluid velocity-inlet boundary condition is applied to the horizontal ice base when melting is enabled. The fluid velocity prescribed here is normal to the boundary and therefore is downward, not horizontal.

   o Edited line 150: The downward fluid velocity prescribed at the horizontal ice face is set by the melt rate, $\dot{m}$, and is a function of the difference between the near-wall cel''s centroid temperature $T_w$ and the ice-ocean interfacial temperature $T_b$, thermal conductivity $k_T$, and density of ice $\rho_i$:

10. line 179: can you provide a reference for "realistic estuarine-like mixing rates"?

    o We appreciate the reviewer bringing the lack of citations to our attention. We have amended line 179 to:

      1. Turbulent mixing, as modulated by $C_{\mu}$, affects intrusion distance to a lesser degree than freshwater discharge velocity when varied over a wide range encompassing likely values on the lower-end for realistic estuarine-like mixing rates (Geyer et al. 2000, Geyer et al. 2008).

11. line 193: rewrite "retrograde bed slope".

    o We have done this.

12. You refer to figure 5 before figure 4 so the two should be swapped.

- o We have done this.

13. line 253: Could we say "vertical baroclinic convective motion"?

- o We appreciate this suggestion and have made this edit.

14. Eq. (9) line 325: this equation doesn't look like a parameterization but rather like the exact expression for the melt rate as a function of the conductive heat flux at the interface.

- o Based on both reviewers' suggestions, we have removed this section from the results.

15. Fig. 5-7: it was not clear to me whether the results you plotted were for the melt-enabled model or not.

- o In Figure 5, the dashed lines represent melt-enabled scenarios as denoted in the caption. To address confusion, we have added:

  - o When melt is enabled the horizontal extent of stratification in the subglacial environment is reduced, but where stratification occurs, it is stronger (e.g. dashed lines in Figure 5).

- o For Figure 6, all data comes from melt-enabled cases, as denoted in the caption. We have added to line 341 to improve clarity:

  - o Here, we tested the sensitivity of equations 6-9 to various choices of ice distance to obtain Tw, Sw, ρw, and u for the melt-enabled cases.

- o For Figure 7, both melt-enabled and no-melting cases are represented as denoted in the legend. We have edited the caption to improve clarity:

  - o Wilson et al. (2020) experimental data (gray markers) and intrusion characteristics found in this study (red and blue markers). The red markers represent simulations with melting enabled, and blue markers represent non-melting simulations. The black dashed line is the numerical solution to Robel et al. (2022) with $\gamma = 2$.

  In addition, we have added to lines 387-389:

  - o Our simulated intrusions for both non-melting and melt-enabled scenarios follow the general trend and scale sensitivity to those identified in previous laboratory experiments (Figure. 7) (Wilson et al. 2020) which are within a factor of 10 to the theoretical prediction (dashed line) from Robel et al. (2022).

16. Eq. (A3): should it be Re_L? Or change L into x?

   o  We appreciate you pointing out this inconsistency. We have amended the equation to be:

   o  $Re_L = \frac{uL\rho_w}{\mu}$

17. Appendix A4: I have found many typos, you need to use capital letters for Reynolds number, kappa has become k etc Eq (A7) has signs/symbols displaced. Please discuss Eq (A7) more carefully. What is tau_eff? What is species diffusion in your case?

   o  In combination with previous comments (e.g. 3, 4,5, 6, 7, and major comments 1 and 5) we have revised the appendix to describe the model in more detail and justify model choices. In this rewrite, we have standardized the symbols used and refined appendix equations.

   o  $T_{eff}$ is the effective shear stress, which includes viscous shearing effects as well as shear from the no-slip boundary conditions.

   o  Species Diffusion in our case represents salt diffusion. Since we are transporting salt as a tracer throughout the fluid, we must include its heat transport into the energy equation.

Works Cited

Pope, S.B. (2000). Turbulent Flows. *Cambridge University Press*, Cambridge, 305-308.

Geyer, W.R., Trowbridge, J. H., & Bowen, M.M. (2000) The Dynamics of a Partially Mixed Estuary. *J. Phys. Oceanogr.* **30**, Iss 8, 2035-2048. https://doi.org/10.1175/1520-0485(2000)030<2035:TDOAPM>2.0.CO;2

Geyer, W.R., Scully, M.E. & Ralston, D.K. (2008). Quantifying vertical mixing in estuaries. *Environ Fluid Mech* **8**, 495–509. https://doi.org/10.1007/s10652-008-9107-2

Roquet, F., Madex, G., Brodeau, L., Nycander, J. (2015). Defining a Simplified Yet "Realistic" Equation of State for Seawater. *J. Phys. Oceanogr.* **45**, Iss 10, 2564-2579. https://doi.org/10.1175/JPO-D-15-0080.1

Response to Reviewer #2 for Manuscript "High-Fidelity Modeling of Turbulent Mixing and Basal Melting in Seawater Intrusion Under Grounded Ice" by Mamer, Robel, Lai, Wilson, and Washam

**Summary:**
This paper studies the estuarine-type dynamics of seawater intrusion/subglacial discharge upstream of an ice sheet grounding line in a CFD model (ANSYS fluent). The subglacial environment is two-dimensional with a large aspect ratio (5cm high and 3-5m long) and is forced by fresh subglacial discharge upstream and salty inflow from the ocean downstream. The authors investigate the structure and distance of the saltwater intrusion, and its sensitivity to various parameters (primarily freshwater discharge velocity). They find that that the height of the subglacial environment and the velocity of the freshwater discharge are strong controls on the intrusion distance, and the strength of turbulent mixing has a somewhat weaker/more ambiguous effect. The authors also investigate basal melt at the upper surface of the subglacial region and how it affects the intrusion (to do so they must change the boundary condition at the top surface away from the no-slip condition), finding that melt decreases the intrusion distance, offering a possible negative feedback.

This is an interesting study of an important process; however, I have several major concerns with the paper that may affect the conclusions. If they are addressed, I would be happy to review a revised manuscript.

We appreciate the thoughtful review and valuable feedback given by reviewer 2. We have identified the reviewer's main concerns to be:

1. The lack of turbulence statistics in describing differences between laminar and turbulent cases and in justifying the modeling choices to vary $C_\mu$.
2. The lack of a pressure and salinity dependent thermal boundary condition for the ice boundary. Furthermore, the lack of a free-slip case to compare the melt-enabled simulations to.
3. The incomplete discussion of a double-diffusive/diffusive mixing regime and inappropriate diffusive melting comparison to modeled melt.
4. Overall issues with clarity in describing the numerous simulations.

To address these concerns, we have re-run each of our simulations with a salinity and pressure-dependent thermal boundary condition and added additional cases with a free-slip ice kinematic boundary condition. We have provided a turbulent viscosity figure in the appendix (Figure D1) to aid in discussing the turbulence metrics. The consideration of diffusive-convective mixing has been limited to the discussion section and removed from the results section. We have provided tables of the various simulations to improve results clarity.

Below, we address each concern in greater detail. The reviewer's comments are written in black, our responses are given in red, and the original manuscript text is in grey.

**First Comment:**

I am concerned about the turbulence modelling and the lack of inclusion of turbulence quantities in the paper. The fidelity (reported to be high!) of these simulations relies on the appropriateness of the turbulence closure, however, the reader is not given any context for the choice/s of C_mu,

The typical value assigned to $C_\mu$ is 0.09, which has been empirically found for simple wall-bounded flows (Pope, 2000). However, for complex stratified flows, or highly energetic jets, a standard constant value for the whole domain may not be appropriate (e.g. Lai et al., 2019).  Since $C_\mu$ could be found via the relationship:

$$C_\mu = \frac{\nu_t \epsilon}{k^2 \rho}$$

it is clear that the 'real' value of $C_\mu$ is highly dependent on turbulence dynamics. In a modeling framework, we have to prescribe $C_\mu$ in order to obtain eddy viscosity, it is not a post-processing derived value. Since a 'true' or 'real' value for $C_\mu$ has not been found for the flow regime considered here (stratified), we deemed it an appropriate approach to modulate this value in order to induce more or less turbulent mixing. Our original intention in setting $C_\mu$ was to span three orders magnitude, with the standard value of 0.09 as the 'middle case'. However, initial runs demonstrated instability within the model when $C_\mu$ was set to 0.9. On the other end, when C_mu is very low ($\to$ 0) the flow regime becomes laminar, which would not be helpful to address the research question at hand. Based on this, we chose to range the value by a factor of ¼.

nor are we given any more intuitive quantities (e.g. the resulting turbulent diffusivity) to better understand the effect of varying C_mu, and how increased turbulence affects the flow.

We agree that turbulent viscosity would be more intuitive in explaining what changing the $C_\mu$ value results in. We have included a figure in the appendix depicting turbulent viscosity values across all freshwater velocities and turbulence levels for the standard geometry (Figure D1).

Without this information and context, I have low confidence in the modelling overall, especially since the effect of varying C_mu is non-monotonic for some cases (Fig 2B) and counter to my expectations for others (Figs 2A & C), i.e. I would have expected that increased mixing would decrease intrusion distance, whereas the authors find the opposite result. My recommendations to address this are as follows:
- o Present the turbulent diffusivity (or viscosity) alongside T, S, u. In main MS or in a supplement

This has been done and is appendix figure D1 in the revised manuscript.

- o Present the Reynolds numbers for the cases

This has been done. The Reynolds numbers are approximately 25, 250, and 2500 for the low, medium, and fastest freshwater cases presented.

- o Give some context for the choice of C_mu for similar flows – I presume the engineering literature can provide. Stratified plane couette flow may be a starting point in terms of a bounded, stratified flow.

To increase turbulent mixing, we could either increase the flow velocity or increase the eddy viscosity via $C_\mu$. Since we want to look at low Reynolds numbers, we modify the $C_\mu$ value to mimic a changing turbulent mixing scenario.

In setting $C_\mu$, we found numerical instability when $C_\mu \to 1$, therefore limiting our upper-end choice to a factor of 2 of the standard value used. For the lower-end case, we wanted to avoid choosing a too small value, since the solution relaxes to laminar flow when $C_\mu \to 0$. This constrained our lower-end choice to a factor of 0.5 of the standard value.

The empirically derived value for $C_\mu$ that is traditionally used, 0.09, is for standard wall-bounded simple flow regimes. Under scenarios of high energetics (like a jet-plume) the $C_\mu$ value can approach 0.3 (Lai et al., 2019). In parts of our domain, a jet-plume does form, therefore making a relatively increased $C_\mu$ value realistic.

**Second Comment:**

I don't understand why the temperature profiles have much sharper gradients than the salinity gradients (c.f. the darkest line in Fig A8A to the darkest line in Fig A8B). The authors mention the steep T gradients (not seen in S) in line 202 *(Such a steep thermocline is most likely due to the temperature boundary condition we imposed on the horizontal ice boundary)* however, I am concerned that the problem runs deeper than different boundary conditions. Why, for example, is temperature at 2cm depth at the GL entrance at 0.5 degrees (i.e. unmodified from ambient conditions), while salinity is <20ppt at the same location? There may be a serious issue with the mixing of these scalars which would significantly affect your results. Temperature-salinity plots may help determine if there is a problem.

We appreciate the reviewer bringing this to our attention. We have found a mis-assigned diffusion coefficient. We have re-run all simulations with this fixed diffusion coefficient and it has increased intrusion distances for all cases (Figure 2 in the revised manuscript). Furthermore, the distribution of heat and salt are now more realistic (Figure 3 in the revised manuscript). We have also provided TS diagrams in the supplementary for both meltenabled and non-melt enabled simulations for all geometries and freshwater velocities tested.

**Third Comment**

By comparing the temperature and salinity profiles as u_f is increased (Figs 3 A,B, Figs A8 A,B,D,E), it appears to me that the greatest control on mixing is time spent in the subglacial channel. For example, at u_f=0.05 cm/s salinity is quite well-mixed over the full depth of the channel, and varies mostly with distance along-channel, implying that diffusion (rather than advection) is dominating transport. For u_f=5cm/s, the top, outflowing layer remains fresh, indicating that the transport dominated by advection. This is somewhat counter to my expectations that turbulent mixing should be stronger for the higher velocity cases. This result should be investigated and discussed. As for comment (1), plots of the turbulent diffusivities for each case may be enlightening.

We have revised Figure 3 to include vertical profiles of thermal forcing, salinity, velocity, and buoyancy frequency across all freshwater velocities tested. This updated figure illustrates the relationship described above: slower freshwater velocities exhibit more uniformity in the vertical direction, while faster velocities lead to higher stratification. This pattern is similar to what is observed in estuaries and is likely due to boundary shear being a more dominant factor in slower freshwater flows. In estuaries, river outflow must overcome tidal shear and wind-driven surface shear to maintain stratification. Similarly, under ice, the fluid must overcome shear from both the ice and rock boundaries.

**Fourth Comment**

The different BC between the melt-enabled and no-melt cases makes it impossible to attribute changes in the intrusion to the effect of melting alone. An additional case is needed with free slip and no melting to better separate these effects.

We have conducted additional free-slip cases to compare the melt-enabled cases to. These results are presented in Figure 2 and Table 2 in the revised manuscript.

**Fifth Comment**

There are numerous different simulations included in the paper, however, few where one variable is systematically changed. This makes the paper/results hard to follow at times. A results table or bar chart showing how the intrusion distance changes across simulations would go some way to addressing this.

We have added three tables in the main text (Tables 1-3)  and 2 tables to the appendix (Tables B1 and B2) to describe the model setup, parameters, and intrusion distance. We did not include the drag coefficient and melt rate since they vary significantly over the length of the intrusion.

**Sixth Comment**

No salinity effect on the melt. In the simulations, the interface is salty (Fig 3E) which will act to depress the freezing temperature and therefore the interface temperature, altering the melt rate. The authors should state why they have not included this extremely important effect (in more detail than "model limitations" line 437). In addition, some discussion of the likely effect of this simplification on the results is needed.

We have re-run all simulations with an updated thermal boundary condition on the ice face following the salinity and pressure dependent freezing point:
$$T_b = S\,\lambda_1 + \lambda_2 + z\lambda_3$$

We have set z to be 1000 m and also prescribed the inflowing freshwater boundary to be at the pressure dependent freezing point.

**Seventh Comment**

Double diffusive framework. Equation 9 is the same as Equation 4 (with marginally different thermal diffusivity) and is therefore not a parameterization to be tested, rather a re-stating of your melt BC. I propose removing Fig 6B and the "diffusive melt" in Fig 5. Notation m_DC and m_DDC (which are interchangeably used) are inappropriate here.

We have removed the double diffusion framework from the results section, limiting the parameterizations to the heat-limited two-equation formulation of Holland and Jenkins (1999). However, we have kept the consideration of the double-diffusive melting mechanism to the regime of seawater intrusion within the discussion section.

**Eighth Comment**

Transfer velocities/Stanton numbers (Figure 6 )– I don't really understand the purpose of C and D. All D shows is the weak dependence of the Stanton number on ustar at low ustar (see the denominator of (7)), and all C shows is the same but multiplied by ustar. Since the S99 parameterization can't accurately predict melting for your simulations, this (trivial) result becomes misleading, as readers may think that it lends support to a certain Stanton number being useful more broadly for modelling ice-ocean interactions. If you want to compare the Stanton number of your simulations to those found in the literature, you need to rearrange equation (6) less the conductive ice flux to solve for the Stanton number (and the (unitless) transfer coefficient \$Gamma_T$) using your model output melt rate, temperature etc.

We have removed this figure and discussion of it from the results section. Instead, we have focused on two ways of using the heat-limited shear driven parameterization framework. In the first approach, we directly calculate the turbulent transfer coefficient along the

horizontal ice boundary using the time-averaged simulation data. The second method involves finding a "tuned" turbulent transfer coefficient. We do this by evaluating the melt rate over a range of turbulent transfer coefficients and find which one has the lowest root mean squared error with the simulated melt rate. Both of these are presented in Figure 5 of the revised manuscript and discussed in the results section.

**Other Comments**

1. Line 83 – move references (carter…) after "non-summer months"
   o Fixed

2. Line 97 - "Our vertical domain size is at the upper bound of the viscous sublayer length scale that could exist between a well-mixed boundary layer and the ice" I don't know what this means, please clarify.
   o The domain size and velocities (Reynolds number) used in these simulations constrain the development of the boundary layer and freestream flow. In this Reynolds number regime, we don't see the development of a complete boundary layer with flow unaffected by the boundary (freestream flow)
   o To clarify, we have made these edits in the main text:
   o  For the tested freshwater velocities, the vertical domain size hinders the development of a full boundary layer. Instead, everywhere in the domain, the fluid feels the effects of the wall boundary.

3. Line 154 – should be dT/dy in your coordinates
   o Fixed

4. Line 177 – "vary by a factor of 1000 **in response to the range of input velocities tested here**".
   o Fixed

5. Line 179 – insert comma before "increased"
   o Fixed

6. Line 180 – "For the middle freshwater velocity (Figure 2 C)," - should this be 2A?
   o Fixed. We appreciate you finding this error.

7. Line 182 – "To contrast the effects of turbulent mixing, we tested a laminar flow case with no turbulent mixing (green line Figure 2 A) and saw no meaningful difference in intrusion distance." It would be good to compare and contrast this case more, i.e.

how different is the T/S structure? If it's not different, then presumably turbulent mixing is not occurring in the channel.

- o We appreciate this suggestion and have included a supplementary figure with vertical profiles for the standard geometry with u_f = 0.5 cm/s across all turbulence levels to compare the effect of turbulent mixing more.

8. Line 197/8 – refs should not be in parentheses.
   - o Fixed.

9. Line 212 – How is Cd calculated? i.e. at what value of y are ustar and u evaluated? Also, as mentioned earlier I think Cd is a main result and this figure should come to the main text.
   - o To calculate $C_d$, we use the relationships between drag coefficient and wall shear stress, $\tau_w$ (Pope, 2000):
     - $C_d = \tau_w \ / \ (0.5 \ \rho \ \bar{u}^2)$
     - $\tau_w \ = \mu \left(\frac{\partial \bar{u}}{\partial y}\right)$
   - o Where $\rho$ is the fluid density, and $\bar{u}^2$, is the mean freestream flow.
   - o We can rearrange to solve for $C_d$:
     - $C_d = \frac{\nu}{2\bar{u}^2} \left(\frac{\partial \bar{u}}{\partial y}\right)$
   - o Where $\nu$ is the kinematic viscosity and equivalent to $\mu/\rho$, the dynamic viscosity divided by density.
   - o The gradient in mean streamwise velocity $\left(\frac{\partial \bar{u}}{\partial y}\right)$ is found by fitting a line to the upper half of the freshwater layer. We provide more details in the revised appendix and supplementary document.
   - o We have refined the methodology of calculating drag and put this figure into the main text as Figure 4. The methodology is discussed in various levels of detail in the revised main text, appendix, and supplementary.

10. Line 241 – "Reduction in velocity gradients arises from an increase in stratification, suppressing turbulence, and the kinematic boundary condition being a velocity inlet and not a no-slip wall." – as per major comment 4, actually you can't isolate these effects currently and a no-slip no-melt case is needed.
    - o We have conducted free slip cases for all simulation geometries and freshwater velocities to compare the melt-enabled cases to. The comparison between these cases have replace the results quoted above.

11. Line 256 –Predominantly horizontal motion?
    - o The motion is predominately horizontal from how the kinematic boundaries are prescribed. However, the horizontal density gradient that arises due to the characteristic 'wedge' shape of the seawater intrusion could introduce vertical convective motion to 'flatten' out the intrusion. This vertical motion

may drive interfacial mixing and is an important mechanism to reduce the strength of stratification.

- o To further clarify, we have edited lines 252-258:

  However, the horizontal density gradient introduced by the characteristic wedge shape of seawater intrusion will drive vertical baroclinic convective motion to flatten isopycnals. Such baroclinic adjustment may be an important source of interfacial mixing, working in tandem with turbulence and double-diffusive convection to reduce stratification within the subglacial environment. This convective-driven mixing mechanism differs from convective mixing caused by a sloping ice boundary, in which a buoyant plume may form. For the idealized scenarios in this study, buoyant convection via ice geometry will not drive mixing and thus melt since the ice is perfectly horizontal.

12. Line 261 – again, you can't attribute this to increased stratification (which MAY decrease shear/drag) because you haven't isolated the effect of the no-slip BC (which WILL decrease shear/drag)
    - o We have amended any discussion comparing the melt-enabled cases to the non-melting cases to be for those with a free-slip boundary condition.

13. Line 286 – Diffusive convective melting also involves convection driven by cooling and is different from diffusive melting. See Martin & Kauffman (1977) section 3 for diffusive melting and Martin & Kauffman (1977) section 4 for diffusive-convective melting.
    - o We appreciate this insight. We have removed the discussion of diffusive convective and double diffusive convective-driven melting from the results section.

14. Fig 3 – Is any averaging (time or space) done to obtain these profiles?
    - o Yes, these are vertical profiles from the time-averaged domain. In order to clarify, we have edited the caption:

      Figure 3. Time-averaged vertical profiles of temperature(A, D), salinity(B, E), and x-component of velocity (C, F) along the seawater intrusion for $u_f$ =0.5 cm/s and medium turbulence $C_\mu$ = 0.09. The distance beyond the grounding line represents the distance (m) upstream of the fixed grounding line. The top row (A, B, C) is for the non-melt enabled case, i.e. the horizontal ice boundary is a wall boundary with a fixed temperature. The bottom row (D, E, F) is for the melt-enabled case where the ice boundary becomes a velocity inlet with freshwater inflow as a function of near-wall temperature.

      And figure discussion at line 200:

The time-averaged vertical profiles of temperature, salinity, and velocity along the intrusion for non-melt-enabled cases (Figure 3) depict a two-layered flow in opposing directions, with a relatively uniform low-sloping vertical gradient in salinity, and a strong thermocline in the 2 cm directly below the ice.

15. Fig 3 - It would be great to see the systematic change in the intrusion as u_f is increased with a series of side-by-side plots, rather than having to move back and forward from figs 3 to A8.
    o We have presented the vertical profiles for thermal forcing, salinity, velocity, and buoyancy frequency in the revised main text as Figure 3.

16. Line 319 – I would not say that the flow is weak at 5,10 cm/s. However, the height of the channel is very small, so the Reynolds number will be small. Again, Re or other turbulence metrics are needed.
    o All cases tested here have low Reynolds numbers ranging from 24 to 2400. We have included this in the discussion of the methods. In addition, the earlier suggestion to discuss turbulent viscosity metrics has been illuminating and we have used this variable to describe the energetics within the flow field.

17. Line 397 – how is the reduced gravity calculated here? Based on the density difference between the freshwater and saltwater?
    o The reduced gravity is calculated based on the density difference between pure freshwater and pure saltwater. However, a point can be made that the reduced gravity should change along the intrusion and amongst the freshwater cases based on the observation made in major comment 3.
    o To improve clarity, we have edited line 398:
        ▪ Calculating the drag coefficient using model output gives Cd with values of order 10−2 to 100. The analytical theory of intrusion distance (L) for an unobstructed water sheet from Robel et al. (2022) is,

$$L = \frac{H^2 g'}{4C_d^2 u_f^2}$$

        where H = 0.05 m is the height of the subglacial environment, g' = 0.20 m/s2 is, and Cd is set to the maximum value within the intrusion. Reduced gravity is referenced to the density difference between the prescribed pure freshwater and pure seawater.

18. Line 427 – "If we anticipate viscous effects to dominate in seawater intrusions under grounded ice, then using the thermal and haline molecular diffusivities as so-called "transport velocities" would be appropriate, similar to the diffusive-convective

framework presented above" – The units (m^2/s vs m/s) are not consistent. To turn the molecular diffusivity into a transport velocity, you need a lengthscale, i.e. the width of the diffusive sublayer. That's the hard part which is not addressed here!

- o We appreciate the reviewer's insight on this. With the model used here, we have a high enough domain resolution to identify the width of the diffusive sublayer. We can define this width to be where the turbulent viscosity is equal to the molecular viscosity. This works because the diffusive sublayer is where molecular diffusivity dominates. Therefore, there should be a point of transition where turbulent viscosity becomes weaker than molecular viscosity. This transition point (i.e. where they are equivalent) will represent the diffusive sublayer's thickness. We have highlighted this limitation when applying this method to large-scale coupled models.

- o We have edited line 427:
  - If we anticipate viscous effects to dominate in seawater intrusions under grounded ice, then using the thermal and haline molecular diffusivities as so-called "transport velocities" would be appropriate. However, this would require knowing the width of the diffusive sublayer which, given computational constraints for coupled ice-ocean models, cannot be resolved.

19. Figure 6 caption – The Stanton number should be $\gamma T /U$, not $\gamma T /C_d$
   - o Fixed.

20. Figure 6 caption last line – are both the dashed and two solid lines from Washam et al 2023?
   - o Both the dashed and solid lines are reported in Table 1 in Washam et al. 2023. However, the dashed line is reported first in Washam et al. (2020) and is cited as such in Table 1 of Washam et al. (2023). We have ultimately removed this figure from the main text.

21. Line 490 – "bolstering the idea that grounding zones are subglacial estuaries (Horgan et al., 2013).
   - o No change is indicated in the comment.

22. Line 566 – seems like a note-to-self.
   - o We appreciate you catching this. It has been removed.

23. Line 594 – either "given by (3)" or "given by equation 3"
   - o Fixed.

24. The appendix is quite bloated. I think some of the figures could be relegated to SI and some should go to the main text. For example, I think Fig A5 is a key result and should go in the main text. Fig A1 could go in SI.

- Based on recommendations from both reviewers, we have shortened the appendix to contain only descriptions of the model (equations, meshing, run-time settings). We have transferred the discussion of the drag coefficient to the main body under the results section. Sensitivity tests and model evaluation alongside other results have been relegated to a supplementary document.

25. Table A1 – inconsistent unit formatting (italics/roman). Look to TC style guide, or use roman which is typical. Units for theta should just be degrees.
    - We have made all units in Roman format and followed the TC style guide in handling denominator values.

26. Table A2 – again, unit formatting. H [cm]
    - We have made all the units in Roman format, following the TC style guide.

27. Figure A5 – the colours are too hard to tell apart.
    - We have refined this figure and included it in the main text.

28. Figures A9 & A10. The figures are labelled "law of the wall" but no interpretation is offered. What is the black vertical line and what does it mean? What would the profiles be expected to look like if a log layer was present? What portion of the flow is being shown? Is y+=0 at the top or bottom of the domain? Do we see a viscous boundary layer, i.e. u+=y+? in addition, it would be much more helpful if y+ was on the y axis, since that's how the model is set up and how all your other profile plots are oriented.
    - In these figures, the vertical profiles are taken over the top half of the domain (2.5 cm from the ice face), to avoid effects from the intrusion interface. Therefore, y+ = 0 is at the top of the domain, where the ice boundary exists. The black line here represents y+ = 30, which would represent the point at which the log-law region of flow would develop and where u+=y+.
    - For these figures, we have expanded on their interpretation and added a characteristic log law profile. This figure has been relegated to the supplementary.

29. Reference 1 (Adusumilli) seems incomplete
    - Fixed.

Works Cited

Pope, S.B. (2000). Turbulent Flows. *Cambridge University Press*, Cambridge, 305-308.

Lai, C.C.K., Socolofsky, S.A. (2019). Budgets of turbulent kinetic energy, Reynolds stresses, and dissipation in a turbulent round jet discharged into a stagnant ambient. *Environ Fluid Mech* **19**, 349–377. https://doi.org/10.1007/s10652-018-9627-3

Washam, P., Nicholls, K. W., Münchow, A., and Padman, L. (2020). Tidal Modulation of Buoyant Flow and Basal Melt Beneath Petermann Gletscher Ice Shelf, Greenland, J. Geophys. Res.-Oceans, 125, https://doi.org/10.1029/2020JC016427

Washam, P., Lawrence, J. D., Stevens, C. L., Hulbe, C. L., Horgan, H. J., Robinson, N. J., Stewart, C. L., Spears, A., Quartini, E., Hurwitz, B., Meister, M. R., Mullen, A. D., Dichek, D. J., Bryson, F., and Schmidt, B. E. (2023). Direct observations of melting, freezing, and ocean circulation in an ice shelf basal crevasse, Sci. Adv., 9, DOI:10.1126/sciadv.adi7638

---

## Author Response (AR2)

Second Response to Editor for Manuscript "High-Fidelity Modeling of Turbulent Mixing and Basal Melting in Seawater Intrusion Under Grounded Ice"

by Mamer, Robel, Lai, Wilson, and Washam

Dear Madeline Mamer and co-authors,

Both reviewers acknowledge the work done to improve your manuscript but still have major concerns. They both find that the salt flux at the ice–ocean interface remains poorly represented, which is a concern for the fidelity of simulated melt rates. They also both notice that this study de-facto investigates a quasi-laminar flow, which is not sufficiently acknowledged. In the end, they think that the modelling tool (ANSYS Fluent), in the way it is implemented here, is not fit for the targeted questions.

Would you be able to address the first concern by clarifying and improving the representation of the salt flux at the ice–ocean interface ($S\_b \neq 0$)?

The simulations currently presented in the manuscript have a boundary salinity that is permitted to evolve in time according to both the intrusion and any simulated meltwater flux. The boundary salinity, $S_b$, is set to the near-ice grid cell (scale of nm's) which has an evolution equation given by B11 (now equation B12 in the revised manuscript). This evolution equation accounts for advective and diffusive transport mechanisms. The boundary salinity is used in equation 1 (the liquidus condition) to set the thermal boundary condition for the ice interfaces. The values for $S_b$ can be clearly seen in Figure 3 (panels B, F, J) at y = 0 and range from 0 ppt to ~ 22 ppt.

In the reference simulations, $S_b$ only evolves due to advection and diffusion from the salt wedge entering the subglacial domain and the freshwater layer exiting the subglacial space as a plume. When we run the simulations with melting, $S_b$ is diluted as fresh, cold water enters the domain normal to the ice boundary. The dilution occurs because Fluent is a finite volume solver and so balances inflows and outflows across each control volume in the grid, while conserving overall mass balance in the domain via a pressure outlet (red arrow Figure 1).

We have made edits to pages 4-5, page 16, and page 28 to clarify this confusion.

For the other concerns, I think that the caveats could be acknowledged in the abstract, the title could be revised without claiming "high-fidelity", and the implications of these caveats could be thoroughly discussed in section 4.

We have edited the title to be:

"Modeling Mixing and Melting in Laminar Seawater Intrusions Under Grounded Ice."

and amended line 5 in the abstract:

"In this study, we investigate turbulent mixing of quasi laminar intruded seawater and glacial meltwater under grounded ice using a computational fluid dynamics solver."

Other mentions of ANSYS Fluent being a high-fidelity solver have been removed.

Furthermore, to address concerns about the simulations being quasi-laminar, we have added to line 81:

"In the experimental set up in this study, we only consider quasi laminar flow in order to facilitate comparison to previous studies (Wilson et. al 2020, Robel et. al 2022) which find that subcriticality (Fr < 1) is required for intrusion development ."

Other concerns by the reviewers including buoyancy impacts on turbulence, the language on diffusion driven melting, adequacy of salt diffusive transport, and issues of interfacial shear instabilities have been addressed in the individual reviewer responses.

I expect a point-by-point response, at least to the major comments by the two referees. If needed and to save time, we can exchange by email on the way forward for some of these comments or for the manuscript in general.

Best regards

Nicolas Jourdain

Second Response to Reviewer 1 for Manuscript "High-Fidelity Modeling of Turbulent Mixing and Basal Melting in Seawater Intrusion Under Grounded Ice"

by Mamer, Robel, Lai, Wilson, and Washham

We appreciate the reviewer's suggestions and thorough comments on the manuscript. We have addressed their concerns below. The reviewer's original text is written in black, and our responses are written in red.

The authors have significantly improved their paper and I can see that how their study will contribute to the understanding of grounding zone dynamics. However, I would like to see additional improvements before it is published. My main criticism is on the boundary conditions implemented, which never really faithfully represent ice-ocean interactions. I believe that the problem being solved isn't a straightforward approximation of the ice-ocean interaction problem, because the authors have to handle the limitations of the Ansys fluent solver. I have made suggestions below to improve discussions of differences (and similarities) between the target problem and the one solved.

==========
==========
General main comments
==========
==========

1) One key limitation of the present work is that the turbulence closure is independent of stratification. This choice may explain why changing the equation of state has limited impact on the dynamics. It should be stated in conclusions that future work should consider more realistic equations of state as the equation of state might significantly impact double diffusive effects, which might be important at low Reynolds.

Buoyancy effects are included in turbulence production by a buoyancy source term represented by $G_b$ in the evolution equation for turbulent kinetic energy (eq B5). Since eddy viscosity is a function of turbulent kinetic energy via the chosen turbulent closure (eq 3), inherently stratification (and any buoyancy effects) will be represented. For the sake of readability, we did not explicitly define this term in the appendix, however, we have now edited line 592 to include:

"Buoyancy effects are included in the evolution equation for turbulent kinetic energy by a source term, $G_b$, which can be solved for by:

$$G_b = g_i \frac{\mu_T}{\rho Pr_T} \frac{\partial \rho}{\partial x_i},$$

where $g_i$ is the component of gravity in the $i^{th}$ direction, $\mu_T$ is the turbulent viscosity, and $Pr_T$ is the turbulent Prandtl number (0.85 default value). The density gradient $\frac{\partial \rho}{\partial x_i}$ is taken over the $i^{th}$ direction. Buoyancy effects are not included in the dissipation equation due to a higher degree of uncertainty (ANSYS 2022)."

2) It should be clearly stated (in abstract and anywhere else appropriate) that this study explores subglacial flows that are quasi laminar (hence in agreement with earlier studies by e.g. Wilson et al 2020?). This statement would certainly make any inspection of the turbulence closure scheme less critical than expected for environmental flows. To support this point, could the authors show a map of the time-averaged TKE/KE, with TKE the pointwise turbulent kinetic energy and KE the resolved kinetic energy? (note that I already greatly appreciated the plot of turbulent viscosity) It might be possible to find in the literature papers discussing the stability of two-layer exchange flows. In fact, the authors might want to have a look at some of the papers by Adrien Lefauve on the inclined duct experiments, which features a two-layer exchange flow (e.g. "Buoyancy-driven exchange flows in inclined ducts" JFM 2020).

In previous theoretical studies of subglacial intrusions (e.g. Wilson et. al 2020 and Robel et. al 2022), the Froude number (equivalent to $u / \sqrt{g'H}$ ) has to be subcritical ($< 1$) for an intrusion to develop. For the geometry of interest in this study, this subcriticality requirement requires subglacial discharge velocities $< 0.05$ m/s for an intrusion to develop. This is why all flows explored here are quasi laminar. To explore more turbulent flows in an intrusion regime (Reynolds number $> 2500$) the domain would have to be a magnitude greater than tested here, which would be much more computationally expensive.

We have edited the title and abstract to represent this choice of flow domain.

New title: "Modeling Mixing and Melting in Laminar Seawater Intrusions Under Grounded Ice."

Edited line 5 in the abstract:

"In this study, we investigate turbulent mixing of quasi laminar intruded seawater and glacial meltwater under grounded ice using a computational fluid dynamics solver."

Furthermore, at line 81 we have added:

"In the experimental set up in this study, we only consider quasi laminar flow in order to facilitate comparison to previous studies (Wilson et. al 2020, Robel et. al 2022) which find that subcriticality (Fr $< 1$) is required for intrusion development."

3) More generally, it would be good to be able to disentangle quantitatively turbulent-mixing effects from laminar-geometry effects (i.e. having proper diagnostics for either types of effects). In particular, I find that the idea of a "well mixed" water column (L259) lacks support for turbulent mixing effects.

[Figure]

**Figure R1**. Profiles of turbulent diffusivities of heat (panels A-C) and salt (panels D-F) normalized by their respective molecular diffusivities from the simulation's last time step. Panels A and D are for $u_f = 5.0$ cm/s, panels B and E are for $u_f = 0.5$ cm/s, and panels C and F are for $u_f = 0.05$ cm/s. The simulations presented in this figure are for the medium turbulent mixing case where $C_\mu = 0.09$. The plotted grey line represents unity, where turbulent diffusivity and molecular diffusivity are equivalent. Darker profiles are near the grounding line and lighter profiles are upstream into the subglacial space along the intrusion distance. Values greater than one indicate turbulent mixing is stronger than diffusive mixing and vise versa.

We appreciate the reviewer bringing this to our attention. Here, we have plotted the turbulent diffusivities of heat and salt normalized by their respective molecular values. In general, turbulent mixing plays a relatively negligible role in transporting heat (Panels A-C), however, has more significance in transporting salt (Panels D-F). Notably, near the boundaries, molecular diffusion dominates as expected in a typical boundary layer. As this reviewer comment suggested, the well mixed behavior of the lowest subglacial discharge (i.e. Figure 3 panels I-L) is due to similar advective and diffusive timescales, not turbulence induced mixing.

We have edited lines 269-272 to clarify this:

"For the lowest freshwater flux, the diffusive and advective timescales are of the same order, which leads to homogeneity in the vertical distribution of heat and salt (Figure 3I-L). In estuaries, strong tides generate shear at the bed, increasing turbulence and vertical mixing

(Montanga et al. 2013). Tides may play a similar role in subglacial seawater intrusion regimes, however, that is not simulated here."

4) I would like to see the three exact equations for the liquidus, dilution and cooling rates at an ice-water boundary in the main text and a discussion of how these three equations are approximated and implemented in the RANS model (this might be the comment I care most about).

The liquidus condition is given by equation 1 at line 121. This equation is used to set the thermal boundary conditions of the ice boundaries.

We have modified line 120 to explicitly state this:

"The ice wall boundaries have a pressure and salinity-dependent thermal boundary condition represented by the liquidus condition:"

The heat flux is calculated via heat conservation at the ice boundary and is given by equation 4 at line 207. This melt rate leads to a mass flux of fresh (zero salinity), cool (at the pressure-salinity dependent freezing point) water that dilutes and chills the fluid domain.

The boundary salinity (e.g. $S_b$ in eq. 1) is set to the boundary adjacent grid point (scale of nanometers) and is permitted to evolve via its own evolution equation (eq. B12). The values for $S_b$ can be clearly seen in Figure 3 at $y = 0$ which range from 0 ppt to ~ 22 ppt. The incoming freshwater set by the melt rate (eq. 4) has zero salinity since it is representing 'melted' ice and therefore dilutes the salinity of the ice-adjacent grid cell (e.g. $S_b$).

To demonstrate how the dilution works, we can revisit that fluent is a finite volume solver, which means it balances inflows and outflows of each finite mesh element and conserves overall domain mass via a pressure outlet (red arrow in Figure 1). Dilution of the near-ice grid cells ($S_b$'s in our model) happens because of this balancing. For example, we consider a simpler case of the salinity evolution equation which only considers advection across two dimensions (ANSYS Fluent considers more dynamics as shown in equation B12):

$$\frac{DS_b}{Dt}\rho_b = -u\nabla\rho_b S_b$$

In discretized form at steady state this would look like:

$$\dot{m}S_i + u_{b-1}S_{b-1} + u_bS_b = 0$$

Where the melt rate (~$10^{-6}$ m/s) is given by $\dot{m}$ and enters the grid point normal to the boundary and $u_{b-1}$ (~$10^{-3}$ m/s) represents the inflow velocity and $u_b$ represents an outflow velocity. The ice salinity is represented by $S_i$ (0 ppt), the inflowing salinity is given by $S_{b-1}$ (34 ppt), and the unknown local grid point salinity is $S_b$. The outflowing velocity must balance in the two inflowing ($\dot{m}$ and $u_{b-1}$) and is therefore slightly greater than $u_{b-1}$. The boundary salinity will therefore be anywhere between the 34 ppt and 0 ppt based on the relative fluxes of melting and background inflow velocity. In the model, more complicated dynamics of diffusion are also occurring which will modify the boundary salinity in addition to the advective effects demonstrated in this example.

We recognize the misunderstanding of salinity boundary condition is due to our previously poor explanation in the methods, especially the misleading line 112 in the original manuscript where we state, "Both ice faces have zero salinity and no salt diffusion across the boundary." This line's intention was to state the ice salinity is zero and we don't diffuse ice into any artificial ice block. However, its placement reads as the boundary salinity, $S_b$, is zero, which is not true in these simulations.

We have made edits to line 123 to clarify this confusion:

"The boundary salinity, $S_b$, is the salinity of the cell filled with water nearest to the ice face and is permitted to evolve during the simulation via the evolution equation B12. For the reference simulations, the boundary salinity evolves in time and space due to advection and diffusion of the subglacial discharge and seawater intrusion. In the simulations with melting, an additional source of freshwater is injected into the near-boundary grid cells, actively freshening the boundary salinity."

And added to the appendix at line 621:

"The boundary salinity, $S_b$, is set to the near-ice grid cell (scale of nm's) which has an evolution equation given by B12. This evolution equation accounts for advective and diffusive transport mechanisms. The boundary salinity is used in equation 1 (the liquidus condition) to set the thermal boundary condition for the ice interfaces. In the reference simulations, $S_b$ only evolves due to advection and diffusion from the salt wedge entering the subglacial domain and the freshwater layer exiting the subglacial space as a plume. For the simulations with melting, $S_b$, is diluted as fresh, cold water enters the domain normal to the ice boundary. This dilution occurs because Fluent is a finite volume solver and so balances inflows and outflows across each control volume in the grid, while conserving overall mass balance in the domain via a pressure outlet (red arrow Figure 1). The thermal boundary condition (given by equation 1) adjusts based on this dilution. Similarly, the chilling of near-ice waters occurs due to local injection of meltwater at the freezing point."

5) The lack of salt flux (sink) at the ice-ocean interface due to melting remains the biggest model limitation. Arguably the salt flux would be small where the ice is in contact with freshwater but would become non negligible where the seawater intrudes close to the ice. The implementation of a buoyancy flux to mitigate this limitation is interesting but the switch to a free-slip boundary is unfortunately far from satisfactory.

We recognize that switching the boundary condition to one that allows for a mass flux of fresh, cool water is not the ideal approach to modeling this boundary, however the simulations including melt are merely an exploratory case to investigate what would happen under a scenario where dilution and cooling occurs. We agree this is a model limitation and have acknowledged in the paper – ultimately, we have tried to circumvent this (i.e. a no melting and a free slip case given by the dashed red line in figure 2) to disentangle the kinematic boundary condition effects and buoyancy flux effects.

6) I still find the motivation for the downward mass (do you mean salt?) flux at the ice base unclear. Am I right to think that your motivation is that your reference model, which you might want to coin "without interfacial salt flux", lacks a salinity flux at the ice base because you impose a zero salinity gradient, and that you aim to compensate it in your "with interfacial salt flux" model? If so, should we think that you turn the salt boundary condition from dz(S)=0 to dz(S) proportional to \dot{m}Sb with Sb the boundary salinity (how is this boundary salinity prescribed then?)? Do you do the same for temperature, i.e. imposing a heat flux proportional to \dot{m}Tb? Either way, please clarify all related sections, which are very important and I believe particularly confusing because you have to deal with Ansys Fluent's lack of flexibility with boundary conditions.

The motivation of the downward mass flux of cool, fresh meltwater, at a rate specified by equation 4, is to simulate the effects of an added buoyancy flux on the persistence and structure of an intrusion. The simulations that serve as the 'reference state' don't include any melting dynamics and therefore don't require a salt flux at the ice boundary. We have modified line 89 to clarify our intention with the melt-enabled simulations:

"We also simulated melting induced by seawater intrusion to investigate how this secondary source of buoyancy affects intrusion persistence and structure."

The boundaries in the 'reference state' have no-slip kinematic conditions with a thermal condition dependent on the local liquidus equation (re eq 1). The purpose of these reference simulations is to compare to previous theory (e.g. Wilson 2020 and Robel 2022) which do not consider melting effects.

We have added to line 84 to properly introduce these "reference simulations":

"The simulations in which we range over subglacial discharge velocities with medium turbulent mixing (e.g. $C_\mu = 0.09$) are our reference simulations to which we will compare all other results. The purpose of these reference simulations is to compare with previous theory that consider

quasi laminar flow and no melting effects. Further testing on potential intrusion control variables is also considered and explored in detail."

When melting is 'enabled' the ice boundary turns into a velocity inlet where fresh, cold water is input into the domain at the rate prescribed by equation 4 (heat conservation at the boundary). The boundary temperature is set by the liquidus condition (eq. 4). The boundary salinity, which is identified as the near-ice grid point's salinity, is solved for by its own evolution equation (eq B12) and transported appropriately.

7) I know this might sound subjective but I do not think that "high-fidelity" is an accurate description of the ice-ocean model considered in this work as there are clear limitations to the turbulence closure scheme being used and the representation of ice-ocean boundary conditions (To be precise I would only consider DNS and LES to be high fidelity, if they can justify the problem solved as a faithful representation of the expected dynamics). Thus I think that "high-fidelity" should be removed from the title of the paper. That is unless the authors make the point that the flows are laminar hence do not require any subgrid scale parameterizations and find a way to implement the exact boundary conditions at the ice-ocean interface (liquidus, dilution, cooling). In fact, since the Reynolds are so low, could you not solve the exact governing equations (i.e. without having to implement a closure scheme)?

We have modified the title based on suggestions from both reviewers:

"Modeling Mixing and Melting in Laminar Seawater Intrusions Under Grounded Ice."

Elsewhere in the text where we state ANSYS Fluent as a high-fidelity model, we have also removed the term.

We have tested each subglacial discharge scenario without a closure scheme (e.g. a laminar case) as shown by the green transects in Figure 2. The goal of this project is to test the effects of turbulent mixing on the ability for seawater to intrude into a buoyant subglacial discharge system, hence why we employed a turbulence closure.

8) The momentum equation (B3) in appendix doesn't have any buoyancy term. Could the authors make sure that buoyancy is considered in their model?

Thank you for pointing out this typo, it is included as a body force with the full variable density considered based on the equation of state used (eq. 2). We have amended equation B3 to:

$$\frac{\partial}{\partial t}(\rho u_i) + \frac{\partial}{\partial x_i}(\rho u_i u_j) = -\frac{\partial p}{\partial x_i} + \frac{\partial}{\partial x_j}\left[\mu\left(\frac{\partial u_i}{\partial x_j} + \frac{\partial u_j}{\partial x_i} - \frac{2}{3}\delta_{ij}\frac{u_l}{x_l}\right)\right] + \frac{\partial}{\partial x_i}\left(-\rho\overline{u_i'u_j'}\right) + \rho g_i$$

The buoyancy term is represented by $\rho g_i$.

9) Some wording choices are very confusing and should be changed, such as "heat limited" or "melt activated". Should you maybe say for the latter "with interfacial salt flux"?

This melt flux does more than dilute the near ice water, it also acts to chill it, so the suggested change in terminology would not be entirely accurate. The use of the term "melt activated" is to distinguish against the reference simulations which have no melting. We have modified this term to be "with melting" throughout the text.

We chose the term "heat limited" to demonstrate that the melting we simulate is only driven by thermal forcings and not dissolution. We have modified this term to be "thermally driven" to describe melt driven mainly by thermal forcings and removed discussion of diffusion-forced melting.

```
==========
==========
```
Specific comments (chronological order)
```
==========
==========

==========
```
Abstract
```
==========
```

L9: I don't understand "Basal melting from seawater intrusion produces buoyant

meltwater which may act as an important negative feedback by reducing near-ice thermohaline gradients." What is the feedback mentioned?

L12: In particular, how it connects to "We conclude that, in times or places when subglacial discharge is slow, seawater intrusion can be an important mechanism of ocean-forced basal melting of marine ice sheets.

We understand this specific comment to be addressing a supposed inconsistency between stating intrusion-induced melting might create a negative feedback that suppresses further melting hinting at an intrusion's potential unimportance in ocean-forced basal melting of marine ice sheets. Our intentions with these sentences are to point out that intrusion can cause melting, which given a flat geometric configuration can lead to stable stratification and blanketing of the ice by a layer of cold fresh water. However, that is unlikely in the real world. In addition, there are other drivers and modulators of intrusion, that are likely to impact an intrusion's ability to cause melting. We have modified these sentences (text in red) to communicate our intention better:

L9: "Basal melting from seawater intrusion produces buoyant meltwater which may create a negative feedback by chilling and freshening near-ice water therefore reducing further melting, however this remains unquantified."

L12: "We conclude that, in times or places when subglacial discharge is slow, seawater intrusion can be an important mechanism of ocean-forced basal melting of marine ice sheets when considering added geometric complexities and ocean conditions."

==========
Introduction
==========

L39: Could you clarify "may lead to a run-away positive feedback if melting outpaces ice advection (Bradley and Hewitt, 2024)."?

In this paper, the authors found that if melting of ice happens 'faster' than ice is advected in to replace the melted ice then the subglacial conduit can grow unabated increasing the intrusion distance, which would lead to more melting, more subglacial conduit growth more intrusion etc.

We have modified this line to be:

"More recent work suggests that shear-driven melting beneath grounded ice can enlarge the subglacial cavity, enhancing seawater intrusion and potentially triggering a runaway positive feedback (Bradley and Hewitt, 2024). This feedback arises when melting exceeds ice advection, preventing upstream ice from replenishing the ablated region and allowing the conduit to grow unchecked. Bradley and Hewitt (2024) identified a regime in

which seawater intrusions could become unbounded, effectively "intruding infinitely.""

==========
Methods
==========

L80-81: Could you indicate the thermal driving associated with the initial open ocean and subglacial water conditions?

The ocean inlet temperature is $T_o = 0.5$ deg C.

The subglacial discharge temperature is set by equation 1 at z = 1000m ($T_f = -0.6778$ deg C).

Therefore the thermal forcing is $T_{tf} = 0.5 - (-0.6778) = 1.1778$ deg C.

We have added this to line 92:

"The thermal driving associated with the initial conditions is approximately 1.2 deg C and the reduced gravity is 0.23 m/s$^2$."

L112: I don't understand "Both ice faces have zero salinity and no salt diffusion across the boundary. For the warmer fluid regime prescribed in these experiments, a non-diffusive boundary is appropriate since thermally-driven ice loss will dominate."

We have identified the first line is this comment to be the main source of confusion for the boundary salinity issue. The intention of this line was to say the ice salinity is zero and we don't account for any ice mass loss by salt diffusion.

We have made edits to line 123 to clarify this confusion:

"The boundary salinity, $S_b$, is the salinity of the cell filled with water nearest to the ice face and is permitted to evolve during the simulation via the evolution equation B12. For the reference simulations, the boundary salinity evolves in time and space due to advection and diffusion of the subglacial discharge and seawater intrusion. In the simulations with melting, an additional source of freshwater is injected into the near-boundary grid cells, actively freshening the boundary salinity."

And added to the appendix at line 621:

"The boundary salinity, $S_b$, is set to the near-ice grid cell (scale of nm's) which has an evolution equation given by B12. This evolution equation accounts for advective and diffusive transport mechanisms. The boundary salinity is used in equation 1 (the liquidus condition) to set the thermal boundary condition for the ice interfaces. In the reference simulations, $S_b$ only evolves due to advection and diffusion from the salt wedge entering the subglacial domain and the freshwater layer exiting the subglacial space as a plume. For the simulations with melting, $S_b$, is diluted as fresh, cold water enters the domain normal to the ice boundary. This dilution occurs because Fluent is a finite volume solver and so balances inflows and outflows across each finite element in the grid, while conserving overall mass balance in the domain via a pressure outlet (red arrow Figure 1). The thermal boundary condition (given by equation 1) adjusts based on this dilution. Similarly, the chilling of near-ice waters occurs due to local injection of meltwater at the freezing point."

For the second sentence identified here, if the fluid is supercooled or lacks sufficient thermal forcing, melting can still occur through dissolution. In this process, salt from the surrounding

fluid enters the ice interface, lowering the local freezing point. This effectively destabilizes the ice structure and leads to its gradual erosion, even in the absence of significant heat input.

We have removed this from the edited manuscript to ensure simplicity and inhibit further confusion.

L138: Isn't "salt diffusion" out of place here? I don't understand "As salt diffuses in the medium, it also transfers heat due to its unique thermal properties, and therefore must also be included." This sounds like a generic sentence not relevant to the chosen configuration. The energy equation should only include time variations, advection and conduction. Also, viscous heating is a highly unusual term for environmental flows.

We have modified this line to:

"Since salt is tracked as an active tracer it transports enthalpy associated with its specific heat and concentration gradients. This must be accounted for in the energy equation Fluent solves."

The viscous heating term is not accounted for in our model setup (i.e. Fluent does not solve for it), however we included it in the appendix since that is how the equation appears in the solver documentation. We have removed this term ($\tau_{eff}\vec{u}$) from equation B11 and the associated text describing the term in the lines shortly after.

L152: This sentence "While the fluid density is different, the salinity and temperature of the fluid remain unchanged since they have their respective transport equations." Is somewhat misleading. My understanding is that you're saying "changing the equation of state" doesn't change the dynamics, which is possible, but likely model dependent. To see a change you would need to have processes (resolved or parameterized) that depend on density, or stratification. One key question then is: do mixing rates (or turbulent diffusivity) depend on stratification in your model? Your description of the turbulent closure scheme used suggests that there is not stratification effects considered, hence the lack of impact of the EOS on the dynamics.

The kinetic energy evolution equation does account for buoyancy effects via $G_b$ which impacts the eddy viscosity via the turbulence closure which feeds back into the momentum equations. In this regime with this model setup, using a higher order equation of state did not noticeably change the dynamics. We have modified the text as shown in major comment 1.

L180: I find your discussion of basal melting very confusing, i.e. for instance, "In some simulations, we also simulate the added buoyancy flux resulting from the heat-limited melting scenario. Here, we neglect melting driven by dissolution, instead focusing on melting driven by thermal equilibrium at the ice boundary. Since the thermohaline conditions of the fluid domain are non-sub-freezing, the neglect of dissolution-induced melting is justified." I find the distinction between melting and dissolution confusing and unjustified for environmental flows such as here (because they are turbulent). My understanding is that fast ice losses due to high temperature flows and slow salt diffusivity resulting in fresh meltwater layers are coined ice

melting, while slow ice losses, allowing for salt diffusion into the meltwater, correspond to ice dissolving. Here the ice is probably always melting, i.e. ice losses are controlled by heat fluxes. You seem to say so but what do you mean by "Heat limited"?

The purpose of these sentences is to provide a justification for why we are not modeling dissolution driven melting. In dissolution, salt from the surrounding fluid enters the ice interface, lowering the local freezing point, destabilizing the ice structure, effectively 'eroding' the ice. To the reviewers last point, in the thermal regime set by the seawater, ice loss is dominated by heat fluxes, which is our intention with the phrase 'heat-limited'.

We have edited the manuscript by removing these sentences for simplicity and to inhibit further confusion.

==========
Results
==========

Fig. 2: I am surprised that the laminar simulation isn't closer to the RANS model for the low subglacial discharge velocity. Could you comment?

The lowest subglacial discharge case shows the highest sensitivity to turbulence modeling choices, as seen in the contrast between the high, medium, and low turbulent mixing cases as well as the laminar simulation. At this weak forcing level, the flow itself generates little inherent turbulence, so the behavior of the intrusion is governed largely by how turbulence is modeled. In the RANS simulations, turbulence is generated primarily by boundary shear (ice and rock walls) and interfacial shear between the buoyant subglacial discharge and the underlying seawater. This mixing promotes entrainment, allowing the intrusion to extend farther into the domain. The resulting tail of relatively fresh water is visible in Figure 2 over the last 5 meters of the intrusion. In contrast, the laminar case cannot capture this turbulence or entrainment, and as a result, the intrusion stalls sooner.

We have added to line 240:

"For the slowest freshwater case, the intrusion is reduced in the absence of turbulent mixing due to the lack of entrainment between the seawater intrusion and the buoyant subglacial discharge. When modeled, this entrainment extends the intrusion by generating a tail of relatively low-density water."

L255: I am confused. I don't see how having a well-mixed column for low subglacial discharge (presumably low shear) is analogous to tidally forced estuaries subject to high shear? Isn't it that for low velocity the freshwater discharge is unable to fill or "occupy" the subglacial conduit, which becomes largely filled with open ocean water? Or is it really linked to turbulence (is there a diagnostic you can show to support this?)?

The decrease in stratification as the subglacial discharge is decreased is due to the convergence of the advective and diffusive timescales and not the imposed turbulence by boundary shear (as shown in Figure R1).

In line with our response to major comment 3, we have modified lines 269-272 to state:

"For the lowest freshwater flux, the diffusive and advective timescales are of the same order which leads to homogeneity in the vertical distribution of heat and salt (Figure 3I-L). In estuaries, strong tides generate shear at the bed increasing turbulence and vertical mixing (Montanga et al. 2013). Tides may play a similar role in subglacial seawater intrusion regimes, however, that is not simulated here."

Fig. 4 and related text: isn't the high Cd simply an indication that the flow is laminar? Could you comment?

The drag coefficient found in these simulations matches what would be expected given the subglacial discharge and geometry of the subglacial space per engineering literature. For laminar flow the drag coefficient can be estimated by $C_d = 16/Re$ where $Re$ is the Reynolds number. For the middle subglacial discharge case tested here (blue lines in Figure 4) the Reynolds number is ~250. This would give $C_d$=0.064. The simulated drag coefficient (~0.05) at locations away from the intruded seawater (at flat part of the curves shown in figure 4) agrees well with this estimate (=0.064).

The interesting part of Figure 4 is the transition to a higher drag coefficient over the length of intrusion (everything downstream of the scatter points in Figure 4) which likely is resulting from transition to a turbulent regime driven by shear at the interface of the intrusion wedge and buoyant outflow. This value of drag coefficient has less straightforward ways of estimating due to the complexities of flow within the intrusion regime.

We have edited line 299 to help clarify this:

"The high drag coefficients simulated here upstream of the intrusion regime are in line with the expected values for laminar flows with these Reynolds numbers. However, over the regime of intrusion the drag coefficient increases by nearly an order of magnitude in all cases tested. The increased drag coefficients over the intrusion are more difficult to estimate and likely result from enhanced turbulence in the interfacial shear layer between the intrusion and the buoyant subglacial outflow.

L323: It's a bit odd to mention that some ocean models use temperature 10 km away from the boundary to estimate melt rates. Isn't few meters to few tens of meters?

Grid cells of this size may be used if one is not directly modeling ice shelf cavities and therefore use ocean forcings from 10's km from the grounding line/underneath ice shelves to estimate a melt rate. In simulations of ice shelf cavities the vertical grid cell size can be on the order of 100 meters and the horizontal grid cell size might be order 10 km. In this case, there arises the question of what to do in the limit your vertical grid cell size approaches 0, which is essentially what happens at the grounding zone, which is the region of interest in this study.

L343: Taking h has "the thickness of the viscous sublayer (Holland and Jenkins, 1999)." is somewhat surprising to me (I would assume somewhere in the log layer) and might be worth double checking.

Thank you for pointing this out, this error has been fixed and is in line with the phrasing used in Holland and Jenkins 1999.

Edited this line to be:

"[…] the thickness of the boundary layer (Holland and Jenkins, 1999)."

==========
Appendices
==========

I assume that equation (B3) is incorrect, and the buoyancy/gravity term is missing? Please double check.

Thank you for pointing this out, we have added the buoyancy term $\rho g_i$ to the right-hand side of the equation.

Why do you use different notations between equations (B3) and (B4)?

We have modified equation B4 to have lowercase velocities similar to B3 with an overbar to denote average quantities:

Equation (B8) seems to have a typo.

We have fixed the parenthesis exponent error in equation B8.

Could you confirm that species diffusion is set to 0 in (B10) and does not impact the heat equation?

The species diffusion term is not zero, however it does not impact the overall heat equation significantly since its magnitude is orders smaller than other terms.

The energy transport by species diffusion is set by $h_{salt}J_{salt}$ where $h_{salt}$ is the enthalpy of salt and $J_{salt}$ is the diffusive flux. The enthalpy can be found by:

$$h_{salt} = \int_{T_{ref}}^{T} c_{p,salt}dT = c_{p,salt}(T - T_{ref})$$

Here, the heat capacity of salt ($c_{p,salt}$) is set to $\sim 10^3$ J/kg K and the reference temperature ($T_{ref}$) is a default value used by the solver set to 298.15 K. Given the temperature of our simulation is approximately 273 K, the enthalpy of salt is about $2.5 \times 10^4$ J/kg. The diffusive flux can be solved for by:

$$J_{salt} = \rho_W D_s \nabla S$$

The fluid density ($\rho_W$) is approximately $10^3$ kg/m³, diffusivity of salt is approximately $10^{-9}$ m²/s, and the salt gradient ($\nabla S$) can be approximated for the middle subglacial discharge flux case to be 0.034 / 5 m. This would result in the energy transport by species diffusion to be about $10^{-4}$ W/m².

Comparably, the other term in equation B11 that accounts for energy transport by thermal diffusion ($k_{eff}\nabla T$) can be estimated given an effective thermal diffusivity $\sim 6 \times 10^{-1}$ W/mK (includes both molecular diffusivity and turbulent diffusivity) and thermal gradient ($\nabla T$) of $\sim 1$ deg C / 5 m. This results in energy transport by thermal diffusion of order $10^{-1}$ W/m².

Therefore, even though species diffusion is contributing to overall energy transport in the system, it is much smaller than other contributions and does not impact the results significantly.

We have added to line 612:

"As salt diffuses in the medium, it also transfers heat due to its unique thermal properties and therefore must also be included. The contribution to heat transport from salt has negligible impacts since it is not a leading order term in equation B10 and therefore tracking species contribution to energy transport does not significantly change the results.

L686: Could you discuss to what extent stratification effects are considered in your turbulence closure scheme (to support your explanation "Turbulent viscosity is greatly reduced over the length of the intrusion, likely due to enhanced stratification suppressing mixing.")?

There is a buoyancy term $G_b$ in the turbulent kinetic energy evolution equation which allows for stratification effects to impact turbulent viscosity (aka eddy viscosity). We have added a few sentences to the appendix to describe this impact:

"Buoyancy effects are included in the evolution equation for turbulent kinetic energy by a source term, $G_b$, which can be solved for by:

$$G_b = g_i \frac{\mu_T}{\rho Pr_T} \frac{\partial \rho}{\partial x_i}$$

Where $g_i$ is the component of gravity in the $i^{th}$ direction, $\mu_T$ is the turbulent viscosity, and $Pr_T$ is turbulent Prandtl number (0.85 default value). The density gradient $\frac{\partial \rho}{\partial x_i}$ is taken over the $i^{th}$ direction. It is not included in the dissipation equation due to a higher degree of uncertainty (ANSYS 2022)."

References

ANSYS: ANSYS Fluent - CFD Software | ANSYS, http://www.ansys.com/products/fluids/ansys-fluent, 2022.

Montagna, P. A., Palmer, T. A., and Pollack, J. B.: Hydrological Changes and Estuarine Dynamics, Springer, New York, NY, 1st edn., https://doi.org/10.1007/978-1-4614-5833-3, 2013.

Robel, A. A., Wilson, E., and Seroussi, H.: Layered seawater intrusion and melt under grounded ice, The Cryosphere, 16, 451–469, https://doi.org/10.5194/tc-16-451-2022, 2022.

Wilson, E. A., Wells, A. J., Hewitt, I. J., and Cenedese, C.: The dynamics of a subglacial salt wedge, J. Fluid Mech., 895, A20, https://doi.org/10.1017/jfm.2020.308, 2020.

Zhu L, Atoufi A, Lefauve A, Taylor J. R., Kerswell, R. R., Dalziel S. B., Lawrence G. A., Linden P. F. Stratified inclined duct: direct numerical simulations. J. Fluid Mech., 969, A20, doi:10.1017/jfm.2023.502, 2023.

Second Response to Reviewer 2 for Manuscript "High-Fidelity Modeling of Turbulent Mixing and Basal Melting in Seawater Intrusion Under Grounded Ice" by Mamer, Robel, Lai, Wilson, and Washam

We appreciate the reviewer's constructive comments and aim to address their concerns below. Reviewer's responses original text is written in black, and our responses are written in red.

Review #2 Mamer et al "High-Fidelity Modeling of Turbulent Mixing and Basal Melting in Seawater Intrusion Under Grounded Ice"

My sincere apologies to the Authors and Editor for taking so long to complete this review.

I appreciate the significant changes that the authors have made, including adding new simulations, in response to the first round of reviews.

Unfortunately, the new information provided suggests to me that the problem formulation and the model are not fit to address the stated aims of the paper, for the following reasons:

A – I don't believe that the RANS model can simulate this particular flow
Firstly, it appears that the simulations are essentially laminar within the intrusion region, based on the similarity between the so-called "turbulent" and "laminar" simulations and Figure D1 which shows that the eddy viscosity is similar to molecular viscosity within the intrusion region (As an aside - I find the non-monotonic relationship between C_mu and turbulent viscosity disturbing!).

It is not entirely clear why the reviewer does not think that the Reynolds-Averaged Navier Stokes Equation are not able to simulate laminar regimes. These equations still apply even in the laminar regime (perhaps especially here, since the closure terms become less important). Mathematically, the Reynolds stress terms appearing in the momentum equations will vanish in the limit of laminar flows. One primary aim of this study is to compare these more complex simulations to prior theory (Wilson et. al 2020 and Robel et. al 2022) in which all cases considered are subcritical (Fr<1), thus limiting the subglacial discharge velocity to essentially laminar conditions. Based on the comments from both reviewers, we have modified the title to be:

"Modeling Mixing and Melting in Laminar Seawater Intrusions Under Grounded Ice."

and amended line 5 in the abstract:

"In this study, we investigate turbulent mixing of quasi laminar intruded seawater and glacial meltwater under grounded ice using a computational fluid dynamics solver."

Furthermore, at line 81 we have added:

"In the experimental set up in this study, we only consider quasi laminar flow in order to facilitate comparison to previous studies (Wilson et. al 2020, Robel et. al 2022) which find that subcriticality (Fr < 1) is required for intrusion development."

The non-monotonic behavior between $C_\mu$ and the turbulent viscosity is consistent with our expectations for non-linear coupling of turbulence models. Turbulent viscosity is linear with respect to $C_\mu$ (e.g. eq 3) given constant turbulent kinetic energy, dissipation, and density. However, this is not the case, since the fields of kinetic energy and dissipation adapt to the closure and then feed back into the turbulence model. Hence the non-monotonic behavior. More simply, turbulence viscosity exists within the turbulent kinetic energy and dissipation evolution equations (eqs. B5 and B6), so they are coupled, which results in non-monotonicity when changing the co-factor in the turbulence closure model.

Unfortunately, based on your simulations alone, you cannot say whether this laminar state is a real feature of this (unusual) system or not. The extreme stratification and small height (and therefore Reynolds number) of the subglacial environment means that turbulence may not develop. However, instability or scouring of the interface, as was seen under certain conditions in the laboratory experiments of Wilson et al 2020, is certainly possible and would influence the flow significantly. However, simulating such features in a stratified flow is not within the capabilities of a RANS model, and is instead the domain of Direct Numerical Simulations (DNS).

We recognize that shear instabilities at the interface of the two layers (Holmboe or Helmholtz instabilities as examples) are possibilities (given the right advective forcings) that would act to increase mixing and are not directly resolved here. However, a recent DNS study investigating stratified flow found that non-laminar interfacial shear layers would occur at higher Reynolds numbers than simulated here (Zhu et. al 2023). Regardless, RANS should be capturing the averaged effect of what these instabilities would lead to in the flow field.

We have added to line 64:

"Interfacial shear instabilities that might be expected of highly stratified flows such as the ones simulated in this study are not explicitly resolved in RANS models, however the averaged mixing effect of such instabilities should be captured. Furthermore, a recent study using direct numerical simulations has identified that such instabilities do not arise at the lower Reynolds numbers we are simulating here (Zhu et. al 2023)."

Since the current literature on seawater intrusion are highly idealized one-dimensional models of two-layered shallow water equations (e.g. no mixing dynamics or vertical resolution), we believe the simulations presented here are helpful in identifying key next questions about the dynamics of this regime. We agree that DNS may be a logical next step, but many robust conclusions are still possible in this modeling configuration.

B – The melt model is inaccurate
I appreciate that, in response to the previous reviews, the authors added the salinity-dependent interface temperature. However, in neglecting to solve the salt conservation equation at the ice-ocean interface, you have two equations (Equations (1) and (4)) and three unknowns (m, Tb, Sb). This is a fundamental flaw in the approach and means that the melt rates will not be accurate.

The boundary salinity is accounted for in the model and permitted to evolve as the model runs. The boundary salinity, $S_b$, is taken to be the near-ice salinity (scale of nm's) and has its own evolution equation (eq. B11 in original manuscript). The values of $S_b$ can be clearly seen in Figure 3 (panels B, F, J) where y = 0. Clearly in this figure, $S_b$ is permitted to evolve to non-zero values. The melt rate is set by the conservation of heat at the boundary and given by equation 4. The liquidus equation that sets the thermal boundary condition (and dependent on $S_b$) is given by equation 1. When melting is turned on and an additional source of fresh, cold water enters the domain normal to the ice boundary, dilution of $S_b$ occurs because Fluent is a finite volume solver and must balance all inflowing and outflowing fluxes for each given grid point. The thermal boundary condition of the ice responds accordingly to this dilution based on equation 1.

We have identified the main source of boundary salinity confusion to be from line 112 in the original manuscript where we state:

"Both ice faces have zero salinity and no salt diffusion across the boundary."

Here, we had intended to say that the ice salinity is zero but recognize the location of this sentence and lack of clarity is misleading.

We have made edits to line 123 to clarify this confusion:

"The boundary salinity, $S_b$, is the salinity of the cell filled with water nearest to the ice face and is permitted to evolve during the simulation via the evolution equation B12. For the reference simulations, the boundary salinity evolves in time and space due to advection and diffusion of the subglacial discharge and seawater intrusion. In the simulations with melting, an additional source of freshwater is injected into the near-boundary grid cells, actively freshening the boundary salinity."

And added to the appendix at line 621:

"The boundary salinity, $S_b$, is set to the near-ice grid cell (scale of nm's) which follows the evolution equation given by B12. This evolution equation accounts for advective and diffusive transport mechanisms. The boundary salinity is used in the liquidus condition (equation 1) to set the thermal boundary condition for the ice interfaces. In the reference simulations, $S_b$ only evolves due to advection and diffusion from the salt wedge entering the subglacial domain and the freshwater layer exiting the subglacial space as a plume. For the simulations with melting, $S_b$, is diluted as fresh, cold water enters the domain normal to the ice boundary. This dilution occurs because Fluent is a finite volume solver and so balances inflows and outflows across each control volume in the grid, while conserving overall mass balance in the domain via a pressure outlet (red arrow Figure 1). The thermal boundary condition (given by equation 1) adjusts based on this dilution. Similarly, the chilling of near-ice waters occurs due to local injection of meltwater at the freezing point."

We recognize there are limitations in how we are modeling melting as described in the discussion. Based on earlier reviews, we attempted to overcome some limitations (e.g. providing a free-slip kinematic boundary condition case to compare the melting cases to). The simulations with melting are merely exploratory cases to look at what would happen to stratification and transport within the intrusion regime given freshening and cooling from local melting.

There appears to be significant confusion from the authors about the role of salt in melting, e.g. Line 182:
"Since the thermohaline conditions of the fluid domain are non-sub-freezing, the neglect of dissolution-induced melting is justified."
The only case in which the salt balance equation can be neglected is if the interface salinity Sb is zero. This only occurs in freshwater, or theoretically if the ice is melting so quickly that that salt transport towards the ice cannot "keep up" with the meltwater flux which would lead the region closest to the ice to become fresh (see figure 1b in Malyarenko et al 2020). So, there is a clear inconsistency between including the interface salinity in your freezing point temperature, but not also including a salt balance equation in the melting calculation.

As noted above, we are not neglecting the salt balance. The intention with line 182 was to justify why we are not modeling melting driven by dissolution. Dissolution occurs when thermal forcing is near zero and so salt can diffuse into the ice crystal, suppressing the local freezing point, effectively "eroding it". Due to comments by both reviewers, we have removed this line from the main text.

C – Salt diffusion does not appear to be correctly represented
Molecular diffusion of salt is of the utmost importance to ice-ocean interactions since this is the way salt is transported across the viscous sublayer adjacent to the ice-ocean interface. In addition, in the absence of turbulence, it is the only mixing process occurring and will govern the evolution of salinity stratification along the salt wedge. Although it is not spelled out in the salinity equation (B11), based on the profiles in Fig. 3 which show that salinity gradients are no steeper than temperature gradients, it appears the salt diffusion is not occurring at the expected slow molecular "rate" based on molecular diffusivity of ~7e-9 m^2/s.

In equation B11 (now B12 in the revised manuscript), the diffusion flux is represented by $-\nabla \vec{J} = -\rho_w D_S \nabla S_i$, where $D_S$ is the diffusion coefficient for salt, here set to 1.5e-9 m²/s. We have updated equation B11 to include the full diffusive term and edited line 620 :

"Where u is the velocity vector, $D_S$ is the diffusion coefficient for salt (set to $1.5 \times 10^{-9}$ m²/s), and R is the production rate from reactions."

Salt is diffusing at a lower rate than temperature throughout the domain. Here we provide two figures from the lowest subglacial discharge scenario ($u_f$ = 0.05 cm/s) with no turbulence closure (i.e. laminar) to demonstrate this. Figure 1 shows a vertical slice of normalized temperature and salinity within the intrusion regime. Note in the upper portion of the intrusion (y > 2.5 cm) heat and salt diffuse at different rates into the freshwater layer that occupies this upper layer of the subglacial space. Figure 2 showcases depth average horizontal transects of normalized temperature and salinity along the intrusion. Once again, the profiles are not

[Figure]

**Figure 1.** Vertical profiles of time-averaged temperature and salinity for the lowest subglacial discharge velocity with no turbulent mixing. Temperature and salinity are normalized via equation 1.

identical, meaning they are diffusing at their own respective rates. Furthermore, the temperature profile is broader and decays to the background value (subglacial value) quicker than salinity does, i.e. it diffuses quicker.

[Figure]

**Figure 2.** Depth-averaged transects of temperature and salinity through the subglacial space. Temperature and salinity are from the time-averaged domain and normalized by equation 1. The grounding point is at x = 2 m.

We normalize temperature and salinity by:

$$\Phi_{norm} = \frac{\Phi - \Phi_{min}}{\Phi_{max} - \Phi_{min}}$$

$$eq.\,1$$

where $\Phi$ is a placeholder for the respective variable being normalized.

Relating these to your specific aims:

"(1) to test previously proposed controls on seawater intrusion distance,
◊ this might be ok, although is questionable given (C)

Figure 1 and Figure 2 demonstrate that salt is diffusing at a slower rate than temperature, and so we are resolving this transport mechanism correctly. We have updated the salt evolution equation to articulate the diffusive flux more thoroughly.

(2) to determine the effects of turbulent mixing on seawater intrusion, and
◊ based on (A) and (C), I don't think you can reasonably do this in your setup

The averaged effects of possible interfacial shear instabilities are captured by RANS models, and existing literature supports that such instabilities do not arise in the Reynolds regime we are testing here (e.g. Zhu et al. 2023). The current state of the field on seawater intrusion are simplified one-dimensional two-layer models and this paper serves to expand on this by providing insight into the vertical structure, stability, and controls on intrusion given the presence of turbulence induced mixing.

(3) to investigate the dynamics of intrusion-induced basal melting
◊ based on (B), this aim has not been addressed.

The boundary salinity is set to the near-ice wall salinity which is permitted to evolve in the model by its own evolution equation (eq. B11 in original manuscript). The dilution of this value happens when cold, freshwater inflows at a rate set by the melt rate (eq. 4). The intentions of the simulations with melting were to explore feedbacks between added buoyancy forcings and persistence of intrusion, not necessarily to produce a constrained value for expected melt rate in the tested regimes of intrusion.

I see two ways forward for this study. One approach would be to address (B) and (C), and focus on the laminar problem. A second approach would be to switch tools and use DNS.

Based on our responses to your three points, we believe this modeling framework is adequate to answer the questions laid out in the beginning of the paper and will provide useful information to those studying seawater intrusion and grounding zone fluid dynamics.

References

ANSYS: ANSYS Fluent - CFD Software | ANSYS, http://www.ansys.com/products/fluids/ansys-fluent, 2022.

Robel, A. A., Wilson, E., and Seroussi, H.: Layered seawater intrusion and melt under grounded ice, The Cryosphere, 16, 451–469, https://doi.org/10.5194/tc-16-451-2022, 2022.

Wilson, E. A., Wells, A. J., Hewitt, I. J., and Cenedese, C.: The dynamics of a subglacial salt wedge, J. Fluid Mech., 895, A20, https://doi.org/10.1017/jfm.2020.308, 2020.

Zhu L, Atoufi A, Lefauve A, et al. Stratified inclined duct: direct numerical simulations. *Journal of Fluid Mechanics*. 2023;969:A20. doi:10.1017/jfm.2023.502